# Bifurcation in brain dynamics reveals a signature of conscious processing independent of report

Claire Sergent [1,2✉], Martina Corazzol[1,2,6], Ghislaine Labouret [1,2,3,6], François Stockart [1,2], Mark Wexler[1,2], Jean-Rémi King [4], Florent Meyniel [5] & Daniel Pressnitzer [4]

An outstanding challenge for consciousness research is to characterize the neural signature of conscious access independently of any decisional processes. Here we present a model-based approach that uses inter-trial variability to identify the brain dynamics associated with stimulus processing. We demonstrate that, even in the absence of any task or behavior, the electroencephalographic response to auditory stimuli shows bifurcation dynamics around 250–300 milliseconds post-stimulus. Namely, the same stimulus gives rise to late sustained activity on some trials, and not on others. This late neural activity is predictive of task-related reports, and also of reports of conscious contents that are randomly sampled during task-free listening. Source localization further suggests that task-free conscious access recruits the same neural networks as those associated with explicit report, except for frontal executive components. Studying brain dynamics through variability could thus play a key role for identifying the core signatures of conscious access, independent of report.

[1] Université de Paris, INCC UMR 8002, 75006 Paris, France. [2] CNRS, INCC UMR 8002, Paris, France. [3] Laboratoire de Sciences Cognitives et Psycholinguistique, Département d'Etudes Cognitives, École normale supérieure, EHESS, CNRS, PSL University, Paris, France. [4] Laboratoire des Systèmes Perceptifs, Département d'études cognitives, École normale supérieure, PSL University, CNRS, Paris, France. [5] Cognitive Neuroimaging Unit, CEA, INSERM, Université Paris-Sud, Université Paris-Saclay, NeuroSpin Center, Gif/Yvette, France. [6] These authors contributed equally: Martina Corazzol, Ghislaine Labouret. ✉email: claire.sergent@u-paris.fr

Neural correlates of conscious perception have been extensively sought by contrasting neural activity for the same external stimulus when it is reported as perceived or not perceived[1,2]. Clear changes in neural activity are associated with reports of conscious perception: activity in sensory regions increases[3–5], processing involves a wider network of areas[3,5,6], global functional connectivity increases[7,8] and it is accompanied by late sustained activity[8–10]. An outstanding challenge, however, is to go beyond correlates of behavioral reports and investigate whether there can be neural signatures of conscious perception even when no overt response is required—which is, it should be pointed out, the general case for our everyday conscious experience[11,12]. Identifying such signatures would rule out potential confounds associated with overt responses, such as decision-making processes that are not necessarily required for conscious processing[13]. A neural signature of conscious access without report would also be of considerable clinical value for probing consciousness in individuals who cannot communicate, such as patients in minimally conscious state or unresponsive wakefulness[14].

In search for such signature, several studies have investigated the dynamics of brain activity at the boundary between non-conscious and conscious processing[15]. These studies have identified sharp neural changes separating, for a same physical stimulus, trials with conscious report from missed trials. A first example was observed in attentional blink experiments, in which the same visual word, embedded in a rapid stream of other stimuli, was sometimes seen and sometimes not[9,16–18]. Evoked potentials beyond 250 ms after the target word were triggered in an all-or-none fashion, directly associated with conscious reports[9]. Other experiments, using visual masking paradigms[10,19], have found that the magnitude of late activity shows a non-linear increase with stimulation strength, around the behavioral detection threshold. Such effects have also been detected in infants[19] and more recently in neural recordings from non-human primates[20,21]. Thus, all of these results point to a qualitative "bifurcation" in neural processes separating conscious versus non-conscious processing.

However, additional steps are required before accepting that bifurcation dynamics are a generic signature of the transition between non-conscious and conscious perception. First, it is unknown whether they generalize to other sensory modalities, or rather represent peculiarities of the visual system. As suggested in several recent reviews, generalizing from vision to e.g., audition is a key step in validating candidate neural signatures of conscious processing[22,23]. Second, the stimulus presentation procedures used in previous studies were arguably complex: they involved high-level perceptual objects, such as faces and words, and included sharp transients in the stimulation itself, two characteristics that might themselves introduce non-linearities in brain responses[24,25]. A neural signature of conscious perception should generalize to simpler cases. Third, and perhaps more importantly, the reporting itself may contaminate the neural response, with potential confounds due to decision-making or even motor preparation. It is, therefore, crucial to probe whether the current candidates for neural signatures of conscious access are associated with spontaneous conscious processing, in the absence of report[13]. Finally, we also need to address a critical methodological concern: the characterization of brain dynamics in previous studies has been mostly descriptive, and, as we explain below, might have confused bifurcation dynamics with unimodal non-linear dynamics, without any actual bifurcation separating conscious from non-conscious processing. We address this issue by formulating explicit models for different types of brain dynamics around the threshold for conscious perception, and by testing their predictions quantitatively.

Here we present a model-based approach to investigate the dynamics of brain activity in response to simple auditory stimuli of various intensity around consciousness threshold. We show that a very specific pattern of variability predicted by the bifurcation model is observed in several independent analyses of behavioral and neural data. Furthermore, fitting the bifurcation model to individual trial-by-trial data shows that it is a statistically better explanation of neural activity beyond 250–300 ms post-stimulus than the non-bifurcation models. It can be used to predict behavioral report from neural data when task-related reports are available, but also to predict the conscious contents of random mind-wandering probes without task-related reports. Overall, these results support the existence of qualitative changes in processing around the perceptual threshold, independent of decisional processes. These changes likely distinguish conscious from unconscious perception and can be decoded from individual brain activity recordings.

## Results

We recorded brain activity using electroencephalography (EEG) in healthy human adults who performed a simple auditory detection task. Vowels (French /a/ or /ə/) were embedded in continuous noise[26], at different signal-to-noise ratios (SNRs) around behavioral threshold. Twenty participants took part both in an active session, where they had to report the identity and the audibility of the vowels, and in a passive session, where they listened to the same sounds but did not have any task to perform on these stimuli; instead they performed a set of visual or amodal tasks unrelated to the auditory stimulation. The order of active and passive sessions was counterbalanced across participants, to test whether performing the auditory task had any impact on subsequent passive listening.

**Modeling framework and predictions.** Importantly, to guide all of our analyses, we developed a computational framework contrasting three plausible models of brain dynamics (Fig. 1). According to a first model, neural activity reflects stimulus strength in a linear fashion, with a gradual transition from non-conscious to conscious processing (Fig. 1A). A second model introduces a simple nuance to the first, not considered in previous studies: neural activity could show a gradual, but non-linear relationship with stimulation strength. Indeed, there is little reason to assume that the physical unit measuring signal strength (dB in our case) maps linearly to neural activity strength[27]. For this model, each stimulus evokes a unimodal distribution of activity, as for the linear model, but the means (or "modes") of these distributions increase non-linearly with stimulus strength (Fig. 1B). Note that in these first two models, there is no qualitative distinction between conscious and non-conscious processing. Our third model, in contrast, postulates a qualitative distinction between conscious and non-conscious processing, which leads to bifurcation dynamics. Across trials, the same stimulus either evokes a "high activity" mode associated with conscious processing, or remains in a "low activity" baseline mode that lumps together all stimuli that are not consciously processed (Fig. 1C). While the conscious mode can show an increase of brain activity with stimulus strength, the non-conscious mode cannot.

This modeling framework highlights the important point that mean activity across trials is a poor metric to distinguish between different types of dynamics (Fig. 1A–C). In particular, both the non-linear unimodal and bifurcation models can lead to a similar non-linear increase in mean activity around threshold. Instead, a unique characteristic of bifurcation dynamics can be found in the variability across trials: because a stimulus close to threshold

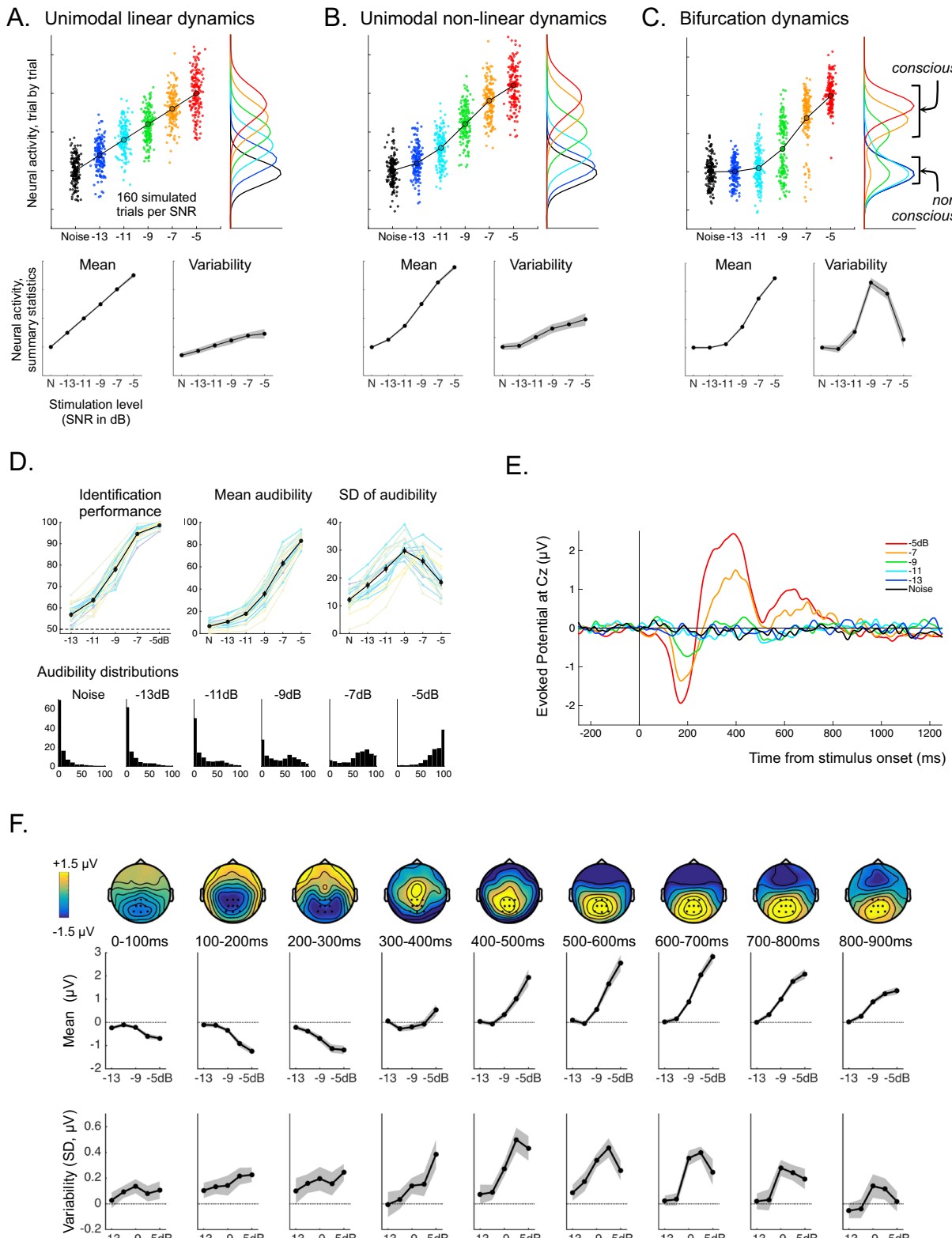

yields a mixture of high- and low-activity trials, this produces a burst in inter-trial variability. For stimulation well above or well below threshold, in contrast, activity is more homogenous across trials, leading to lower inter-trial variability. This produces a highly specific non-monotonous profile in the variability response function for the last model, when variability is plotted as a function of stimulation strength (Fig. 1C).

**Behavior in the active sessions**. We first applied this framework to the analysis of the behavioral results in the active sessions. In active sessions, participants had to report the identity of a vowel sound embedded in noise and rate its audibility. Mean identification performance followed a classic psychometric function, here with an inflection point between −9 dB SNR and −7 dB SNR. Mean audibility followed a similar pattern

**Fig. 1 Models, behavioral results, and evoked potentials in the active sessions. A–C** show the predictions of three possible models relating stimulus strength and neural activity. For each model, simulated trial-by-trial neural activity is represented as a function of the stimulation level (SNR). The corresponding distributions for each SNR are shown next to each graph with the same color-code. The bottom row shows the profiles of mean neural activity and variability of neural activity across trials (standard deviation) as predicted by each model. The profiles are averaged across 20 simulated participants who would have performed 160 trials per condition each (error bars correspond to standard error of the mean). **D** shows the behavioral results: identification performance, mean audibility and standard deviation of audibility across trials as a function of SNR, for each participant (faint colored lines) and averaged across all 20 participants (thick black line). Error bars represent ± standard error of the mean, $n = 20$ participants. The bottom row shows the distributions of audibility responses averaged across participants, for each SNR level. **E** Group averaged potentials evoked by the target vowels at electrode Cz and at each SNR level ($n = 20$ participants). **F** The top row shows the topographies of the event-related potentials at different time windows following targets at the highest SNR. The middle and bottom rows show the mean and variability of evoked activity as a function of SNR at these different time windows over a group of centro-parietal electrodes (highlighted on the topographies). Shaded areas correspond to ± SEM around the group average ($n = 20$ participants). Source data for this figure are provided as a Source data file.

(Fig. 1D). Importantly, audibility was assessed using a continuous scale so we could derive a meaningful variability measure for behavior. In a 2-alternative forced choice task, such as the one used for identification, variability is maximal at 50% performance by construction[28]. In contrast, the continuous measure of audibility imposed no such constraints, as variability could vary freely at all levels tested, provided they were sufficiently remote from floor or ceiling. Thus, any putative discontinuity in sensation may manifest itself by a burst in variability in the audibility judgments, following the modeling described in Fig. 1A–C (see also refs. [16,17]).

As predicted by bifurcation dynamics, the standard deviation of audibility showed a non-monotonic profile with increasing SNR, with a burst in variability around detection threshold. Moreover, the underlying distributions of audibility ratings at threshold SNRs were bimodal, with one mode around 0% audibility and one mode above 50% audibility (Fig. 1D), again as predicted. So, signs of a qualitative change between conscious and non-conscious processing, which had previously been observed in attentional blink and masking paradigms[9,10,16,17,29], generalize to a simple stimulation paradigm, with no sharp transients in the stimulation itself. Tellingly, this non-monotonic profile of variability was present in each and every participant (Supplementary Fig. 1).

**Event-related potentials during the active sessions**. An event related analysis of the EEG data showed that the vowels evoked classic auditory potentials in these active sessions (Fig. 1E, F). A first negative waveform was visible around 150 ms post-stimulus (N1), followed by a central positive waveform associated with posterior negativity (250–500 ms, P2), and then a sustained centro-posterior positivity (500–900 ms, P300). These evoked potentials were modulated in latency and amplitude by stimulus strength (Fig. 1E).

To go beyond the grand mean analysis, we examined the dynamics of the compound activity of 9 electrodes around Pz, which captured most of the N1 and P300 (Fig. 1F). Mean activity across trials generally showed a non-linear increase in intensity with stimulus strength, starting from the earliest latencies corresponding to the earliest stages of processing. As explained above, this non-linear increase is compatible with either the non-linear unimodal or the bifurcation dynamics models. In contrast, the relationship between variability across trials and stimulus strength changed drastically from early to late processing. Before 300 ms, variability remained about constant or showed a monotonous increase with stimulation strength. Then, for later latencies, variability became non-monotonic, exhibiting a clear peak around threshold (−9 to −7 dB SNR). These first results suggest a switch from unimodal non-linear dynamics to bifurcation dynamics occurring around 300 ms after stimulation onset.

**Multivariate pattern analysis during the active sessions**. To obtain a measure of stimulus-related neural activity that was both exhaustive, taking into account the pattern of activity across all 64 electrodes, and also based on each individual's activity patterns instead of the grand average, we performed a series of multivariate pattern analyses (MVPA). For each individual and at each time point, an automated classifier (l2-regularized logistic regression) was trained across all 64 electrodes to discriminate target absent from target present trials at the highest stimulation SNR. We then used these classifiers to predict target presence at the same SNR, via cross-validation, as well as other, less favorable SNRs, via direct generalization. This procedure is illustrated in Fig. 2A. Training and testing phases were performed at the same time points, but also at different time points in a so-called temporal generalization procedure[30] in order to identify the classification features that were sustained over time.

The MVPA transforms the multivariate EEG activity at each trial, each time point and for each participant into a representation best expressing the neural response to stimulus presence versus stimulus absence (each dot in the illustrative Fig. 2A). Just as an evoked potential, the MVPA projection can be viewed as a measure of brain activity over a region of interest, but with the region of interest derived from the data to optimally distinguish the presence of a stimulus. This projected activity can be used to try and predict whether the stimulus was indeed present or absent on that trial: the distance from the decision criterion—i.e., the criterion that best separates stimulus present versus absent trials (the black line in Fig. 2A)—indicates the strength of the prediction toward stimulus presence (positive distance) or absence (negative distance). In the following, we term this distance the projected neural activity and use this continuous measure to assess variability across trials from the MVPA output.

First, to quantify classification performance, we computed the area under the receiver operating curve (AUC), which measures the separation between the distribution of projected neural activity for target present and for target absent trials: 1 indicates perfect separation, 0.5 indicates undistinguishable distributions (illustrated in Fig. 2A). For stimuli with the highest SNR, significant classification started at around 100 ms (Fig. 2B, black contours indicate periods where classification was significantly above chance with $p$-corrected $<0.05$). There was an initial period during which temporal generalization was limited to a short time window of about 50 ms, suggesting a sequence of short, non-overlapping processing stages. Beyond 250 ms, however, generalization was observed over longer time windows, suggesting a period of sustained processes with a temporal extent of about 200 ms. Below chance off-diagonal performance indicated that some early patterns of activity recurred later with reverse polarity, as has already been observed with the temporal generalization technique[30]. Finally, temporal generalization indicated that some patterns of activity observed between 250 and 500 ms, were

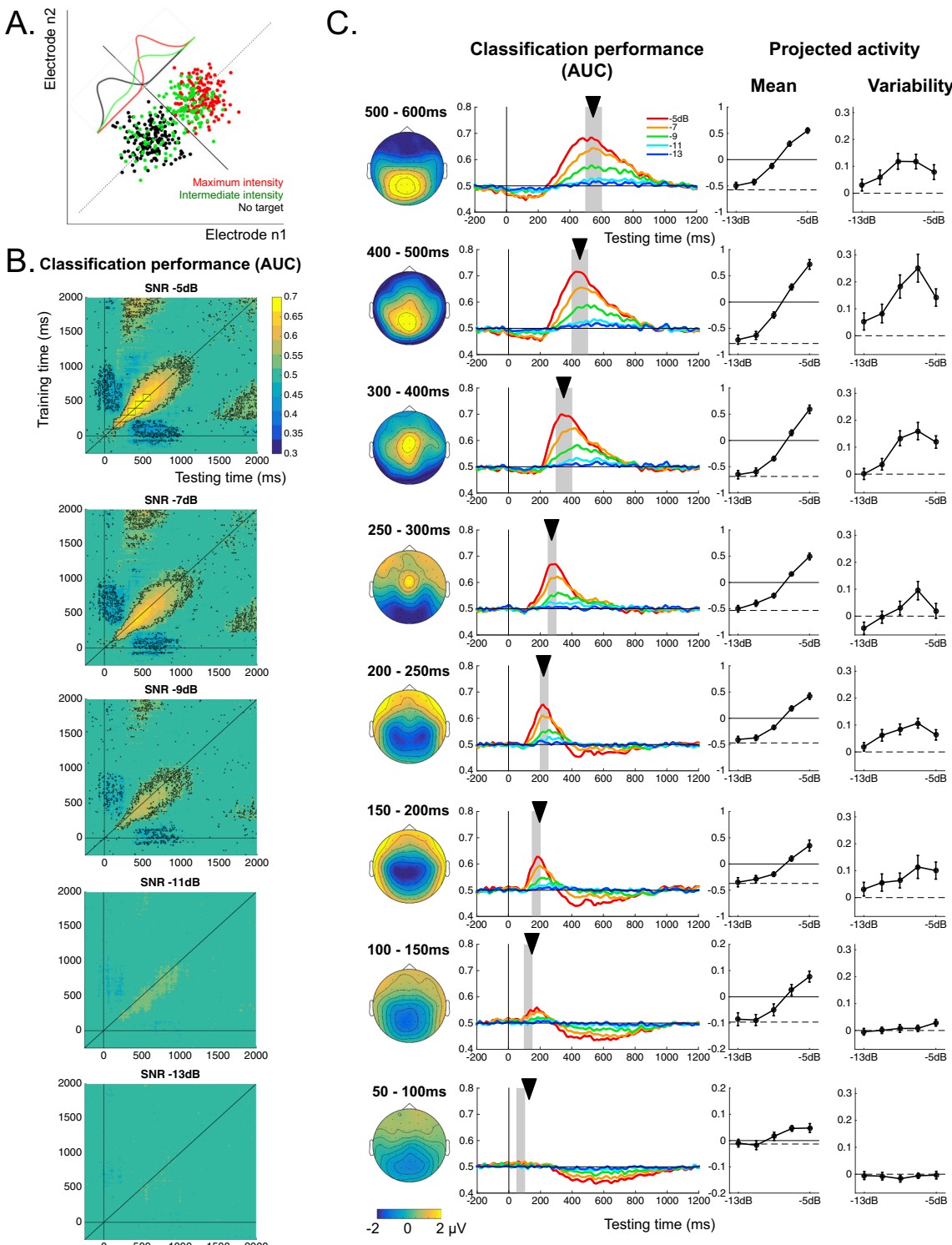

reactivated around 1750–2000 ms, probably in anticipation of the response, since the response screen appeared between 2000 ms and 3000 ms. All this is consistent with results of previous studies using the same approach[31–34]. Also, as expected, classification performance declined with declining SNR, but it still maintained a qualitatively similar pattern of generalization over time (Fig. 2B).

Second, in order to further investigate the underlying brain dynamics, we analyzed the mean and variability of the trial-by-trial

projected activity derived from MVPA. The mean projected activity displayed a non-linear increase with SNR for all time windows beyond 150 ms, compatible with either non-linear unimodal or bifurcation dynamics (Fig. 2C; non-linearity was assessed by verifying that the derivative of the curve changed significantly as a function of SNR using one-way repeated measures ANOVAs at each time window: after 150 ms all $F(4,76) > 4.9$; all $p$-FDR-corrected <0.005). The trial-to-trial variability, in contrast, displayed either a flat or slightly increasing profile for latencies up to 200 ms

**Fig. 2 Dynamics of the different processing stages during the active session (analyzed using multivariate pattern classification). A** Illustration of the multivariate pattern classification analysis in 2D (two electrodes). Each point represents activity at electrodes n1 and n2 for a single trial, in three conditions: target absent (black), target present with maximal SNR (red), and target present around threshold SNR (green). The decision boundary (black line) is computed to maximize accuracy in classifying target present with maximal SNR versus target absent in a training subset of the data. The so-called projected activity for a specific test trial is defined as the distance between that trial's point in the multivariate space and the decision boundary. The distributions of these projected activity measures for the different conditions are plotted along the decision axis (dotted line). The classifier performance was assessed using the AROC of the distribution for target present against the target absent distribution (see "Methods"). **B** Time-generalization analysis. The classification accuracy is shown for each combination of training time point and testing time point. Black contours indicate periods where classification deviated significantly from chance at p corrected <0.05 (two-sided non-parametric sign test[76], Benjamini–Hochberg FDR correction). **C** Dynamics of neural activity at selected time windows (represented by squares on the top panel in (**B**)). The left column represents the group average topographies at each time window for the strongest SNR (−5 dB), the next column shows the group average time courses for the different SNRs, i.e., the sensitivity of the classifier trained at a given time window (indicated by the shaded area and black arrow) and tested across the whole duration of a trial. The third column shows mean projected activity as a function of SNR for each time window of interest (arbitrary unit). The last column shows the variability of projected activity across trials as a function of SNR, i.e., the standard deviation of the projected activity across trials (arbitrary unit). Error bars correspond to ± SEM around the group average (n = 20 participants). Source data for this figure are provided as a Source data file.

post-stimulus; then, from 200 to 250 ms onwards, these profiles changed into a non-monotonic profile with a peak in variability around threshold SNR, and correlated significantly with the behavioral variability profiles measured for each individual on the audibility scale (Student t-tests on correlation coefficients between individual neural and audibility profiles of variability, see "Methods": in the first three time windows, before 200 ms, $t(19) = 0.12, -0.90, 0.71$, respectively, all p-FDR-corrected > 0.5, effect sizes Cohen's $d < 0.2$; on the fourth time window 200–250 ms $t(19) = 3.06$ p-FDR-corrected = 0.010, Cohen's $d = 0.68$; for the following time windows beyond 250 ms, all $t(19) > 4.10$, all p-FDR-corrected < 0.01, all Cohen's $d > 0.9$). This variability peak is consistent with bifurcation dynamics for processing stages beyond 250 ms post-stimulus onset.

**Link between bifurcation dynamics and behavioral report during the active sessions.** The data are so far consistent with qualitative predictions of the bifurcation model for latencies beyond 250–300 ms post-stimulus. Does the link hold for finer details of the observed neural activity? We assessed the distributions of trial-to-trial projected activity at various SNRs and compared them with the predictions from our models. The bifurcation model predicts markedly different distributions compared to the unimodal linear or non-linear models, since it is the only model that predicts bimodal distributions in response to stimulations around the perceptual threshold (Fig. 3A, left and middle panels). The distribution of neural activity as assessed with the MVPA was strikingly consistent with the predictions of the bifurcation model, at the level of the individual (Fig. 3A right) or at the level of the group (Supplementary Fig. 2).

Since the neural activity analysis showed bimodal distributions, we could then ask the central question of whether these two modes corresponded to conscious versus non-conscious perception. We split trials according to participants' audibility reports, setting a criterion of 30% audibility for "heard" versus "not heard" trials. This criterion was derived from the trough of the bimodal audibility distributions observed for SNRs of −9 dB and −7 dB (see Fig. 1D), corresponding to threshold in identification performance. Figure 3B shows the evoked potentials for "heard" and "not heard" trials separately for the SNR of −7 dB. The two models diverge on their predictions for the "not heard" trials. For the unimodal model, mean activity increases with stimulus strength for any trial, heard or not, consistent with a signal detection theory framework[28]. By contrast, a bifurcation model predicts that, for "not heard" trials, mean activity should stay the same whatever the stimulus strength, and this activity should be equal to that observed when no stimulus is presented. For "heard" trials, all models predict that increasing stimulus strength should

increase neural activity. In other words, the bifurcation model predicts an interaction between consciousness report (heard/not heard) and SNR, whereas the unimodal model predicts no interaction. Results are shown in Fig. 3C for the time window between 300 and 400 ms, and Supplementary Fig. 3 for all latencies. Neural activity closely followed the prediction of the bifurcation model. A linear mixed effect model with conditions SNR × Report (heard/not heard) showed significant effects of report (likelihood ratio test, $F_{1,24.3} = 30.3$, $p = 1.12 \times 10^{-5}$), SNR ($F_{1,20.5} = 54.7$, $p = 3.27 \times 10^{-7}$) and their interaction ($F_{1,19.4} = 7.86$, $p = 0.011$); see "Methods" for details.

All our analyses so far focused on time windows of interest. We next analyzed the neuro-behavioral correlation continuously along the timeline (see "Methods"). We correlated the neural mean and variability profiles (as in Fig. 2C) and the corresponding behavioral mean and variability profiles (as in Fig. 1D) at each time point and for each subject (Fig. 3D). For mean profiles, a neural/behavioral correlation was observed over a long period of time, from stimulus onset up to beyond 1.5 s. This correlation captured the whole period during which the brain responded to the stimulus. For variability profiles, the neural/behavioral correlation started around 250 ms and only lasted until 700–800 ms post-stimulus. This shorter period signed the moment when the brain response displayed bifurcation dynamics

**Model fitting and Bayesian model comparison during the active sessions.** All the analyses up to here suggest that the bifurcation model is better able to capture the important features of the data beyond 250 ms post-stimulus. We formally tested for this claim by quantitatively fitting the computational models to the trial-to-trial neural data, using a Bayesian model comparison approach. We compared three models: the bifurcation model, the non-linear unimodal model, and a baseline "Null" model in which the stimulation strength has no effect on neural activity. The linear unimodal model (Fig. 1A) was not included in the comparison because it is a special case of the non-linear unimodal model. We used contiguous time windows of interest of 30 ms each, so that the trial-to-trial signal was more robust to small temporal fluctuations. For each participant and at each time point, the parameters of the models were estimated on a training set, and the likelihood of each model with these parameters was then calculated on an independent testing set. This cross-validation procedure allowed correcting for the different number of parameters across models (see "Methods"). Then, we performed a random-effect Bayesian model comparison at the group level[35,36], in order to estimate the probability for each model to be more frequent in the population of participants than the other two, i.e., protected exceedance probability. Results are

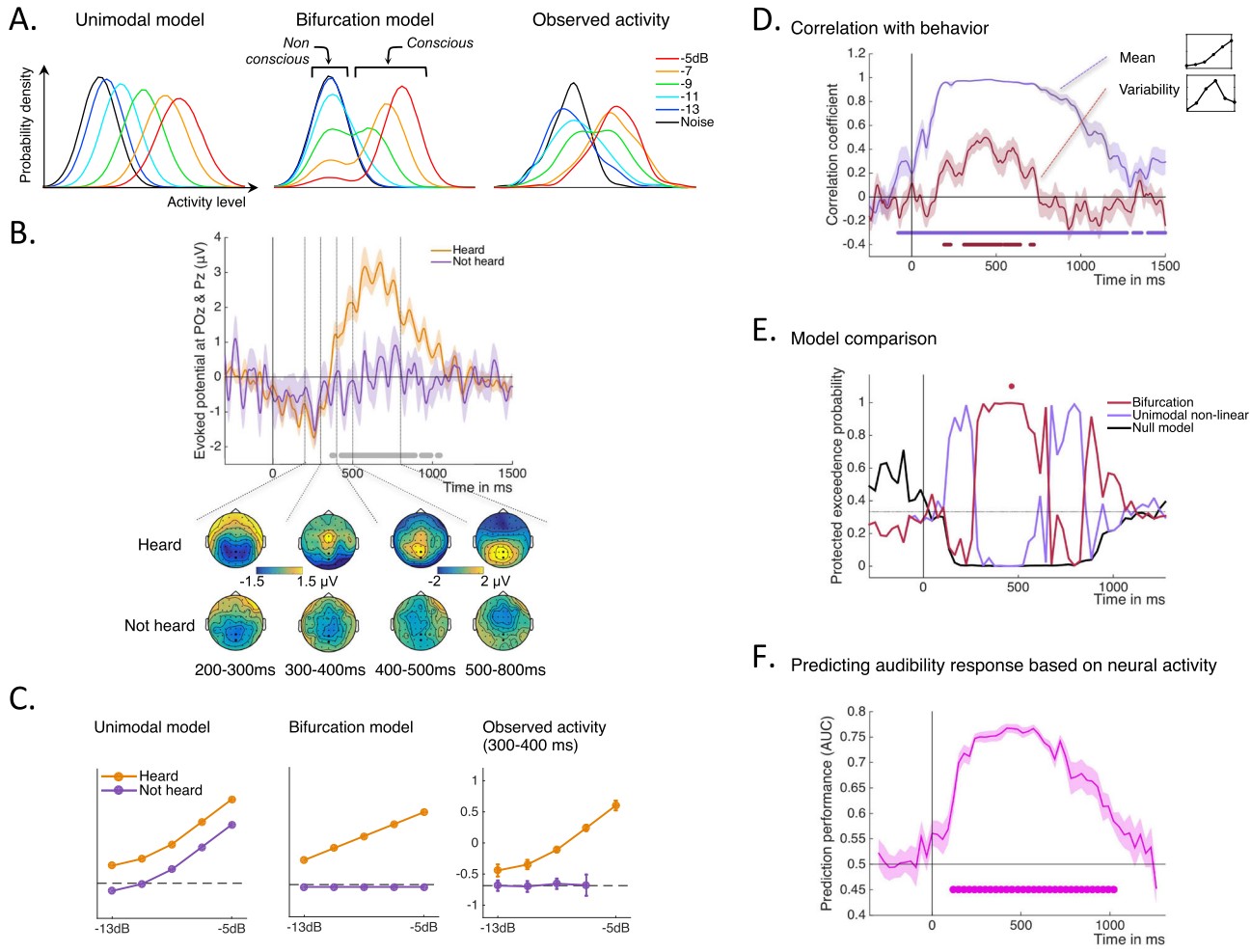

**Fig. 3 Correlation with behavior, Bayesian model comparison, and prediction of conscious reports in the active session. A** Predicted distributions of activity across trials as a function of SNR level for unimodal non-linear and bifurcation models (left and middle) compared with distributions of multivariate neural activity observed within the 400–500 ms time window for participant 14. **B** Grand average evoked potentials at electrodes POz and Pz for stimuli at threshold (−7 dB), shown separately for "heard" and "not heard" trials. Shaded area shows SEM ($n = 20$ participants), gray dots represent time points where the difference is significant at $p$-corrected <0.05 (two-sided paired $t$-tests, FDR correction). Below are shown the corresponding topographies at various time windows. **C** Mean activity as a function of stimulus strength, shown separately for "heard" versus "not-heard" trials. Model predictions for the unimodal model (left) and bifurcation model (middle) are compared to the mean projected activity that was actually observed for "heard" and "not heard" trials during the 300–400 ms time window (grand average across participants, arbitrary unit, error bars represent SEM, $n = 20$). **D** Correlation coefficient between the individual mean profile (blue) or variability profile (red) of projected neural activity with the corresponding mean or variability profile in subjective audibility. The shaded areas correspond to SEM ($n = 20$ participants). The periods of significant correlation are denoted as thick lines of the corresponding color (one-sided $t$-tests, $p < 0.05$, FDR corrected). **E** Bayesian model comparison: protected exceedance probability at the group level for the unimodal non-linear, bifurcation, and null models. **F** Prediction of behavioral audibility from the neural projected activity under the assumption of bifurcation dynamics. Group average performance in predicting trial-by-trial audibility responses (heard versus not-heard) is shown for intermediate SNRs (−11, −9, and −7 dB). Thick dots denote periods were prediction performance was better than chance (two-sided $t$-tests, $p < 0.05$, FDR corrected). The shaded areas correspond to SEM ($n = 20$ participants). Source data for this figure are provided as a Source data file.

shown in Fig. 3E. During baseline, the protected exceedance probability of the Null model was slightly above the other two as expected. Then, there was a first period between 0 and 250 ms after stimulus onset where the unimodal model showed the highest probability. After 250 ms, there was a switch in favor of the bifurcation model, whose protected exceedance probability remained above 95% for most of the period between 250 and 700 ms. Then there was a last period between 700 and 900 ms where the unimodal model seemed most favored again. The Bayesian model comparison thus formally confirms that, within the broad period of about 1.5 s where the brain responds to the stimulus, there is a period between 250 and 700 ms post-stimulus onset where neural activity shows bifurcation dynamics.

**Predicting behavioral reports based on neural activity in the active sessions.** As a final and critical test, we probed the predictive power of the bifurcation model: if the bifurcation in neural activity is a signature of the split between non-conscious and conscious processing, we should be able to predict future consciousness report on each trial based on whether neural activity at the time of bifurcation belongs to the "high state" or the "low state" of the bifurcation. As described above, we fitted the bifurcation model to individual participants' projected activity across all trials. From this fit we could derive the predicted distributions (mean and standard deviation) for the "low-state" and the "high-state" at each level of stimulation. For each trial we could therefore compute the likelihood that activity recorded on that trial belonged

to the high versus the low state (Bayes Factor, see "Methods"), and take this measure as a prediction of whether the stimulus would be reported as heard or not. As shown in Fig. 3F, this approach predicted consciousness reports with a high accuracy, especially between 250 ms and 700 ms post-stimulus (AUC above 0.75). It should be emphasized that these trial-by-trial predictions were obtained without using any information about the participants' behavioral responses at any stage of the procedure, including during the fitting of the model. The method is instead principled on the hypothesized structure of neural data, as captured by a bifurcation model. As far as we know, this contrasts with all previous approaches to decode conscious reports in which supervised classifiers had to be trained using neural data labeled with the participants' responses. Such a modeling approach can thus be generalized to cases where there are no overt responses, or a limited number of them, as shown now.

**Comparing brain activity in the active and passive sessions.** One essential issue remains to interpret bifurcation dynamics as a general signature of consciousness—one that is common to all investigations using neural/behavioral correlations: do neural processing of reported stimuli reflect conscious access per se, or rather decision-making processes involved in reporting?[13] As reports are themselves discontinuous choices, they could induce bifurcation dynamics unrelated to perceptual consciousness. And indeed, most current models of signal detection and decision making in humans assume that dichotomous dynamics only arise at the decision stage[28,37]. To address this issue, we recorded brain activity during "passive" sessions, in which participants received the same auditory stimulation as in the active sessions but were not required to perform any task on the sounds. Instead, various tasks unrelated to the sounds were randomly introduced in between trials: a speeded visual reaction task, an amodal task requiring mental arithmetic or general culture answers, a mind-wandering probe, or a non-speeded visual reaction task. To control for potential carry-over effects in listening strategy between active and passive sessions, the order of sessions was counterbalanced across participants.

The main hypothesis was that neural responses due to decision-making should disappear in the absence of a stimulus-related task, whereas neural responses reflecting core conscious access should persist. Thus, if late sustained activations only reflect decision-making, they should disappear during passive listening. In contrast, if they reflect a signature of conscious access, they should remain in the passive sessions.

Surprisingly, neither prediction was entirely fulfilled: as illustrated in Fig. 4A, the P3-like positivity that characterized late activations during active listening disappeared under passive listening. This corroborates some previous observations[38–40]. But, interestingly, this did not mean that late activity disappeared altogether: indeed, passively listening to the sounds still evoked late sustained activations well beyond the initial auditory components, up until 700 ms post-stimulus. Importantly, these activations had a different topography from what was observed in the active session, distinguishing them unequivocally from the P300. The evoked potentials at Cz further illustrate this point: the positive deflection observed from 250 ms onwards in the active sessions, typical of a P300 component, is replaced by a negative deflection in the passive sessions.

Which processes could be responsible for the late waveforms still observed in the passive sessions? We hypothesize that these waveforms actually reflect spontaneous covert conscious access, independent of a task. If so, similar processes must also be present during the active sessions, as conscious access is also experienced, but they should be embedded in other processes related to

decision-making for the report. We tested this hypothesis by performing a cross-classification analysis, where a decoder trained on one type of session was tested on the other type of session, for each subject individually (see "Methods"). Results showed that the neural activity in the active sessions could be decoded by training a classifier on the passive sessions, at least within a 250–600 ms time window (Fig. 4B top row, middle column). Still, such cross-decoding left sizeable residual late activity compared to training within the active sessions (Fig. 4B top row). In contrast, for the complementary analysis of decoding activity in passive sessions after training on active sessions, the late activity was fully captured, almost as well as when both training and testing was performed on the passive sessions (Fig. 4B bottom row). This pattern of result is consistent with the hypothesis that late activity in the passive sessions reflects core conscious access mechanisms that are supplemented with decision-making mechanisms in the active sessions when a task is required.

**Comparing reconstructed sources in the active and passive sessions.** Source reconstruction of the EEG activity, shown in Fig. 4C, gave further support to this hypothesis. For both types of session and beyond 250 ms, activations were observed in a broad network of areas encompassing auditory cortex and other temporal, parietal or frontal areas. The main difference between the active and the passive sessions was the presence of strong activations in the premotor, motor cortex, and supplementary motor area for the active sessions. This difference in cortical activity presumably accounts for the markedly different late topographies observed in Fig. 4A. We then computed the time course of activations at reconstructed sources in regions around auditory cortex, inferior-prefrontal cortex, and supplementary motor area (Fig. 4D). Both temporal and inferior-prefrontal regions responded with sustained activity beyond 250 ms, with or without a task. In contrast, the activation of the supplementary motor area was all-or-none, only observed when a task was required in the active sessions.

**Brain dynamics in the passive sessions.** We next performed the same series of dynamical analyses in the passive sessions in order to assess whether bifurcation dynamics also characterized stimulus processing in the absence of behavioral report. We first assessed mean and variability profiles of neural activity in the passive sessions, both for evoked activity in a region of interest over the temporal electrodes (Fig. 5A) and for projected activity derived from MVPA (Supplementary Fig. 4). Beyond 300 ms, we confirmed the presence of the specific pattern associated with bifurcation dynamics, with a burst of trial-to-trial variability for intermediate SNRs. Compared to the active session, this burst was shifted toward higher SNRs ($-7$ dB to $-5$ dB). This is an expected consequence of a lack of directed attention to the sounds in the passive sessions.

Then, we set to compare the burst of variability in neural data to behavior on a participant per participant basis. However, because there were no reports of the sounds in the passive sessions, we correlated the neural activity recorded during the passive session with the behavioral profiles collected in the active sessions, taking the SNR shift into account (Fig. 5B, see "Methods"). For mean profiles, we observed significant correlations between projected activity in the passive session and audibility in the active session, for a long period of about 1 s after stimulus onset. Furthermore, a correlation was also observed for variability profiles, mostly between 200 and 600 ms, signing a period of bifurcation dynamics.

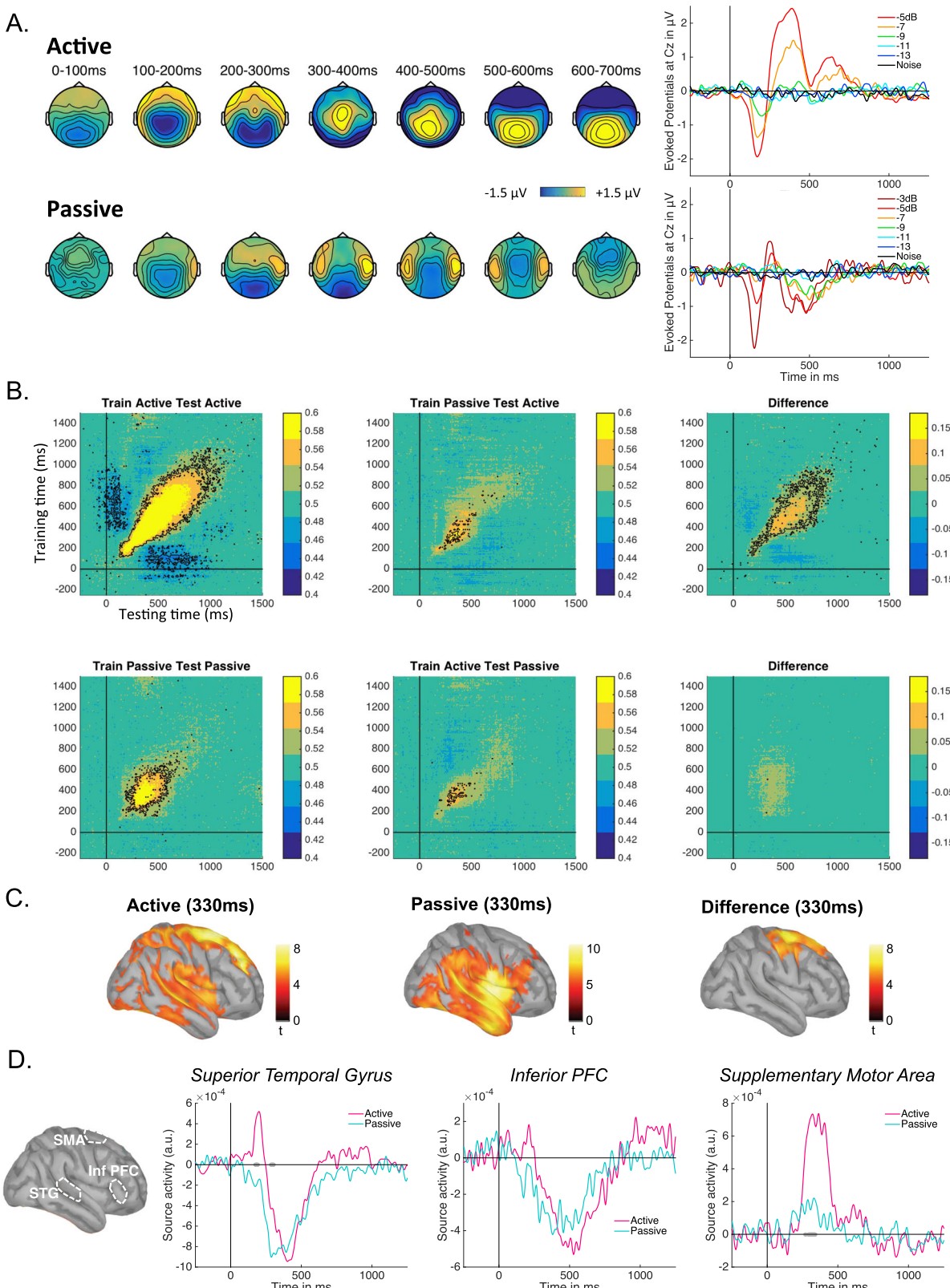

**Predicting mind-wandering content from neural activity in the passive sessions**. Finally, although no auditory task was required from participants, we asked whether we could predict their spontaneous conscious access to sounds during passive listening, based on neural activity. For that, we analyzed the "mind wandering" probes included at random times during the passive sessions. These probes asked the participants to report what they had on their mind at that moment, using one of four options: the sound, the visual or amodal tasks, their own thoughts, or nothing/they were falling asleep (see "Methods"). For the majority of mind-wandering

**Fig. 4 Comparison of neural activity in the active and passive sessions. A** Group averaged topographies at various time windows following stimulus presentation in the active and passive sessions at −5 dB. Group averaged evoked potentials at Cz for the different SNRs in the active and the passive sessions. **B** Classification scores (AUC) for the cross-classification analysis, at an SNR of −5 dB. Regions where classification was significantly different from chance are outlined in black (two-sided non-parametric sign test[76], $p < 0.05$, FDR-corrected). **C** Source reconstruction at 330 ms post-stimulus. The activation maps show the $t$-values of a paired $t$-test across subjects between mean activations for stimulus present at −5 dB versus stimulus absent, in the active or passive sessions (left and middle) and between mean activations for the active versus passive session for stimulus present at −5 dB (right). Only significant activations are shown ($p < 0.05$, FDR corrected). **D** Group averaged time course of source reconstruction in three regions of interest, shown for the active or passive sessions (SNR = −5 dB as above). Thick gray dots indicate periods of significant difference (two-sided paired $t$-tests, $p < 0.05$, FDR corrected). Source data for this figure are provided as a Source data file.

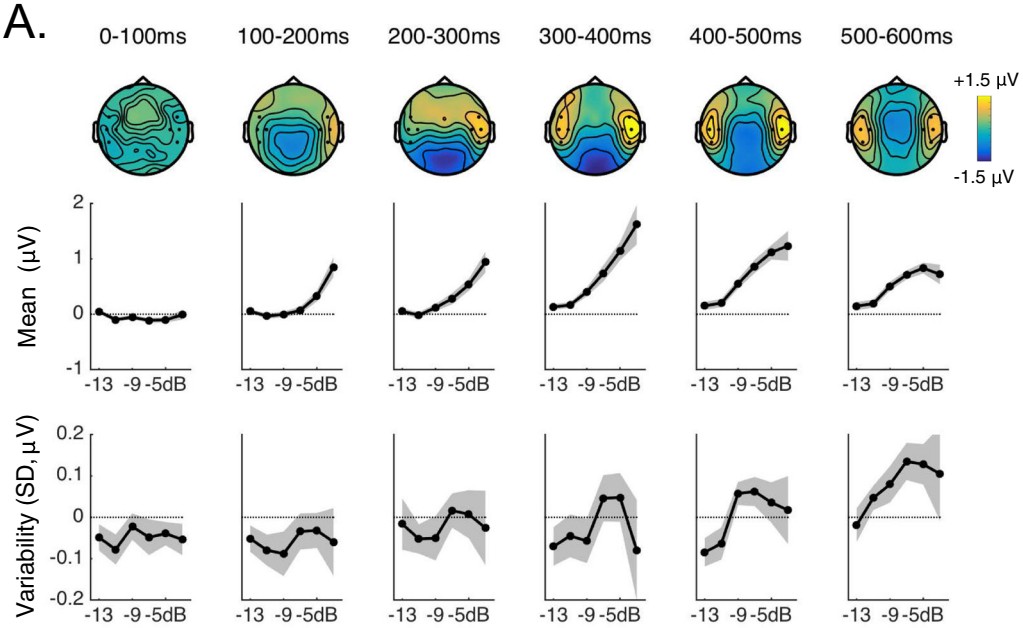

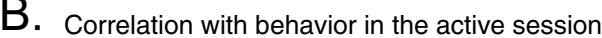

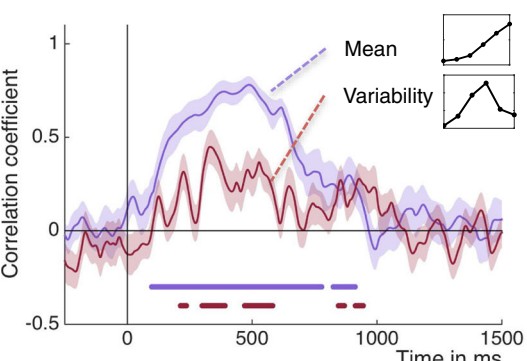

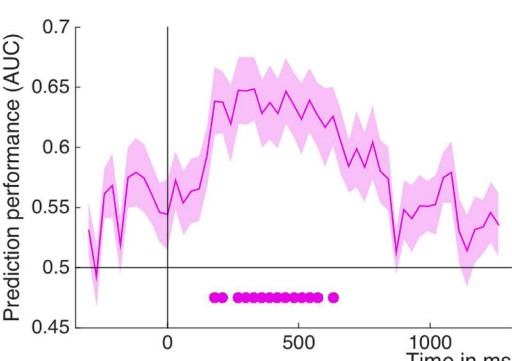

**Fig. 5 Neural dynamics for passive sessions. A** The top row shows the group averaged topographies at different time windows following targets played at −5 dB SNR. The middle and bottom rows show the mean and variability of evoked potentials as a function of SNR, for different time windows and over a group of temporal electrodes highlighted on the topographies. Shaded areas are SEMs ($n = 20$ participants, from which 10 at −3 dB). **B** Correlation coefficient between the individual mean profiles (blue) or variability profiles (red) of projected activity in the passive session with the individual mean profiles or variability profiles of behavioral audibility during the active session (shifted by 1 or 2 SNR levels). The shaded areas are SEMs ($n = 20$ participants). Periods of significant correlation are denoted in thick lines of the corresponding color (one-sided $t$-tests, $p < 0.05$, FDR corrected).
**C** Prediction of mind-wandering content from the neural projected activity under the assumption of bifurcation dynamics. The graph shows the group average performance in predicting, trial by trial, whether the participant responded that the sound was on their mind at the moment of the probe or not. All SNRs are included in this analysis. The shaded areas are SEMs ($n = 20$ participants). Thick dots denote periods were prediction performance was better than chance (two-sided $t$-tests, $p < 0.05$, FDR corrected). Source data for this figure are provided as a Source data file.

probes, participants reported either focusing on the visual and amodal tasks or on their own thoughts; they very rarely reported an absence of content or sleepiness (Supplementary Fig. 5). Overall, the sound constituted 19% of the reported mind-wandering contents, and this proportion increased with the intensity of the sound that preceded a mind-wandering probe (Supplementary Fig. 5). We thus assessed if we could predict whether or not a participant's mind-wandering was directed toward the sounds or something else on a particular trial, based on the preceding neural activity. We used the same method as for the active sessions: we fitted the bifurcation model to the neural data, independently of any behavioral response, and used this fit to compute the probability that neural activity evoked by passive listening to a sound on each trial belonged to the "high" versus "low" state of the model. As shown in Fig. 5C, activity recorded between 250 ms and 700 ms after the sound predicted accurately whether the random mind-wandering probes included consciousness of sound or not, given the same external audio stimulation. In other words, with the bifurcation model of conscious access, we could use neural activity to predict whether participants were spontaneously aware of sounds in a passive listening condition.

**Controlling for covert task-related strategies in the passive sessions**. In order to further insure that this late activity genuinely reflected spontaneous conscious access and not some form of covert decision-making, we tested the effect of the order of the active and passive sessions. The activation patterns observed in the passive sessions were very similar in the groups of subjects who performed the passive session either first or second: there were no significant differences in topographies, time course, or the ability to predict mind-wandering responses based on the bifurcation model (Supplementary Fig. 6A–C). This suggests that there were no carry-over effect of the task performed in the active session on activity observed in the passive session. We also tested seven additional naïve subjects who only performed the passive session (see "Methods"). In these subjects, we observed the same activation patterns, with late sustained activity starting around 250–300 ms, similarly predicting mind-wandering responses (Supplementary Fig. 6D–F).

**Generalization to non-speech auditory stimuli**. Our auditory paradigm was intended to be simple and ecologically relevant. However, it used vowels as sounds to be identified. Vowels perception is often thought to be categorical, although this is debated[41]. In any case, some language specific processes related or not to categorical perception cannot be ruled out as a cause for the late bifurcation dynamics we observed. We, therefore, performed an additional control experiment, in which the vowels were replaced by pure tones (high pitch versus low pitch, with a large pitch difference so that the task was easy, see "Methods"). All other details of the procedure remained the same. Five additional participants were recruited and performed both an active and a passive session with the pure tone stimuli. These simpler stimuli evoked very similar late activity, with bifurcation dynamics, as the ones observed for vowels, both in the active and in the passive session (Supplementary Fig. 7). This brings further evidence that these late activity and dynamics might relate to general conscious access mechanisms that are common across stimulus types and task requirements.

## Discussion
In the present study, we characterized the dynamics of the brain's response to auditory stimuli of various intensities around perceptual threshold. Guided by a conceptual model, and unlike previous studies[10,19], we focused on inter-trial variability in order to reveal unique dynamical characteristics that are lost

when averaging activity across trials. Our main finding is that between 250–300 ms and 600–700 ms post-stimulus onset, the brain responses showed a distinctive burst of inter-trial variability at stimulation strengths around threshold. This was a signature of bifurcation dynamics: for the same physical stimulation, some trials led to high activity and others to low activity. This observation was repeated with and without behavioral reports, using simple event-related analysis but also more complex decoding techniques, and formally validated using a Bayesian approach. The bifurcation dynamics were closely related to perceptual reports when available: trials with high versus low activity led respectively to reporting conscious perception or the absence of conscious perception for the same external stimulus. Using the bifurcation model, we could thus effectively predict behavioral responses from brain activity, on a trial per trial basis. As the technique used was principled on the bifurcation model and did not require access to behavioral responses, we could extend the prediction to passive listening cases, for which we predicted the content of mind-wandering probes.

Based on those results, we propose that bifurcation dynamics are a general signature of the transition between conscious and non-conscious processing, which in turn allows identifying the neural signature of conscious access proper: a global brain activity arising between 250 and 700 ms post-stimulus, supplemented by additional decisional processes when a task is required.

In the following discussion, we show that these results validate and generalize dynamical predictions of the global neuronal workspace theory, while introducing an important nuance between non-linear and bifurcation dynamics. We also show that other predictions of the global neuronal workspace model, relating to the role of the P300 waveform, are invalidated by our results, and we propose an update of the model that accounts for these findings. More generally, we show that the present results can help advance several prominent debates about the neural basis of consciousness. Finally, we discuss the perspectives of this approach, notably in terms of diagnosing consciousness in non-communicating patients.

**Variability analysis supports dynamical predictions of the global neuronal workspace model**. The hypothesis that conscious access is associated with qualitative changes in the configuration of activity throughout the brain is a key prediction of the global neuronal workspace theory[17,42,43]. In this theory, conscious access arises through the broadcasting of local sensory information to a global network of areas. It is the reciprocal coupling within the global network that naturally yields bifurcation dynamics, as "ignition" either succeeds, leading to conscious access, or fails, leading to non-conscious processing[9,17,20]. The present results validate these dynamical predictions.

Furthermore, we extend the available evidence favoring this theory in a number of ways. An increasing number of studies now emphasize the importance of comparing neural correlates of conscious access across different sensory modalities in order to establish the generality of candidate neural signatures of consciousness[22,23,44–46]. However, so far bifurcation dynamics have only been formally tested for vision, and mostly using visual stimuli that contained themselves sharp transients[9,10,16,19,20], which also raises the concern that sharp transients may be responsible for some of the non-linear effects observed in brain responses. Here we show that these peculiar dynamics generalize to conscious perception in audition, in a very simple stimulation protocol (no masking, no manipulation of attention) where the stimulus itself is devoid of any sharp transients.

Further to this generalization, our modeling approach adds an important precision to the predictions of the global workspace

model by highlighting the distinction between bifurcation and non-linear dynamics.

**Distinguishing bifurcation and non-linear dynamics can help clarify the timing of conscious access per se.** Importantly, developing formal dynamical models was instrumental for distinguishing different periods in the neural correlates of conscious perception that were previously confused. Our modeling shows that, contrary to what has been suggested in previous works[10,19], non-linear dynamics in averaged activity cannot be used as a definitive signature of bifurcation dynamics, and hence of conscious access, since it can also be observed in the absence of bifurcation. This important distinction can help clarify seemingly contradictory findings on the timing of conscious access[11,12]. Non-linear increases in activity have been hypothesized to signal conscious access as early as 100 ms[47], but other reports claimed that such transition occurred only after 200 ms[9,10,19]. Here, we observed that early sensory processing, before 200 ms, followed a non-linear increase in magnitude, probably related to local recurrent loops within sensory areas[48], but that the underlying dynamics were unimodal. This indicates that these early activations are probably "preconscious"[42,49]: they precede the moment when global brain activity actually splits between conscious and unconscious processing, yielding bifurcation dynamics.

**Do bifurcation dynamics relate to conscious access or decision-making?** Another long-lasting controversy, and perhaps the most acute criticism of the global workspace hypothesis, is that bifurcation dynamics might relate to the task of reporting the stimulus and not conscious access proper[13]. This is a major issue, for the foundations of the theory, but also as covert conscious access without explicit reports surely represents the vast majority of our everyday conscious experience. Our results suggest a unifying view of overt and covert conscious access. Late neural bifurcation dynamics were still observed in the absence of any explicit report. However, source localization combined with cross-classification analyses showed that the network supporting late activity in the covert case was a sub-part of the one supporting late activity in the overt case.

More generally, the present results can also interrogate current models of signal detection and decision-making in humans. Standard and successful models of behavioral performance such as signal detection theory or evidence accumulation assume that the dichotomy between reported versus not reported stimuli arise at the decisional stage, while the perceptual stages exhibit no qualitative change in operation around threshold[28,37]. Our study reveals that bifurcation dynamics in neural activity occur even in passive listening conditions, without any request to take decisions or provide overt responses.

**The P300 component, the global workspace, and the global playground.** Our study also provides important elements for the current controversy about the P300 EEG component, which is debated as a potential signature of conscious access[13,38–40]. The present results show that conscious access in the absence of decision-making does not produce a P300 waveform, contrary to what has been proposed so far by the global neuronal workspace model[43]. This observation is consistent with several other recent experimental results[38–40]. But the present study also goes beyond these previous works by clarifying the relationship between the P300 and the signature of conscious access per se. Indeed, using a cross-classification analysis (Fig. 4B) we could demonstrate that the late sustained waveform that signs conscious access in the absence of a task (as characterized by bifurcation dynamics) is included in the P3-like waveform observed in the active

condition. In other words, the P3-like waveform observed when making this very general contrast of stimulus presence versus absence during active sessions, is actually a composite waveform that includes two overlapping components: the signature of conscious access per se, with its bilateral positivity, and an additional central positivity that corresponds to the P300 in a strict sense, which specifically reflects decision processes[50].

These results thus bring clarifications to previous debates. They show that, when no decision is required, the processes associated with conscious access per se can unfold in the absence of a P300. However, these "core" conscious access mechanisms still correspond to late and sustained activity, within the same time range as the P300. Finally, when a task is required, the additional P300 mechanisms that result from the task are concomitant and probably closely articulated with these conscious access mechanisms. Therefore, our results suggest that, while the P300 is not a signature of conscious access per se, it reflects decision processes that are closely associated with conscious access mechanisms.

With this in hand, we would like to propose the following update to previous formulations of the global neuronal workspace model. The present results indicate that covert conscious access might be subtended not by a global workspace, but rather by a subset of it, which we may term a "global playground". This "global playground" would be a broad network of areas among which sensory representations are shared and maintained for several hundreds of milliseconds, thus offering wider cognitive possibilities than automatic unconscious processing, but with no specific agenda. When a task is required, this global playground is augmented by decision-making processes and turns into a global workspace.

**The role of the frontal lobes in overt and covert conscious access.** In the same vein, our results provide interesting elements about the role of the frontal lobes in conscious perception, which is currently hotly debated[51,52]. According to our source reconstruction analysis, some frontal areas might be an integral part of the core network for spontaneous conscious access even in the absence of an overt task, as attested by late and sustained activations in inferior frontal sources during passive listening (Fig. 4C, D). Other frontal areas, such as the Supplementary Motor Area, in contrast, completely disengaged during passive listening.

At this stage, we cannot exclude the possibility that the frontal activations observed here under passive listening are artifacts of EEG-source reconstruction, merely reflecting a spillover of temporal sources. Indeed, and especially with EEG, source reconstruction is known to be susceptible to misattribution. Therefore, a possible interpretation of our results is that passive listening evokes late and sustained local activations within the temporal lobes, not necessarily connected to a wider network. And indeed, the topography of this late activity is evocative of focal temporal sources (Fig. 4A), consistent with this interpretation.

At face value, however, the reconstruction analysis suggests that the frontal cortex plays a role even in spontaneous task-free conscious processing, and that performing a task further emphasizes frontal activity by recruiting additional frontal territories, notably related to motor planning. Additional observations are consistent with this interpretation: first the difference between the active and passive conditions is only visible in highly focal regions within the dorsal frontal areas (Fig. 4C). So if, as commonly admitted, active listening evokes a wide network of areas, then removing focal executive sources should still leave a wide network at play during passive listening. A second argument is independent from source reconstruction and its potential flaws, but rather relates to the dynamics of this

late waveform associated with covert conscious access: late and sustained activity is typically associated with brain-wide activations and functional coupling, as suggested both by experimental work[53] and simulations of large scale network models of the primate brain[54].

In conclusion, the present results open an interesting alternative on the type of network subtending late sustained activations in the absence of report; this issue now needs to be addressed using spatially-resolved techniques such as fMRI.

**Potential clinical applications**. Our choices for a widespread and affordable recording technique, EEG, and for a simple auditory stimulation protocol, are beneficial for future work examining the transferability of our paradigm to clinical situations. Because bifurcation dynamics can be read-out from brain activity alone in individual participants, it might potentially be applicable for diagnosing conscious processing of external stimuli in non-communicating patients[55,56]. In current practice, in order to diagnose whether a non-communicating patient consciously perceives external stimuli, we mostly rely on detecting whether the patient can overtly or covertly preform a task on the stimulus[14,57–60], or can otherwise engage in executive processing[61], presumably limiting the diagnosis to high-functioning patients. Another current approach is to distinguish different types of patients based on spontaneous brain activity or connectivity during resting state[56,62,63], but this limits interpretation in terms of the awareness of the external world. Our paradigm allows detecting the emergence of spontaneous conscious access to external stimuli, without an overt or covert task. An interesting avenue for future research is to assess whether the paradigm can be used to examine conscious processing even in patients with deteriorated cognitive functions or at earlier stages of recovery.

Specifically, a possible method to transfer the paradigm to patients could be to: (1) characterize the EEG response to external stimuli at various stimulation strengths, in individual patients, to detect the range of stimulation values producing a sharp change in the response function; (2) detect patients with bifurcation dynamics around those stimulation values, using variability analysis, to assess the presence of conscious access; (3) fit bifurcation models to trial-by-trial brain activity, so as to be able to infer fluctuations in conscious access over time in these patients.

In conclusion we propose that this framework for analyzing brain dynamics might provide the missing tool for isolating core signatures of conscious access from other neural correlates of conscious access that either precede conscious access, such as early sensory processing, or are a consequence of it, such as task-related processes[11,13,55]. These analyses and models might potentially be transposable to any type of stimulus, task or population. This opens important avenues for future work for probing conscious perception in a variety of situations where the question of consciousness remains elusive, including clinical settings with non-communicating patients[55].

## Methods

**Main experiment**. We measured the EEG activity of adult participants, while they heard sounds presented in threshold equalizing noise at different sound levels around the audibility threshold[64]. Participants performed both an active session, in which they had to report their perception of the stimuli, and a passive session, in which they had no task to perform on the auditory stimuli, but did receive visual, amodal, or mind-wandering probes (detailed below). The order of the sessions was counterbalanced across participants.

*Participants*. Following recommendations for EEG studies[65] and previous studies (e.g., ref. [10]), the total number of participants required was estimated at 20. Twenty-five native French speakers aged 18–30 years took part in the experiment. Two chose to discontinue participation before the end, and three were excluded, following pre-established criteria, on the basis of too many artifacts in the EEG recordings (more than 25% of trials containing an artifact). The remaining 20 participants, all right-handed, included 10 women, and had a mean age of 23.4 years (range 21–29).

The study was validated by the ethics committee of Paris Descartes (CERES). Participants all gave informed consent and received a compensation of 20 € per hour of attendance.

*Stimuli*. The stimuli were two vowels, French "a" and "e", of the same duration (200 ms). They were synthesized using the MBROLA software (v3.02b)[66] with the following parameters: allophones 'a' and 'e' from the FR4 database (French female voice), 200 ms duration, 200 Hz pitch at 50% of the duration. These two vowels were chosen because of their close identification performance profiles, as revealed by behavioral pilots. Vowels were superimposed on a special background noise that equalizes the masking of all frequencies for human ears: Threshold Equalizing Noise or TEN[26]. The noise was played continuously without repetition for the duration of each block. A different noise file was played for each block. The stimuli were presented at five levels of signal-to-noise ratio (−13 dB, −11 dB, −9 dB, −7 dB, −5 dB). We previously determined, in a behavioral pilot, that the central level (−9 dB) corresponded to threshold audibility level in most participants. Some trials were composed of noise alone, without superimposed stimuli. We ran the same number of trials (160) for each of the stimulation levels (including "noise alone").

*Stimulus presentation and task*. For each participant, the experiment included two sessions on two different days, with the same stimuli but a different task: the "active" session required attentive listening, and a behavioral report on the stimuli; the "passive" session involved passive listening, where the task on the stimuli was replaced by distracting tasks as well as tasks allowing to estimate mind wandering[67,68].

Each trial started with the presentation of a fixation point, which remained on screen until the appearance of a response screen. The target vowel could be played any time between 1 and 3 s after the beginning of the trial. It lasted 200 ms, and was followed by a random delay between 2 and 3 s before a response screen was presented. Participants were asked to keep their eyes on the fixation point and to avoid moving or blinking, except during the response periods. The TEN noise was present continuously throughout each block; it stopped only during the pauses between the blocks. Participants used the computer mouse to give their response. Written instructions were provided to the participants, and we checked with them for their understanding. Each experimental session started with a training block.

During the active session, participants were asked to perform two tasks on each trial:

1. Identify the vowel: the two response options, "A" and "E", were shown to the left and right of the fixation point (initial position of the mouse at fixation), with the side assigned to each vowel changing between blocks, as announced at the beginning of each block. Positioning the cursor with the mouse over one of the letters highlighted a square around this choice. The choice was validated by a left click. Participants were asked to make a guess when they were not able to identify the target.
2. Report the subjective audibility level of the vowel: once they had responded to the first question, participants were asked to report how well they heard the target by placing a cursor on an analogous scale represented by a horizontal bar, labeled "0" on the left and "max" on the right. The initial position of the cursor was random. Participants could slide the cursor on the scale with the mouse, and validated their choice with a left click. They were instructed to use the entire scale, as follows: answer zero when they did not hear anything, use the left half of the scale for cases where they doubted a vowel was present, and the right side when they were certain a vowel was present; move the cursor to the right when the sound was more audible, independently of their ability to identify it. The highest answer was to be used when the vowel was perceived as the most audible stimulus in this experiment. The scale had 11 levels (0–10), "0" and "max" at both ends of the scale were the only landmarks presented to the participants.

Prior to the experiment, the participants were trained on both tasks during a training block where they received feedback on identification responses and where vowel at maximum volume were signaled, to allow participants to adjust their audibility judgment. During the experimental blocks, feedback on identification performance was only given at the end of each block.

During the passive sessions, the stimulation periods were identical to those in the active session, but at the end of each stimulation period, the task on the stimulus was replaced by one of the following four tasks, randomly intermixed across trials:

1. A speeded response task: participants were required to click on the mouse whenever a large green circle was displayed at fixation.
2. Mind wandering probe: participants were asked to answer the question "What is on your mind just now?", by choosing one of the four options: "the sound ", "the task", "my thoughts", "nothing/I feel sleepy".
3. Quiz: participants were asked to answer simple questions (arithmetic operations, questions of general culture…) with four answer choices.
4. Just a "Click to continue" message.

At the end of each block, participants answered a questionnaire on their experience (attention, mind-wandering…) during the previous block. The questionnaire is reproduced below:

In how many trials did you hear and recognize a vowel?

In how many trials did you hear something without recognizing it?

How often were you thinking of the sound? (0) never (1) once (2) sometimes (3) often (4) very often.

How often were you thinking of the various tasks you had to do? (0) never (1) once (2) sometimes (3) often (4) very often.

How often were you thinking about visual fixation, eye blinks, and how comfortable or uncomfortable you were? (0) never (1) once (2) sometimes (3) often (4) very often.

How often were you thinking about something else? (0) never (1) once (2) sometimes (3) often (4) very often.

How often was your attention captured by something in the environment that was not the experiment? (0) never (1) once (2) sometimes (3) often (4) very often.

How often were you falling asleep? (0) never (1) once (2) sometimes (3) often (4) very often.

Each session contained 960 trials (2 vowels × 6 sound levels × 80 repetitions), grouped into 20 blocks of 48 trials (about 7 min per block), for a total duration of 2.5–3 h with pauses. An optional additional block was provided in the event of a technical problem leading to the loss of part of the data. By combining the two vowels, we obtained 160 trials per sound level (before removing the artifacts). For the last ten subjects, we included an additional SNR level of −3dB in the passive session, in prediction of a shift in auditory threshold during this session relative to the active one.

*Material.* The experiment was carried out using an EEG system from Brain Products with 64 active electrodes, placed according to the standard 10–20 arrangement. Recordings were performed using the software provided by Brain Products (BrainVision Recorder 2016 release). The reference electrode was FT10 (right preauricular). Recording was continuous throughout the experiment, with a sampling rate of 500 Hz. Following the manufacturer's recommendations, a high pass filter at .003 Hz (time constant of 60 s) was applied during recording.

The experiment was coded using Matlab and the Psychtoolbox (http://psychtoolbox.org/)[69–71], and ran on a PC with an ASIO certified sound card for precise timing of auditory stimuli. The timing and synchronization of the auditory and visual stimuli was verified to be millisecond correct using an oscilloscope. The audio stimuli were presented via a supra-auricular headset (Beyerdynamic DT 770 PRO 80 ohm for all experiments except for the first control experiment where we used a Sennheiser HD 429). A pilot study allowed verifying that the headset did not cause any interference to the EEG recordings. The visual stimuli were displayed in gray over a black background, on a 50 cm diagonal screen (format 4/3), with a refresh rate of 66 Hz. Participants were seated 60 cm from the screen.

*Data preprocessing.* The following processing steps were applied to the EEG data using the Matlab based FieldTrip toolbox (http://www.fieldtriptoolbox.org/)[72]:

- High-pass filtering at 0.4 Hz and stop-band at 50 Hz with deletion of the linear component, performed on the continuous signal.
- Visual detection of electrodes with poor signal and reconstruction by interpolation of neighboring electrodes (separately for each experimental block).
- Removal of eye movement artifacts (blinks and saccades) was performed by identifying and removing the eye movement components using an independent component analysis (ICA) (separately for each block).
- Epoching around the onsets of the vowel, from −500 ms to +2000 ms. For trials with no vowel, a fictitious sound start was randomly chosen, matching the statistics of vowel presentation in the other trials.
- Manual rejection of remaining artifacts: visual inspection of the trial-by-trial data, and exclusion of trials containing artifacts (jumps, drifts, blink residues). On average across participants 94% (±4%) of trials were retained, giving an average of 150 trials per sound level condition (including "noise alone"; range [127–160]).
- Re-referencing to the average of the electrodes
- Baseline correction over the 500 ms preceding the onset of the stimulus.

As indicated below, two types of low-pass filtering were used during the course of the analysis:

- for visualization purpose, low-pass filters at 30 Hz were applied to time-course data: ERPs in Fig. 1E. and time course of classification sensitivity Fig. 2C second column.
- to reduce high-frequency noise in individual data for analyses carried out time-sample by time-sample (instead of large time windows of interest), a low-pass filter at 10 Hz was applied prior to the analyses.

*Event-related analysis, mean and variability profiles.* For each participant, the signal at each electrode was averaged across trials within each experimental condition to obtain event-related potentials. Grand average were then obtained by averaging these evoked potential across participants. For display purpose, the time courses of these grand averages were low-pass filtered at 30 Hz (Fig. 1E). The grand average topographies were obtained with the same data (no filtering), averaged over the different time windows of interest (Figs. 1F, 2C, 4A).

For establishing the mean and variability profiles of the signal over a group of electrodes, the trial-by-trial signal at each time point was first averaged across these electrodes. Then the temporal average of this signal for selected time windows was computed for each trial. This constituted a summary of the trial-by-trial signal over a specific group of electrodes and a specific time window. For each subject we then computed the mean and the standard deviation of this summary signal across trials, separately for each stimulation level. In order to adjust for variations across subjects in the overall mean and variability levels, the mean and standard deviation recorded in the absence of stimuli ("noise") was subtracted respectively to the means or standard deviations observed at the other stimulation levels. These mean and variability response profiles were then averaged across participants (Fig. 1F).

*Multivariate pattern analysis (MVPA).* Multivariate pattern analysis was performed using the MNE-Python toolbox (https://martinos.org/mne/stable/index.html)[73,74], with a suit of tools for temporal decoding and generalization of decoding developed by Jean-Rémi King[75], who also developed custom-made scripts to fit the particular purpose of the present analysis.

The aim of the analysis was to assess how well the multivariate pattern of activity across electrodes on each trial predicted whether a sound stimulus had been presented or not on that trial. The procedure was as follows: in a 10-fold cross-validation procedure, a linear classifier was first trained to classify the presence versus absence of a vowel in a training subset of data including stimulus absent trials and stimulus present trials with maximal SNR (9/10th of the trials in each of these conditions), using a l2-regularized (C = 1) logistic regression. The classifier was then tested on the remaining trials: the remaining target absent trials, the remaining target present trials with maximal SNR, but also 1/10th of the trials with the other SNRs (generalization of the classification across conditions; see Fig. 2A). It is important to highlight that there was always a similar number of trials (150 ± 6) in each of the two conditions that were compared (vowel absent versus vowel present at a specific SNR). Trials for training and trials for testing were taken from distinct blocks of trials separated by rest periods. For each test trial we extracted a prediction value (i.e., the signed distance to the decision plane), which was all the more positive that the prediction was in favor of the presence of a vowel, and all the more negative that the prediction was in favor of its absence. Classification sensitivity at each cross-validation step was evaluated by comparing the distribution of these predictions for target present trials at each SNR against target absent trials using a signed-rank test, to derive an area under the receiver operating curve (AUC) (Fig. 2). At the end of the cross-validation procedure, the intermediate classification sensitivities were averaged to derive global classification sensitivity for each experimental condition. Beyond this classical measure of classification performance, we also kept trial-by-trial prediction values, as required for testing our different dynamical models and predictions. This procedure was applied at each time step independently.

*Temporal generalization patterns.* Temporal generalization was assessed by training at a specific time step and testing at other time steps. Notice that, even for temporal generalization, training was always performed on the reference contrast: maximal SNR versus stimulus absent. To reduce computation time and data size for the temporal generalization analysis we applied a decimation factor of 5 (from a sampling rate of 500 Hz rate to 100 Hz). Other than that the procedure was identical to the one described above.

As with evoked potentials, classification sensitivities at different SNRs were filtered along the testing time with a low-pass filter at 30 Hz for visualization purpose (Fig. 2B, C). For each SNR level, classification sensitivity at each training and testing time was tested against 0.5 (chance) using a two-sided non-parametric sign test across subjects[76], and these statistics were then corrected for multiple comparison using the False Discovery Rate Benjamini–Hochberg procedure[77,78]. In Fig. 2B black contours indicate periods where classification was significantly above chance with $p$-corrected < 0.05.

*Generalization across active and passive sessions.* For each subject we performed generalization of classification across the active and passive sessions for the contrast "stimulus present with SNR −5 dB" versus "stimulus absent". At each time point, a classifier was trained on the data of the active session and tested on the data of the passive session (or conversely) using a ten fold cross-validation procedure, as described for generalization across different SNR levels (see above). Temporal generalization patterns and group-level statistics on these temporal generalization patterns were carried out with the same procedure described above for generalization across SNRs (Fig. 4B).

*Analysis of trial-by-trial projected activity following MVPA, over time windows of interest.* Trial-by-trial predictions evaluated by MVPA constituted an estimate of stimulus-related activity across electrodes; we also call these measures projected

activity. Temporal windows of interest were determined based on the general decoding and temporal generalization profile observed for the maximal contrast (maximal SNR versus noise only, corresponding to squares in Fig. 2B top panel). We averaged prediction values over these temporal windows of interest along both the training and the testing time (without filtering), and then calculated the mean and variability (SD) of this activity across trials as a function of SNR (Fig. 2C). Since the baseline variability level, observed for trials with noise only, was quite different across subjects, for each individual and each time window this baseline was subtracted to the SD observed for the other stimulation conditions, before averaging the variability profiles across subjects (Fig. 2C rightmost column).

We also computed the correlation between the neural variability profiles and the behavioral variability profiles at each time window. At the level of individual subjects, we computed the correlation coefficient between the neural variability profile at each time window and the behavioral variability profile of this individual; these individual correlations were thus carried out on as many points as SNR levels in the experiment (6 in the active sessions). We then used Student t-tests on these individual correlation coefficients to assess whether they were significantly different from 0 at the group level.

We next looked at the distributions of trial-to-trial MVPA projected activities and compared them with the distributions predicted by the different dynamical models (Supplementary Fig. 2). For each individual the trial-by-trial projected activities within each time window of interest were first z-scored, with all conditions together, in order to homogenize the range of prediction values across participants. Then for each subject the distribution of activity within a particular experimental condition was assessed using kernel density estimation (ksdensity function in matlab, default bandwidth, i.e., optimal for normal densities) over 91 points between −9 and +9. These individual distributions were averaged across subjects to obtain the group distributions shown in Supplementary Fig. 2B. "Heard" and "not heard" trials were sorted based on audibility ratings, ratings at 30% or above being considered as "heard" (30% is the median audibility observed at threshold stimulation −9 dB, see Fig. 1D).

We conducted a linear mixed-effect model on mean activity across trials within a time window of interest (300–400 ms), with conditions SNR × report (heard/not heard) (Fig. 3C). Our mixed-effects model included both random intercepts for each participant, as well as random slopes for report (heard/not heard) and SNR and their interaction: this model had significantly lower deviance (83.8) than a model that only included intercepts as random effects (deviance 212.0, $\chi^2_2 = 128.2, p < 10^{-15}$) or one that included only intercepts and main-effect slopes (deviance 154.9, $\chi^2_4 = 71.1, p < 10^{-13}$). We verified that the residuals are approximately normally distributed using a normal Q–Q plot. Our model showed significant effects of report (likelihood ratio test, $F_{1,24.3} = 30.3, p = 1.12 \times 10^{-5}$), SNR ($F_{1,20.5} = 54.7, p = 3.27 \times 10^{-7}$) and their interaction ($F_{1,19.4} = 7.86, p = 0.011$).

*Analysis of trial-by-trial projected activity following MVPA, time-sample by time-sample.* We then analyzed the dynamics of projected activity time-sample by time-sample over the course of the whole time window. Only the projected activity derived from training and testing at the same time were retained (corresponding to the diagonal of the temporal generalization patterns shown in Fig. 2B. with no decimation factor). In order to reduce high-frequency noise in individual data, a low-pass filter at 10 Hz was applied prior to the analyses. For each subject and at each time point we derived the correlation coefficient of the mean and variability profiles observed on projected activity, with the subject's profile of mean audibility and variability in audibility collected during the active session (Fig. 3D). These individual correlations were thus carried out on as many points as SNR levels in the experiment (6 in the active sessions). Since subjective audibility was not collected during the passive session, we used each subject's behavioral profile during the active session to perform the correlation with the neural activity recorded during the passive session. A shift in the auditory threshold between active and passive sessions was to be expected, therefore, we determined for each subject the value of the shift for which behavioral and neural profiles in the passive session correlated best: the behavioral profiles in variability were either shifted by 1 dB (the audibility mean or SD at −13dB was replaced by the audibility for noise, the audibility mean or SD at −11 dB was replaced by the audibility mean or SD at −13 dB and so on) or by 2 dB. For each subject we retained the shift for which correlation with neural activity in the passive session was the strongest, i.e., the sum of correlation coefficients above 3 sd of the baseline was maximal. The best shift was of 1 dB for 9 of the subjects, and 2 dB for 11 of the subjects. We then used Student t-tests on these individual correlation coefficients to assess whether they were significantly different from 0 at the group level (Figs. 3D and 5B).

The next analyses (model fitting and model comparison) are based on the same neural data (projected activity along the diagonal low-pass filtered at 10 Hz) but considered trial by trial. The data of each individual trial was averaged over 30 ms time windows in order to reduce computation time and further stabilize the signal. We considered 3 different dynamical models, plus a "null" model that served as a baseline (Model 0).

*Model 1: unimodal linear.* In this model the activity evoked at each SNR level follows a Gaussian distribution with standard deviation σ centered on a mean μ that increases linearly with SNR. In other words, the probability to reach activity

level x for an input SNR of *snr* is given by:

$$p(activity = x | SNR = snr) = \frac{1}{\sigma(snr)\sqrt{2\pi}} e^{-\frac{(x - \mu(snr))^2}{2\sigma(snr)^2}} \quad (1)$$

with

$$\begin{cases} \mu(snr) = slope \times (snr - maxsnr) + \mu_{maxsnr} \text{ if } \mu(snr) > \mu_{noise} \\ \qquad\qquad\qquad \text{and} \\ \mu(snr) = \mu_{noise} \text{ if } \mu(snr) \le \mu_{noise} \end{cases} \quad (2)$$

and

$$\sigma(snr) = slope_\sigma \times \mu(snr) + intercept_\sigma \quad (3)$$

where *maxsnr* is the maximal input value in the experiment (either −5 dB or −3 dB, for 10 subjects in the passive session), $\mu_{maxsnr}$ is the average activity for this input value (Note that the intercept can be expressed as $\mu(0) = \mu(maxsnr) - slope \times maxsnr$. The current formulation was preferred to the classical $y = slope \times x + intercept$ so that fitting directly provided an estimate of mean activity for one of the SNRs that were actually presented during the experiment.), and $\mu_{noise}$ is the average activity in the absence of stimulus. Note that the mean activity evoked by an input cannot be below the mean activity in the absence of input. This model was used for illustration purposes (Fig. 1A) but it was not included in the model fitting and comparison since it can be considered as a special case of Model 2.

The free parameters in this model are:, ***slope, $\mu_{maxsnr}$, $\mu_{noise}$, $slope_\sigma$, $intercept_\sigma$***

*Model 2: unimodal non-linear.* In this model the activity evoked at each SNR follows a Gaussian distribution with standard deviation σ centered on a mean μ that increases non-linearly with SNR following a logistic function, modified so that baseline activity level can be different from zero. For this model, the probability to reach activity level x for an input SNR of *snr* is given by:

$$p(activity = x | SNR = snr) = \frac{1}{\sigma(snr)\sqrt{2\pi}} e^{-\frac{(x - \mu(snr))^2}{2\sigma(snr)^2}} \quad (4)$$

with

$$\mu(snr) = \frac{L}{1 + e^{-k(snr - threshsnr)}} - \frac{L}{1 + e^{-k(maxsnr - threshsnr)}} + \mu_{maxsnr} \quad (5)$$

and

$$\sigma(snr) = slope_\sigma \times \mu(snr) + intercept_\sigma \quad (6)$$

where L determines the curve's amplitude between the two asymptotes, k is the steepness of the curve, *threshsnr* is the sigmoid midpoint, and hence the threshold SNR, *maxsnr* is the maximal input value in the experiment (either −5 dB or −3 dB) and $\mu_{maxsnr}$ is the average activity for this input value. The constant term is expressed in relation to the point ($maxsnr$, $\mu_{maxsnr}$) because the estimated $\mu_{maxsnr}$ can easily be compared with the measured activation at maximal SNR.

The free parameters of this model are: ***k, threshsnr, $\mu_{maxsnr}$, L, $slope_\sigma$, $intercept_\sigma$***

*Model 3: bifurcation.* In this model the activity evoked at each SNR has a probability $\beta(snr)$ to belong to a high state (Gaussian distribution centered on $\mu(snr)$) with $\mu(snr)$ increasing linearly with SRN level), and a probability $(1 - \beta(snr))$ to belong to a low state (Gaussian distribution centered on $\mu_{noise}$, the baseline activity observed in the absence of stimulation). For this model, the probability to reach activity level x for an input SNR of *snr* is thus given by:

$$p(activity = x | SNR = snr) = \beta(snr) \times \left( \frac{1}{\sigma\sqrt{2\pi}} e^{-\frac{(x - \mu(snr))^2}{2\sigma^2}} \right) + (1 - \beta(snr))$$
$$\times \left( \frac{1}{\sigma\sqrt{2\pi}} e^{-\frac{(x - \mu_{noise})^2}{2\sigma^2}} \right) \quad (7)$$

with

$$\beta(snr) = \frac{1}{1 + e^{-k(snr - threshsnr)}} \quad (8)$$

and

$$\mu(snr) = \frac{L_{high}}{1 + e^{-k_{high}(snr - threshsnr)}} + step \quad (9)$$

Where $\beta(snr)$ reflects the proportion of "high state" trials (conscious) as a function of SNR, and follows a logistic curve between 0 and 1 with slope **k** and inflexion point **threshsnr**. Conversely, the proportion of "low state" trials (non conscious) is $1 - \beta(snr)$ Within the low state, activity across trials follows a Gaussian distribution with standard deviation σ centered on $\mu_{noise}$, which is the mean activity level in the absence of signal. Within the high state, activity across trials follows a Gaussian distribution centered on $\mu(snr)$ which follows a logistic function of SNR with baseline **step**, amplitude $L_{high}$, slope $k_{high}$ and inflection point **threshsnr**.

The free parameters of this model are: ***σ, k, threshsnr, $\mu_{noise}$, step, $L_{high}$ and $k_{high}$***

*Model 0: null model.* This model assumes that the external input has no effect on the activity. Irrespective of whether a stimulus was presented or not, and

irrespective of the SNR, activity follows a Gaussian distribution centered on $\mu$ with a standard deviation of $\sigma$. These are the two free parameters of this model.

*Bayesian models comparison.* We compared the performance of the different models in explaining the trial-to-trial data observed for each subject. The data that were fitted were the trial-by-trial projected activities (as for "analysis time sample by time sample" above) temporally averaged over 30 ms time windows in order to further stabilize these individual trial data. For each subject, each 30 ms time window, the best parameters for each model were estimated by maximum likelihood, i.e., by finding the parameters maximizing the product of the likelihoods $p(activity = x_{itrial}|SNR = snr_{itrial}\&ModelX\ is\ true)$ across the different trials (or, equivalently, maximizing the sum of the log likelihoods). The parameter search was achieved using the Nelder-Mead optimization method[79,80] as implemented in the Matlab function "fminsearch.m". In order to compare our different models, which differ in their complexity (in particular, they have different numbers of free parameters), we used the models' predictive accuracy, which we approximated using the cross-validated log likelihood of each model[81]: the best parameters were estimated on a training set of trials (4/5th of the data), and the (cross-validated) log likelihood of the model with these parameters was tested on the remaining trials (1/5th of the data, coming from other blocks of trials). The log likelihoods estimated on the test-sets were then averaged across the 5 cross-validations. At the level of individual subjects, we compared the different models by computing the difference in log likelihoods between pairs of models.

The parameter searches were initialized as follows:
Initialization for Model 0: $\mu$ is initialized to 0 and $\sigma$ to 1.
Initialization for Model 1:

- *slope$_\sigma$*: slope between the SD of activity at the minimal ($-13$ dB) and the maximal SNR (either $-5$ dB or $-3$ dB); *intercept$_\sigma$*: intercept of the corresponding line.
- *slope*: slope between mean activity for the minimal SNR ($-13$ dB) and the maximal SNR (either $-5$ dB or $-3$ dB).
- $\mu_{maxsnr}$: mean of trials with maximal SNR.
- $\mu_{noise}$: mean of trials with noise only.

Initialization for Model 2:

- *slope$_\sigma$*: slope between the SD of activity at the minimal ($-13$ dB) and the maximal SNR (either $-5$ dB or $-3$ dB); *intercept$_\sigma$*: intercept of the corresponding line.
- $k$: slope between mean activity at the minimal SNR ($-13$ dB) and the maximal SNR (either $-5$ dB or $-3$ dB).
- *threshsnr*: middle point between minimal SNR and maximal SNR (either $-9$ dB or $-8$ dB).
- $\mu_{maxsnr}$: mean activity of trials with maximal SNR.
- $L$: double the value of $\mu_{maxsnr}$.

Initialization for Model 3: $\sigma$, $\mu_{noise}$, $L_{high}$, $step$, $k$, $k_{high}$ and $threshsnr$.

- $\sigma$: the standard deviation for trials with noise only.
- $\mu_{noise}$: mean activity of trials with noise only.
- $L_{high}$: difference between mean activities at the maximal versus minimal SNR ($-13$ dB).
- $k$ and $k_{high}$: slope between mean activity at the minimal SNR ($-13$ dB) and the maximal SNR (either $-5$ dB or $-3$ dB).
- $step$: difference between mean activities at the minimal SNR ($-13$ dB) and noise only trials.
- *threshsnr*: middle point between minimal SNR and maximal SNR.

At the level of the group, we performed a Bayesian model comparison by computing the protected exceedance probability, i.e. the probability of each model to be more frequent than any other model in the general population[35,36]. Since Model 1 can be considered as a special case of Model 2, the comparison involved Models 0, 2 and 3. Correction for multiple comparisons across different time points was performed using the Simes method[82].

*Predicting conscious report based on trial-by-trial neural activity and the bifurcation model (active sessions, Fig. 3F).* As described above, we fitted the bifurcation model to individual subject's data recorded at each time point. This provides the expected distributions of activity for the two distinct "high state" and "low state" of the model, separately for the different SNRs. We used these model distributions to compute, for each trial, the likelihood that the activity recorded at this trial and time point belonged to the "high state" versus the "low state". Of note, these trial-by-trial Bayes Factors (BF: likelihood for high over likelihood for low) were computed without injecting any information about the subjects' response at any stage. We then measured how well these BFs discriminated trials where subjects reported having heard the sound versus not using an AUC (audibility criterion at 30%, as in all other analysis distinguishing "heard" and "not heard").

*Predicting mind-wandering content based on trial-by-trial neural activity and the bifurcation model (passive sessions, Fig. 5C).* We conducted the similar analysis as the one described above in the passive sessions, on trials with mind-wandering

probes. We measured how well the resulting BFs discriminated trials where subjects reported that what they had on their mind was the sound, versus something else.

*Source reconstruction.* For each individual we reconstructed the sources of evoked potentials at each time point for different stimulation levels and task conditions. Then, at the group level we performed statistical across-subjects t-tests on these reconstructed sources comparing $-5$ dB versus absent stimulation levels (in the active and the passive session) and comparing $-5$ dB in the active versus the passive sessions. Source reconstructions over the cortical surface were performed with Brainstorm[83], (http://neuroimage.usc.edu/brainstorm). We used a default anatomy from the Montreal Neurological Institute (ICBM152), at a resolution of 15002 vertices over the cortical surface. The forward model was performed using the OpenMEEG BEM method[84]. The inverse solution was calculated using a minimum norm method[85] using the sLORETA measure and no constrain on the dipoles' orientations. Statistical comparisons between different experimental conditions were performed using paired $t$-tests across the different subjects at each vertex. These statistics were corrected for multiple comparisons using a FDR method.

**Control experiment 1: passive condition only.** In order to further control for the influence of the active session on stimulus processing in the passive session, we included 10 new naïve participants who only performed the passive version of the experiment.

*Participants.* We tested 10 participants aged from 21 to 25 year-old (mean = 23.6 years). They all gave informed consent and received a compensation of 20 € per hour of attendance. The study was validated by the ethics committee of Paris Descartes (CERES). Three participants were excluded from the analysis due to poor quality of the EEG recordings, resulting in an absence of auditory evoked potential even for stimulation with highest SNR.

*Stimuli and material.* Stimuli and material were identical to the passive sessions in the main experiment, except for the range of auditory stimulation, which was shifted by 3 dB: $-11$ dB, $-9$, $-7$, $-5$, $-3$, and $-1$ dB, plus noise only trials.

*Preprocessing.* For this control experiment, preprocessing was equivalent to the preprocessing applied in the main experiment, but using the EEGlab toolbox:

- High-pass filtering at 0.4 Hz using a Chebyshev Type I filter (pass-band = 0.3 Hz, stop-band = 0.4 Hz, stop-band attenuation = 80 dB), and a low-pass Chebyshev Type II filter at 45 Hz (pass-band = 44 Hz, stop-band = 46 Hz, stop-band attenuation = 80 dB) performed on the continuous signal.
- Data were epoched around the onsets of the sound, from $-500$ ms to $+2000$ ms. For trials with no sound, a fictitious sound-start was randomly chosen, matching the statistics of sound presentation in other trials.
- A baseline correction of 500 ms preceding sound onset was applied.
- Automatic epochs rejection was performed using FASTER toolbox[86]. Three metrics were used to exclude epochs contaminated by noise: the amplitude range, the deviation from the channel average and the variance. Epochs were excluded if one of the previous metrics exceeded 3.5 times the Z-score computed over all data. This procedure was iterated until all epochs metrics were smaller than 3.5 Z-score.
- Average re-referencing.
- Finally, non-brain artefacts (eye movements, ballistocardiac noise, sensors movements and other electrical noises) were detected and rejected using independent component analysis (ICA)/blind source separation (BSS) with UW-SOBI (1001 times delays). The UW-SOBI algorithm[87] is an adaptation of the well-known SOBI algorithm[88,89].

*Data analysis.* Data analysis was the same as for the main experiment.

**Control experiment 2: replacing vowels with simple tones.** In order to further test the generality of the late waveforms as a signature of conscious processing we performed the same experiment as the main one, but with simple tones.

*Participants.* We tested 5 participants aged from 23 to 39 year-old (mean = 27 years). They all gave informed consent and received a compensation of 20 € per hour of attendance. The study was validated by the ethics committee of Paris Descartes (CERES). All participants were included in the analysis.

*Stimuli.* The stimuli were 2 tones, matched in duration to the vowels used in the main experiment (200 ms). Tone 1 and tone 2 were pure tones with respective frequencies f1 = 1000 Hz and f2 = 2236.1 Hz, and 30 ms fade-in and fade-out. They were synthetized using a MATLAB code. The tones were presented within the same background noise as in the main experiment, at 6 possible levels of signal-to-noise ratio (SNR) in the active condition ($-22$ dB, $-19$ dB, $-16$ dB, $-13$ dB, $-10$ dB or no tone) and at 7 levels of SNR in the passive condition ($-22$ dB, $-19$ dB, $-16$ dB,

−13 dB, −10 dB, −7 dB or no tone). These intensity levels were determined based on behavioral pilots over 5 independent participants showing that −16 dB corresponded to threshold audibility level. Some trials were composed of noise alone, without superimposed stimuli. The audio stimuli were presented via a supra-auricular headset (Beyerdynamic DT 770 PRO 80 ohm).

*Task.* The protocol was the same as in the main experiment, except that in the active sessions, instead of vowel recognition, participants were required to distinguish tone 1 and tone 2 (designated as "low-pitch" and "high-pitch" respectively). Four participants performed the passive session first, and one performed the active session first.

*Preprocessing.* Preprocessing was the same as for Control Experiment 1.

*Data analysis.* Data analysis was the same as for the other experiments.

**Statistics and reproducibility.** The present study includes one replication of the passive session of the main experiment (10 new participants, see above). The results were successfully replicated (Supplementary Fig. 6). The study also includes a replication using different stimuli (tones instead of vowels) for both passive and active sessions (5 new participants that took both sessions, see above). The results were successfully replicated (Supplementary Fig. 7).

**Reporting summary.** Further information on research design is available in the Nature Research Reporting Summary linked to this article.

## Data availability

The EEG data that support the findings of this study are available in OSF [https://osf.io/aw3t5/] with the identifier https://doi.org/10.17605/OSF.IO/AW3T5. Other data underlying the conclusions are available from the authors upon request. Source data are provided with this paper.

## Code availability

The custom codes that have been used to analyze these data are available in OSF [https://osf.io/aw3t5/] with the identifier https://doi.org/10.17605/OSF.IO/AW3T5.

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

## Acknowledgements

We thank Sebastien Marti, Jérôme Sackur, Stanislas Dehaene, Daphné Rimsky-Robert, Jacobo Sitt, Paolo Bartolomeo, and Lionel Naccache for useful comments. We thank Brian C. J. Moore for providing threshold equalizing noise. This work was funded by two grants from the Agence National de la Recherche to C.S.: ANR-14-CE15-0013-03 (CogniComa) and ANR-17-CE37-0004-01 (FlexConscious) and two grants to D.P.: ANR-17-EURE-0017 and ANR-10-IDEX-0001-02.

## Author contributions

C.S., G.L., M.C., and D.P. designed the experiments; C.S. developed the models; G.L., M.C., and F.S. acquired the EEG data; C.S., G.L., M.C., F.S., M.W., J.-R.K., and F.M. analyzed the data; C.S. wrote the first draft of the manuscript; C.S. and D.P. wrote the final version of the manuscript with help from all co-authors.

## Competing interests

The authors declare no competing interests.
