## [Peer Review File · Nature Communications]

Reviewers' comments:

Reviewer #1 (Remarks to the Author):

Sergent and colleagues present an EEG study in which they note that if correlates of consciousness follow "bifurcation dynamics" (as suggested by the Global Workspace Theory), there ought to be a change in the variance, more than (or in addition to) a change in the mean of neural responses. They demonstrate that audibility reports are most variable at SNRs with mixed consciousness, and that at late latencies (~250 ms and after) there is a correlation between the variability of multivariate decoding (distance to boundary) and reports. A model with bifurcation dynamics best captures the data, and this is true even in the absence of reports.

The report is interesting, and touches on many aspects that are currently central in the study of consciousness (e.g., are the correlates of consciousness primarily in the front or back of the brain? Are many of the previously reported NCCs contaminated by requests for report?). Overall, the study presents nuanced answers, for example demonstrating that consciousness does evoke long and sustained responses, but may not be the same as those frequently reported (Figure 4A).

I do have a couple of broad concerns:

1) The focus here is on variability and the results are suggested to be a "universal signature of consciousness". But really aren't the results showing that EEG responses are most variable when consciousness is mixed? It may be an index of the transition from no consciousness to consciousness, but it can't be a measure of consciousness itself.

a. Even the model shows that at very low or very high SNR, variance is the same, it's the mean that is different.

b. The reports relate to different contents of consciousness, but not necessarily to different levels of consciousness. Seemingly in one case subjects are conscious of noise, and in other cases they are conscious of a vowel.

c. It is hard to determine whether the results are trivial or not. I think the authors should emphasize in the main text that the reports of audibility are not 2 alternative-force choice, as if they were, then of course it's trivial: by definition highest variance in responses is at the PSE in 2AFC.

d. Figure 2 is fairly confusing, as Figure 2C (3rd and 4th column) are predictions. In fact, wouldn't the central analysis of the paper be figure S2? Here we don't see the variance pattern expected.

2) I think the authors need to be careful not to categorize this as a brain signature of consciousness – if anything (but see above), it is a signature at the EEG level. The first and second models presented are clearly stick figures if we talk about neural response as opposed to EEG. Neurons fire with Poisson dynamics – variance increases with the mean – and thus biologically, models 1 and 2 are just not plausible. It is not clear to me that we know enough about spiking activity-to-EEG in order to make predictions at the EEG level that would take into account what we know about neurons.

Minor comments:

1) Given that it is currently central in the study of consciousness, the authors could discuss a bit more their findings related to report and whether "consciousness is in the back or the front of the brain". Work by Hakwan Lau comes especially to mind.

2) In the introduction the authors nicely summarize the ignition literature. I missed reference to Joglekar et al., 2018 (Neuron), van Vugt et al., 2018 (Science), and Noel et al., 2019 (J. Neuroscience). The first shows that ignition patterns immerge naturally given known anatomy, and then latter show ignition at the level of single neurons.

3) The authors make reference to the fact that their study is in the auditory domain and thus helps clarify whether signatures that have been reported in visual awareness apply generally. I do believe this is meaningful, but see Noel et al., 2018 (J Cogn Neuro), for a report showing that the presence of late latency responses generalizes to audio and audio-visual domains.

4) Can the authors discuss what is the meaning of the below chance off-diagonal accuracy decoding (Figure 2B)?

5) In Figure 3 the authors do examine neural correlates as a function of report, but I find this

figure to be already at a very high level (mostly correlations are shown). Perhaps the authors could choose 1 SNR (that which generates an approximately equal split between aware and not) and show variance and mean responses as a function of report?

I sincerely hope these comments help improve an already very interesting study.

Reviewer #2 (Remarks to the Author):

The goal of this study - to discover "a universal brain signature of conscious processing independent of report" - is laudable, and of fundamental neuroscientific and psychological importance. On the face of it, it should also appeal to a broad general audience. Unfortunately however, I think this study falls short on a number counts and, as such, fails to reach its potential.

My comments (in no particular order):

One of the four key reasons that the authors give for conducting this study, is (Page 3) "First, it is unknown whether they {such non-linear input-output dynamics as a generic signature of conscious perception} generalize to other sensory modalities, or rather represent peculiarities of the visual system". Their response to this is to conduct their study entirely in the auditory domain, which is a shame because they really can't claim, based on their results, that these signatures are generalizable (i.e. to other domains). What they really need to do is to include one or more analogous experiments in other domains showing similar results. As such, their title "a UNIVERSAL brain signature..." is misleading and somewhat inappropriate. I might mind this less if they reconfigured their paper (and especially the title) to focus this on what it is - an *auditory* signature - but unfortunately, I think this would then rather seriously detract from the impact of the paper. The holy grail here is to truly find a *universal* signature of conscious processing and, given this limitation, this study falls well short of that.

The authors state on Page 3. "Second, the stimulus presentation procedures are arguably complex, whereas a neural signature of conscious perception should apply to simpler cases more representative of natural perception" and on Page 14 "Here we show that these peculiar dynamics generalize to conscious perception in a new sensory modality, audition, and in a very simple paradigm, arguably closer to ecological stimulation conditions". I am not sure I buy this. The task they choose is undoubtedly 'simple', but I'm not sure its a "simpler case" than many of the visual tasks that have been used previously. Nor do I think that detecting vowels among SCN is "more representative of natural perception" than those other tasks (how often do we try to detect vowels among SCN in everyday life?). "arguably closer to ecological stimulation conditions" perhaps, but the authors need to qualify that, because it's not clear to me that it's really true.

On Page 3 the authors state "Finally, because consciousness is attached to a single mind, a neural signature of consciousness should ultimately be observable at the level of the individual and not only as an average across participants..." and on Page 10 "Remarkably these bifurcation dynamics could be detected at the individual level in all participants". Perhaps I haven't fully understood exactly what they have done to test these claims, but while I accept that they detected these patterns in all participants, I am not convinced that the results were *predictive* at the individual level (which is, I think, what they would like to claim). To demonstrate that, I would like to have seen an experiment where they actually used their EEG signature to predict whether a participant had consciously experienced any particular stimulus or not. I appreciate that the authors may have sort of done the statistical equivalent of this, but I don't believe it's what they did in any practical way. To do so, would make this paper accessible to a wider audience. Just to be clear, what I am proposing is that, to support their claims, they would carry out the passive version of their task and at various (e.g. random) times stop the task and tell the participant whether or not they think they had perceived the stimulus that had just been presented. This might seem like a nuance, but it's critically important if the authors are going to make the sorts of claims about clinical utility that they make. e.g. Page 16 "The new signature described here opens important clinical perspectives for detecting the emergence of spontaneous conscious access to the external environment even in patients with deteriorated cognitive functions and at earlier

stages of recovery". If this is true then their method needs to be able to predict, in an individual patient, whether a stimulus was consciously perceived or not.

Page 16. "In current practice, in order to diagnose whether a non-communicating patient consciously perceives external stimuli, we mostly rely on detecting whether the patient can overtly or covertly perform a task on the stimulus (14, 43-46)" Mostly, yes, but there are a number of recent examples where this has been achieved in non-communicative patients without requiring any decision or 'active' response (e.g. Naci and colleagues, PNAS 2014). Similarly Page 17 "This opens important perspectives for probing conscious perception in a variety of situations where the question of consciousness remains elusive, including clinical settings with non-communicating patients". In general, I feel that these statements would be better justified if the authors had actually shown that this was the case. It's all too easy to claim 'clinical relevance' but in practise this is a much harder problem than finding a task that works in healthy brains and assuming, therefore, that it would i) work in patients and ii) reveal anything about those patients. I do sympathize with these authors - clinical relevance is an important aim and a lofty goal, but I think their claims are premature.

Page 5 "Unlike the identification data, audibility ratings for single trials were not binary". Why wasn't this also true for identification? Wouldn't the authors have predicted that it would be true for both audibility ratings and identification?

Page 11. "Neural responses due to decision-making should disappear in the absence of a task, but neural responses reflecting conscious access should persist". Yes, but how do you stop participants from making a decision anyway? In this situation (passively listening to vowels embedded in SCN) it seems likely that participants would naturally ask themselves the question from time to time "Did I just hear an 'E'?", particularly if they had previously been involved in the active version of the task. Again, this would be at least partially solved by a predictive study in which naive subjects were told to just do the passive task and then predictions were made about whether they perceived the stimuli or not.

Page 11. "Surprisingly, neither prediction was entirely fulfilled: late waveforms were still observed, but their topographies were very different from the sustained P3-like positivity characteristic of active listening (Fig. 4A)"

This possibly confirms my previous point. Perhaps there is a little bit of decision making in the passive task?

Many of my concerns are summarized by the authors cautious final statement on Page 13. "In conclusion the analysis of brain activity during the passive session suggests that, even in the absence of behavioral report, stimulation at threshold yields two types of neural response, presumably corresponding to trials with and without covert conscious perception" Given all my concerns above, I am not sure that *presumably* is enough to support the strong claims the authors are making here.

Reviewer #3 (Remarks to the Author):

This study exploits an interesting experimental design and advanced EEG analysis with the aim of distilling the neural events underlying conscious access by comparing measurements obtained in the presence (active condition) and in the absence (passive condition) of explicit reports. In line with previous studies employing no-report paradigms (see, for example, Pitts et al., 2014a and 2014b), Sergent and colleagues find that a prominent P3 wave associated with frontal sources of activity is absent or strongly reduced in the no-report (or passive) condition. This result is relevant as the presence of P3b wave associated with a robust activation of frontal cortical areas (obtained in active conditions) was previously proposed as a robust signature of conscious access. Interestingly, the author still find significant classification accuracy, in both the active and passive

conditions, between single trials in which the target sound was present and trials in which the target sound was absent. They also show that the variability profile of the behavioral performance in the active condition correlate with the variability of the classifier performance, in a way that can be explained by a bifurcation dynamics model. Importantly, performing cross-classification analysis demonstrates that training the classifier in the passive session (sound present vs sound absent) allows them a significant decoder performance when testing the classifier in the active session.

I find the general approach original and the results potentially relevant for the field. However, there are several issues (both conceptual/interpretational and methodological) that the authors need to address in order to ensure that the results are solid and interpretable.

Major concerns

Conceptual/interpretational

1 – since the primary objective of the paper is to contrast neural responses in the presence and in the absence of report, it is crucial that the ERPs obtained in the active and passive conditions (at -5dB) are clearly presented and compared. Hence, Figure 4 should clearly show, in addition to scalp topographies, the classic traces (voltage vs time) normally employed to illustrate P3-like components. This display is needed to qualify statements that are, in the text, rather unclear and vague, such as “Surprisingly, neither prediction was entirely fulfilled: late waveforms were still observed, but their topographies were very different from the sustained P3-like positivity characteristic of active listening” (pag 10) or “It shows that the P300 is not a “pure” signature of conscious access, since its typical topography is not observed in the absence of report, but it is a composite waveform that does include the core signature of conscious access together with decision-making processes.” (Pag 10).

How does the time course of the two traces actually differ in the report and no-report condition? A simple plot of the trace is fundamental for the reader to have a sense of the main result. When plotting the traces obtained in the two conditions, please use the same Y-scales (also, remove autoscaling in the topographical plots of Fig.4a). I would then replace the qualitative statements reported above with a simple description of this fundamental figure. It is also very useful that the reader can directly compare these waveforms to those obtained in previous studies, where the P3 has been used as a marker of consciousness in both healthy subjects and unresponsive brain-injured patients.

2- Also related to the interpretation, it is difficult to understand how the present result “validates a key prediction of the global workspace model of consciousness (17, 34, 35). In this model, conscious access arises through the broadcasting of local sensory information to a global network of areas”. For example, reference (35) clearly states that the GNW model predicts that conscious access coincides with the “ignition” of a large-scale prefronto-parietal network, whereas the present data suggest that large scale prefronto-parietal network activations are related to reporting but not to conscious access. The source modeling of passive condition (sound present vs absent) in Fig.4 shows a center of mass around the temporal opercular cortex with no local maxima in the frontal areas typically involved in the GNW model. Is there a late non-linear activation in frontal sources that is separate and temporally distinct from the temporo-parietal activation? For example, it would be important to show how do the time series of the currents (dipoles) in the temporal and the frontal cortex differ. The few sources that are weakly active in the inferior part of the frontal lobe are more likely to reflect the smearing typical of distributed inverse solutions rather than the ignition of strong independent prefrontal sources. This is another key point that needs to be clarified in the manuscript.

3 – How can the authors exclude that the late activation is not a correlate of phonemic processing (even in the passive condition)? Indeed, late components (between 200 and 300 ms) very similar to the ones found in the present work are known to be involved in phonemic processing (Dittinger et al., 2018; Koerner et al., 2017). Why not using simpler tones or clicks with slightly different features as in previous works? Would late component be strongly reduced in this case, especially in the passive condition? This possibility, if not tested, must be acknowledged. More generally, at the present stage, I would be very cautious in claiming that these components are a core signature

of conscious access (abstract and elsewhere).

4 – A further reason for caution: how can a novelty effect be excluded? N150-P250 potentials (vertex waves) can be recoded when stimuli (including auditory) occur with long interstimulus interval (tens of seconds) and then habituate becoming much smaller for interstimulus intervals down to 1 s (Mancini et al., eNeuro 2018). What was the interval between the occurrence of the target stimuli (vowels)? Was there a control for habituation at around 1 Hz? If not, it is important to acknowledge this potential confound due to parallel activation of saliency networks (Liang et al., Cer Cor 2013).

Methodological

1- the authors state that “from 250ms onwards, neural variability profiles correlated with the behavioral variability profiles” (Page 7, Line 23), and report four results with multiple comparisons correction and one without (300-400ms). I believe that performing multiple comparisons correction but not taking it into account when interpreting the results is rather misleading. In my opinion, in the case of non-significant results exact-corrected p-values should be reported and interpreted (with the additional information of significance when uncorrected). In a similar way, I believe that p-values by themselves are not very informative, and that they should be accompanied, when possible, by the corresponding statistics and degrees of freedom [and ideally effect sizes].

2- Similarly, the mixed effects model reported in (Page 8, Line 22) is not specified in the methods section. Did it include a random intercept only, random intercept and slope (etc)? Moreover, the results of this analysis (coefficients, degrees of freedom, standard deviations, post-hoc comparisons, etc.) are not reported. Did the authors check the residuals to see if assumptions were met? All this information is important in order to interpret the findings, given that significant results can be easily found with incorrect mixed effects models. Also, when interpreting this result, I assume the authors are focusing on the interaction, however, I believe that the explanation of why this is informative regarding the authors’ hypothesis should be included in the text as it might not be self-evident to the reader.

3 - In Figure 3A the authors present the results of the window from 300-400ms, which did not survive the multiple comparisons correction in the previous analysis. Why did they choose this particular time window? It seems like this choice was motivated by an a priori hypothesis, rather than in a data-driven manner. This needs to be clarified.

4 - In the analysis of the correlation at every time point, the authors refer to the prediction values (as stated in the methods section) as “neural mean and variability profiles” (in the results section) and as “neural activity” (in the figure legend). And also state that “These prediction values are a trial-by-trial measure of target-related neural activity” (Page 7, Line 14). I believe that equating neural activity with the output of a classifier is misleading, given that classification performance can be influenced by non-neural factors. In addition, this refers to the discrimination between conditions, not to the activity directly associated with each of them. The authors should use a different wording or provide an explanation of why prediction values can be equated to neural activity.

5 - The authors mention that “some trials were composed of noise alone, without superimposed stimuli”, but the number of target-absent trials is not clearly reported, which is important, given that class imbalances can bias the performance of a classifier. I believe that the number of observations of each class that were used with the classifiers (after trials rejection) should be clearly stated as it is a crucial information in order to evaluate the results.

6 - The authors state that black contours of Figure 2B (and Figure 4A) indicate periods of above chance classification, however there are statistically significant classification with $AUC < 0.5$. I infer that the authors used a two-tailed t-test. In order to test if performance was above chance, the test should have been one-tailed. Additionally, the type of t-test used is not reported in the methods section and should be described (was it Student or Welch?). Moreover, the validity of

these tests relies on the assumption of normality but this was not assessed. I would personally suggest to confirm that no violations of normality were present, to use a non-parametric approach, or to state why a parametric test would be preferred.

7 - The observation of significant misclassifications rises the question of why this could have happened? Do the authors have an explanation for this phenomenon? In addition, in the generalization across time analysis of Figure 2B there are significant classifications when training at around 250-500 and testing at 1750-2000ms (and vice versa). Can the authors comment on this discrimination between conditions that occurs almost 2 seconds after stimulus presentation?

8 - The colormap of the AUC shows discrete colors, but the AUC is a continuous variable. How were these colormaps constructed? Did the authors discretize these values?

9 - It is not clear to me how the analysis of the correlation between the neural variability profiles and the behavioral variability profiles was performed. In the methods section the authors state that the mean and variability profiles of the classification performance observed at each time window were regressed with the profile of mean and variability in audibility, at the subject level. Did this regressions include 5 observations only? The results section states a result at the group level, and mentions "t-tests on regression slope", what are the observations included in this test? Does this correspond to a test against 0 using the slopes of individual subjects? Please provide a more detailed description of this analysis.

10 - the authors declare that "from 250ms onwards, these neural variability profiles correlated with the behavioral variability profiles" (Page 7, Line 23), and report four results with multiple comparisons correction and one without (300-400ms). I believe that performing multiple comparisons correction but not taking it into account when interpreting the results is rather misleading. In my opinion, in the case of non-significant results exact-corrected p-values should be reported and interpreted (with the additional information of significance when uncorrected). In a similar way, I believe that p-values by themselves are not very informative, and that they should be accompanied, when possible, by the corresponding statistics and degrees of freedom [and ideally effect sizes]).

11 - Similarly, the mixed effects model reported in (Page 8, Line 22) is not specified in the methods section. Did it include a random intercept only, random intercept and slope (etc)? Moreover, the results of this analysis (coefficients, degrees of freedom, standard deviations, post-hoc comparisons, etc.) are not reported. Did the authors check the residuals to see if assumptions were met? I believe that all this information is important in order to interpret the results, given that significant results can be easily found with incorrect mixed effects models. Also, when interpreting this result, I assume the authors are focusing on the interaction, however, I believe that the explanation of why this is informative regarding the authors' hypothesis should be included in the text as it might not be self evident for a potential reader.

12 - In Figure 3A the authors present the results of the window from 300-400ms, which did not survive the multiple comparisons correction in the previous analysis. Why did they choose this particular time window? Was this hypothesis driven? If so, I believe that it should be clearly stated on the manuscript.

13 - In the analysis of the correlation at every time point, the authors refer to the prediction values (as stated in the methods section) as "neural mean and variability profiles" (in the results section) and as "neural activity" (in the figure legend). And also state that "These prediction values are a trial-by-trial measure of target-related neural activity" (Page 7, Line 14). I believe that equating neural activity with the output of a classifier is misleading, given that classification performance can be influenced by non-neural factors, and that it implies the discrimination between conditions, not the activity directly associated with one of them. I think that the authors should use a different wording or provide an explanation of why prediction values can be equated with neural activity.

14 - In the analysis between passive and active sessions, I suggest the author's to include a control analysis to verify how different were the late waveforms between subjects that performed

the active condition first with respect to those who performed it last. I believe that this is an important point in order to avoid possible confounding factors in the interpretation of their results.

15 - The authors provide source localization results that were computed using a template and no electrode position digitization. This can create considerable imprecisions in the spatial localization activity of the sources. How were these results at the single subject level?

16 - In (Page 5, Line 22) there is a typo: it says Figs instead of Fig.

17 - Throughout the manuscript the authors report mostly binary p-value based statistical results ($p < 0.5 > p$). As mentioned above, I think that, when possible, including all the relevant information required to evaluate the results (statistic, degrees of freedom, etc) and performing the proper checks of each test's assumptions is of vital importance to be able to assess their validity.

Minor

- I think that the fact that the order of experimental sessions ("active" and "passive") was counterbalanced across participants should also be explained in the methods section corresponding to the experimental design, in order to avoid misconceptions on potential readers that might not read the article linearly.

- I believe that mentioning that the AUC is a measure of sensitivity is a terminological imprecision. A measure of sensitivity would be, for instance, the True Positive Rate. I would suggest the authors to use a more precise wording.

Response to reviewers

General comments to all reviewers, summary of the changes:

We have now revised our manuscript entitled “Bifurcation in brain dynamics reveals a signature of conscious processing independent of report” (NCOMMS-19-34713A). We would like to sincerely thank the Reviewers for their thorough, insightful, and constructive comments. All of their suggestions and concerns have been addressed, as detailed in the point-by-point response below. Briefly, the main changes are as follows:

- Reviewers #2 and 3 questioned whether the bifurcation dynamics that we claim as a core signature of conscious access was truly universal, notably asking whether it might be dependent on language-specific processes linked to vowels. The Editor confirmed that new data was required to address this point. We thus conducted a completely new EEG control experiment, using this time pure tones instead of vowels. The results were replicated, showing that late bifurcation dynamics occur in different experimental settings (Fig.S7). We have also reworked the text: we changed the title to avoid using the adjective “universal”, which we recognize was premature, and we now develop in more details the arguments in favor of the generality of the neural signature that we describe.
- Reviewer #1 remarked that some of our models of neural dynamics were not biologically plausible, since variance was not allowed to change as a function of mean activity, in contrast with what is often observed in neural systems. We upgraded the models to include this feature. Thanks to this improvement, the link between model and data has been extensively strengthened: our models can now predict all qualitative trends of the data, including the linear increase of variance with mean observed during the initial period that precedes bifurcation dynamics (new version of Fig.1).
- Following comments from reviewer #2 and 3, we conducted a new analysis testing the effect of session order (active / passive) and acquired a new set of EEG data on naïve subjects who only undertook the passive sessions. We confirmed that the late bifurcation dynamics observed during task-free listening could not be explained by covertly performing the active task (Fig. S6).
- Reviewer #2 wondered whether our claimed signature of conscious access could be used to predict behavior. In addition to a new split-trial analysis of the data (new Fig.3), we also ran a completely new set of analyses to further address this major conceptual issue. Using our model-based approach, for which we fitted the bifurcation model to neural data, we show that we can predict trial-by-trial conscious reports of individual participants when a task was required (Fig.3F). Most interestingly, this prediction could be achieved without ever accessing the actual behavioral reports during the fitting procedure, thanks to the structure of the model. So, we could also use the same approach to predict the conscious content of participants for random mind-wandering probes in the task-free sessions (Fig.5C). We believe that this new analysis establishes a very strong link between bifurcation dynamics and conscious access with or without report. It also brings further encouraging arguments for using this approach in clinical applications.

As shown by these main changes, we truly believe that the Editor and Reviewers’ comments have provided us with ideas to considerably strengthen the manuscript. We are grateful for

this opportunity and hope that you will find the new version of the manuscript suitable for publication.

Our detailed responses to the reviewers are in blue.

The changes made to the manuscript itself to address the reviewers' concerns are highlighted in yellow.

Response to Reviewer #1:

Sergent and colleagues present an EEG study in which they note that if correlates of consciousness follow “bifurcation dynamics” (as suggested by the Global Workspace Theory), there ought to be a change in the variance, more than (or in addition to) a change in the mean of neural responses. They demonstrate that audibility reports are most variable at SNRs with mixed consciousness, and that at late latencies (~250 ms and after) there is a correlation between the variability of multivariate decoding (distance to boundary) and reports. A model with bifurcation dynamics best captures the data, and this is true even in the absence of reports.

The report is interesting, and touches on many aspects that are currently central in the study of consciousness (e.g., are the correlates of consciousness primarily in the front or back of the brain? Are many of the previously reported NCCs contaminated by requests for report?). Overall, the study presents nuanced answers, for example demonstrating that consciousness does evoke long and sustained responses, but may not be the same as those frequently reported (Figure 4A).

I do have a couple of **broad concerns**:

- 1) The focus here is on variability and the results are suggested to be a “universal signature of consciousness”. But really aren't the results showing that EEG responses are most variable when consciousness is mixed? It may be an index of the transition from non-consciousness to consciousness, but it can't be a measure of consciousness itself.
 - a. Even the model shows that at very low or very high SNR, variance is the same, it's the mean that is different.
 - b. The reports relate to different contents of consciousness, but not necessarily to different levels of consciousness. Seemingly in one case subjects are conscious of noise, and in other cases they are conscious of a vowel.

We found all these remarks very relevant. They called for clarifications around the term “signature of consciousness”, and for the use of more precise wordings. As noted by reviewer #1, the burst of variability that we observe is not directly a signature of conscious processing, but a **signature of the transition between non-conscious and conscious processing**. As such, it can, in turn, be used as a **signature of conscious state**: if an individual shows such bifurcation of global brain activity, it reveals that their brain is in a state that handles both unconscious and conscious processing of external stimulation, in other words, a conscious state. Furthermore, here we use these bifurcation dynamics as a tool to isolate **the core neural signature of conscious processing** from other neural correlates that are either premises (early sensory processing) or consequences of it (task related processing) (Aru, Bachmann et al. 2012, Sergent and Naccache 2012): indeed, bifurcation dynamics mark the moment when brain activity spontaneously splits into either a “high” or a “low” state of activity across

trials, matching conscious versus unconscious processing of the same stimulus for stimulation levels around the detection threshold.

The **new text** now clearly reflects the different senses in which we use the word “signature”, and their relationship. Below are a few examples of how we amended the text. Furthermore, a **new analysis** shows how we can use the bifurcation model to predict conscious processing based on trial-by-trial neural activity (**Fig.3F** and **Fig.5C**).

Title

Bifurcation in brain dynamics reveals a signature of conscious processing independent of report.

Introduction page 3:

However, additional steps are required before we can confirm that bifurcation dynamics are a generic signature of the transition between non-conscious and conscious perception.

Discussion page 19:

Based on those results, we propose that bifurcation dynamics are a general signature of the transition between conscious and non-conscious processing, which in turn allows identifying the neural signature of conscious access proper: a global brain activity arising between 250 and 700ms post-stimulus, supplemented by additional decisional processes when a task is required.

c. It is hard to determine whether the results are trivial or not. I think the authors should emphasize in the main text that the reports of audibility are not 2 alternative-force choice, as if they were, then of course it's trivial: by definition highest variance in responses is at the PSE in 2AFC.

We agree that this is a key point. We now emphasize it in the new manuscript, results section, page 6:

Importantly, audibility was assessed using a continuous scale so we could derive a meaningful variability measure for behavior. In a 2-alternative forced choice task, such as the one used for identification, variability is maximal at 50% performance by construction (28). In contrast, the continuous measure of audibility imposed no such constraints, as variability could vary freely at all levels tested, provided they were sufficiently remote from floor or ceiling. Thus, any putative discontinuity in sensation may manifest itself by a burst in variability in the audibility judgments, following the modeling described in Fig 1 A-C (see also (16, 17)).

d. Figure 2 is fairly confusing, as Figure 2C (3rd and 4th column) are predictions. In fact, wouldn't the central analysis of the paper be figure S2? Here we don't see the variance pattern expected.

We acknowledge that the use of the term “prediction” in Fig2C might have been confusing: but this measure is really a measure of brain activity related to the presence of the stimulus. We now make this point clear throughout the paper, and replaced the term “prediction” by the term “projected activity”. This starts with the **legend and labels used in Fig.2**, and in the text, result section **page 8**:

The MVPA transforms the multivariate EEG activity at each trial, each time point and for each participant into a representation best expressing the neural response to stimulus presence versus stimulus absence (each dot in the illustrative Fig.2.A.). Just as an evoked potential, the

MVPA projection can be viewed as a measure of brain activity over a region of interest, but with the region of interest derived from the data to optimally distinguish the presence of a stimulus. This projected activity can be used to try and predict whether the stimulus was indeed present or absent on that trial: the distance from the decision criterion – i.e. the criterion that best separates stimulus present versus absent trials (the plain line in Fig.2.A.) – indicates the strength of the prediction towards stimulus presence (positive distance) or absence (negative distance). In the following, we term this distance the “projected neural activity” and use this continuous measure to assess variability across trials from the MVPA output.

Figure S2 uses the same measure, but splits the trials according to subjects’ reports (heard versus not-heard). Here the burst in variability for intermediate stimulation strength disappears, as predicted by the bifurcation model. Indeed, if this burst in variability is due to the coexistence of two groups of trials (high and low activity) for threshold stimulation, and if these two groups correspond to conscious and non-conscious trials, then splitting the trials according to conscious report should eliminate the burst. This is what we observed in this figure.

2) I think the authors need to be careful not to categorize this as a brain signature of consciousness – if anything (but see above), it is a signature at the EEG level. The first and second models presented are clearly stick figures if we talk about neural response as opposed to EEG. Neurons fire with Poisson dynamics – variance increases with the mean – and thus biologically, models 1 and 2 are just not plausible. It is not clear to me that we know enough about spiking activity-to-EEG in order to make predictions at the EEG level that would take into account what we know about neurons.

We agree with this remark, and we fixed this in the new version of the article: **unimodal models (1 and 2) now include new parameters allowing the variance across trials to change linearly with the mean.** Thanks to this insight from reviewer #1, **model 2 is now even closer to the experimental data in the first period of activity, the one that shows unimodal dynamics (before 250-300ms):** indeed we see from Fig1F for example that variance profiles are not flat before 300ms post-stimulus, but instead increase monotonically with mean.

Importantly, **this improvement in modeling unimodal dynamics does not affect our central argument and our central results:** as in the previous version of the paper, what critically distinguishes bifurcation dynamics from unimodal dynamics is the prediction that variance should show a **non-monotonic** change as a function of stimulation strength. In contrast, unimodal dynamics should be associated with a monotonic change of variance with stimulation strength, irrespective of whether variance changes linearly with mean or stays flat.

Below are the details of the changes we implemented:

- The **new versions of Model 1 and Model 2** (unimodal linear and unimodal non-linear dynamics) now include the free parameters of slope and intercept that describe the linear relationship of variance with mean, following this formula: **variance = slope*mean + intercept.** These changes are detailed in the **Methods section** (Models definition, starting page 47). Model 3 (bifurcation) has also been improved: while in the previous version of the article mean activity in the “high state” varied linearly with stimulation strength, in this new version of the article it is allowed to vary as a

logistic function of stimulation strength. Variance of the low and high states however remains fixed, as in the previous version (same value for both states, no variation with mean), so as to insure good separability of Model 2 and 3.

- Figure 1A. and B., which illustrate the different models, has been changed to reflect these new versions of the models
- The results of the fits with the new versions of the models can be found in Figure 3.E

Minor comments:

1) Given that it is currently central in the study of consciousness, the authors could discuss a bit more their findings related to report and whether “consciousness is in the back or the front of the brain”. Work by Hakwan Lau comes especially to mind.

This is indeed an important debate, and we have included the following paragraph to the discussion (p.23):

In the same vein, our results provide a nuanced picture about the role of the frontal lobe in conscious perception, which is currently hotly debated (49, 50). Source reconstruction suggests that some frontal areas might be an integral part of the core network for spontaneous conscious access even in the absence of an overt task, as attested by late and sustained activations in inferior frontal sources during passive listening (Fig4.C and D). Other frontal areas, in contrast, completely disengaged during passive listening, such as the Supplementary Motor Area. This suggests that the frontal cortex may play a role even in spontaneous task-free conscious processing, and that performing a task further emphasizes frontal activity by recruiting additional frontal territories, notably related to motor planning as decisions are often followed by action. However, this proposition needs to be confirmed using techniques with better spatial resolution, such as fMRI.

2) In the introduction the authors nicely summarize the ignition literature. I missed reference to Joglekar et al., 2018 (Neuron), van Vugt et al., 2018 (Science), and Noel et al., 2019 (J. Neuroscience). The first shows that ignition patterns immerge naturally given known anatomy, and then latter show ignition at the level of single neurons.

Joglekar et al. and van Vugt et al. were cited in the Discussion section of the previous version of the manuscript, but we were missing the more recent Noel et al 2019. Thank you for pointing this out, this is a very relevant study indeed. It is now cited, along with van Vugt et al, in the introduction of the new manuscript (p.3):

Such effects have also been detected in infants (19) and more recently in neural recordings from non-human primates (20, 21).

3) The authors make reference to the fact that their study is in the auditory domain and thus helps clarify whether signatures that have been reported in visual awareness apply generally. I do believe this is meaningful, but see Noel et al., 2018 (J Cogn Neuro), for a report showing that the presence of late latency responses generalizes to audio and audio-visual domains.

Again we are grateful to reviewer #1 for pointing us to this very relevant literature. It is now cited in the new manuscript, discussion page 20:

An increasing number of studies now emphasize the importance of comparing neural correlates of conscious access across different sensory modalities in order to establish the generality of candidate neural signatures of consciousness (22, 23, 43-45).

4) Can the authors discuss what is the meaning of the below chance off-diagonal accuracy decoding (Figure 2B)?

Below chance off-diagonal accuracy decoding is commonly observed in temporal generalization diagrams (King and Dehaene 2014): it denotes that some neural activity patterns can recur with reversed polarity. This precision has been added page 9 of the new version of the manuscript:

Below chance off-diagonal performance indicated that some early patterns of activity recurred later with reverse polarity, as has already been observed with the temporal generalization technique (30).

5) In Figure 3 the authors do examine neural correlates as a function of report, but I find this figure to be already at a very high level (mostly correlations are shown). Perhaps the authors could choose 1 SNR (that which generates an approximately equal split between aware and not) and show variance and mean responses as a function of report?

We changed the **Figure 3** so as to provide a more detailed visualization.

Fig.3.A. shows the distributions of activity across trials for one subject at the different SNRs. This illustrates that the distribution at SNR -9dB and -7dB in particular seems to reflect a mixture between two unimodal distributions, in line with a bifurcation model. SI.Fig3 further illustrates activity distributions for this subject, split as a function of report, along with similar graphs at the group level.

Fig.3.B. shows the evoked potentials separately for “heard” and “not heard” reports at SNR -7dB, the SNR for which the split between heard and not heard was the most balanced across the different subjects.

Finally, variance and mean responses as a function of report for each SNR are also shown in **Fig S2**.

I sincerely hope these comments help improve an already very interesting study.

We believe they do, and we are grateful for these comments.

Response to Reviewer #2:

The goal of this study - to discover "a universal brain signature of conscious processing independent of report" - is laudable, and of fundamental neuroscientific and psychological importance. On the face of it, it should also appeal to a broad general audience. Unfortunately however, I think this study falls short on a number counts and, as such, fails to reach its potential.

My comments (in no particular order):

One of the four key reasons that the authors give for conducting this study, is (Page 3) "First, it is unknown whether they {such non-linear input-output dynamics as a generic signature of conscious perception} generalize to other sensory modalities, or rather represent peculiarities of the visual system". Their response to this is to conduct their study entirely in the auditory domain, which is a shame because they really can't claim, based on their results, that these signatures are generalizable (i.e. to other domains). What they really need to do is to include one or more analogous experiments in other domains showing similar results. As such, their title "a UNIVERSAL brain signature..." is misleading and somewhat inappropriate. I might mind this less if they reconfigured their paper (and especially the title) to focus this on what it is - an *auditory* signature - but unfortunately, I think this would then rather seriously detract from the impact of the paper. The holy grail here is to truly find a *universal* signature of conscious processing and, given this limitation, this study falls well short of that.

We agree that our use of the term "universal" in the previous version of the manuscript might have been misleading. We therefore changed the title for:

Bifurcation in brain dynamics reveals a signature of conscious processing independent of report.

Beyond the title change, the new version of the manuscript now addresses this question in three ways:

1) Although we understand that using the word "universal" seems premature, it none-the-less remains that both the new framework that we developed and the results that we obtained constitute a decisive turn for generalizing bifurcation dynamics as a marker of the transition between conscious and unconscious processing, and hence progress towards identifying universal neural signatures of consciousness. We therefore improved the manuscript so as to further clarify the scope and relevance of the present contribution in relation to existing work. In particular we now explain more clearly: (1) why it was particularly critical with regards to current debates to probe the generality of bifurcation dynamics in the specific directions that we chose (Introduction page 3), and (2) why our new framework constitutes a key methodological advance for probing the generality of this marker, by providing analyses and models that are readily transposable to any type of stimulus, task or population (e.g. Discussion page 24).

2) Besides these improvements in our argumentation, the new analysis and control experiment (control experiment 1) that we conducted following reviewer #2's suggestions clearly demonstrates the potential of this new approach for predicting

conscious experience based on neural activity alone (see point 3 below, and Fig.3F, Fig.5C, Fig.S4D&E of the new version of the manuscript).

3) Finally, the new version of the manuscript includes another control experiment (control experiment 2), in which we generalize our results to non-speech auditory stimuli (simple tones; see Results section page 18, and Fig.S7 of the new version of the manuscript).

The authors state on Page 3. "Second, the stimulus presentation procedures are arguably complex, whereas a neural signature of conscious perception should apply to simpler cases more representative of natural perception" and on Page 14 "Here we show that these peculiar dynamics generalize to conscious perception in a new sensory modality, audition, and in a very simple paradigm, arguably closer to ecological stimulation conditions". I am not sure I buy this. The task they choose is undoubtedly 'simple', but I'm not sure its a "simpler case" than many of the visual tasks that have been used previously. Nor do I think that detecting vowels among SCN is "more representative of natural perception" than those other tasks (how often do we try to detect vowels among SCN in everyday life?). "arguably closer to ecological stimulation conditions" perhaps, but the authors need to qualify that, because it's not clear to me that it's really true.

In the present version of the manuscript we now clarify in what respect our stimulation procedure (stimulation around threshold) is simpler than visual masking or attentional blink - the two procedures that have been used in previous experiments on bifurcation dynamics - and why this is important with regards to current debates. Notably, it has been argued that the type of transition observed between conscious and non conscious processing might relate to the type of stimulus or protocol used, with simple stimuli at threshold yielding early and smooth transitions, and complex stimuli embedded in masking or attentional protocols yielding late and sharp transitions (Windey, Vermeiren et al. 2014, Windey and Cleeremans 2015). Furthermore, visual transients as in masking protocols or attentional blink protocol hardly ever occur in real life. There is a possibility that this in itself might be the origin of non-linearities; this possibility has to be discarded in order to argue that bifurcation dynamics relate to conscious processing in general.

These arguments are now better developed in the new version of the manuscript, Introduction page 3 and Discussion page 20.

On Page 3 the authors state "Finally, because consciousness is attached to a single mind, a neural signature of consciousness should ultimately be observable at the level of the individual and not only as an average across participants..." and on Page 10 "Remarkably these bifurcation dynamics could be detected at the individual level in all participants". Perhaps I haven't fully understood exactly what they have done to test these claims, but while I accept that they detected these patterns in all participants, I am not convinced that the results were *predictive* at the individual level (which is, I think, what they would like to claim). To demonstrate that, I would like to have seen an experiment where they actually used their EEG signature to predict whether a participant had consciously experienced any particular stimulus or not. I appreciate that the authors may have sort of done the statistical equivalent of this, but I don't believe it's what they did in any practical way. To do so, would make this paper accessible to a wider audience. Just to be clear, what I am proposing is that, to support their claims, they would carry out the passive version of their task and at various (e.g. random) times stop the task and tell the

participant whether or not they think they had perceived the stimulus that had just been presented. This might seem like a nuance, but it's critically important if the authors are going to make the sorts of claims about clinical utility that they make. e.g. Page 16 "The new signature described here opens important clinical perspectives for detecting the emergence of spontaneous conscious access to the external environment even in patients with deteriorated cognitive functions and at earlier stages of recovery". If this is true then their method needs to be able to predict, in an individual patient, whether a stimulus was consciously perceived or not.

We are grateful for these excellent suggestions; the suggested analysis is indeed a very relevant and direct way to test for our claim. Following this suggestion, we looked at how well we could predict subjective report based on preceding trial-by-trial activity and the bifurcation model, both in active sessions and in passive sessions ("surprise" mind-wandering questions). As detailed below, the results consolidate our claim and provide powerful evidence for the link between conscious content and our proposed neural signature.

In practical terms, **we performed two new analyses and acquired a new set of EEG data**. These new results are shown in Fig.3.F (predicting audibility response based on preceding neural activity in the active sessions), in Fig.5.C. (predicting mind-wandering response based on preceding neural activity in the passive sessions) and in Fig S6 panel F (complementary experiment with 7 new participants who only performed the passive condition). In all cases we show that applying the bifurcation model to brain activity recorded between 250-300ms and 700-800ms after stimulus presentation allows predicting, trial-by-trial, the subsequent audibility response (active sessions) or mind-wandering response (passive sessions). Below we detail these new analyses.

This "prediction analysis" was first performed in the **active sessions** (Methods page 52-53). Fitting the bifurcation model provides the expected distributions of activity for the two distinct "high state" and "low state", for each subject and each time point. We used these model distributions to compute, for each trial, the likelihood that the activity recorded at this trial belonged to the "high state" versus the "low state". Of note, these trial-by-trial Bayes Factors (BF: likelihood for high over likelihood for low) were computed without injecting any information about the subjects' response at any stage. Still, these BFs accurately predicted subsequent audibility reports, with higher BFs (more likely to belong to the "high state") leading to "heard" trials compared to "not-heard" trials (Fig.3F).

A similar prediction analysis was performed in the **passive sessions**. Indeed, the passive sessions already included random probing of the subjects' mental content, in the spirit of reviewer #2's suggestion ("mind-wandering probes", see Methods section), but these data were not analyzed in the previous version of the manuscript. We therefore conducted this analysis for the revision. In one fourth of the trials, at random, participants were asked to answer the question "What is on your mind just now?" by choosing among four options: "the sound", "the task", "my thoughts", "nothing / I feel sleepy" (nb: "the task" in the passive sessions referred to the quiz questions and visual reaction time tasks that were unrelated to the sounds). Importantly, this was a mind-wandering question and not a question directed on the perception of the sound specifically, to prevent subjects from carrying out a covert task on the sound. We

conducted the same analysis as the one described above (Methods page 53). The results are shown in Fig.5C: between 250 and 600ms post-stimulus, BFs associated with the activity recorded on each trial (likelihood that activity on this trial belongs to the “high state” versus “low state” of bifurcation dynamics) allows predicting whether the subject reports that they had the sound on their mind versus something else.

Finally, we performed the same analysis on a new set of data with 7 new participants who only performed the passive session (10 participants in total, 7 with sufficient data quality, see Methods page 54). This complementary experiment replicated what was observed in the passive sessions of the main experiment: Fig S6 panel F.

Page 16. "In current practice, in order to diagnose whether a non-communicating patient consciously perceives external stimuli, we mostly rely on detecting whether the patient can overtly or covertly perform a task on the stimulus (14, 43-46)" Mostly, yes, but there are a number of recent examples where this has been achieved in non-communicative patients without requiring any decision or 'active' response (e.g. Naci and colleagues, PNAS 2014).

We are grateful to the reviewer for pointing us to this very relevant literature. It is now acknowledged in the revised version of the manuscript (Discussion page 24)

Similarly Page 17 "This opens important perspectives for probing conscious perception in a variety of situations where the question of consciousness remains elusive, including clinical settings with non-communicating patients". In general, I feel that these statements would be better justified if the authors had actually shown that this was the case. It's all too easy to claim 'clinical relevance' but in practise this is a much harder problem than finding a tasks that works in healthy brains and assuming, therefore, that it would i) work in patients and ii) reveal anything about those patients. I do sympathize with these authors - clinical relevance is an important aim and a lofty goal, but I think their claims are premature.

First we would like to emphasize that we are well aware of the challenges of testing non-communicating patients, since, among the co-authors, Claire Sergent and Martina Corazzol are experienced researchers in this domain, with several published studies on vegetative and minimally conscious patients, testing interventions and diagnosis procedures (Faugeras, Rohaut et al. 2011, Faugeras, Rohaut et al. 2012, Corazzol, Lio et al. 2017, Sergent, Faugeras et al. 2017).

Far from being a remote intention, the way we constructed our protocol was tailored for allowing an easy transfer to testing patients: this directed our choice of auditory stimuli (since it is the easiest sensory modality that can be tested in patients), as well as the decision to embed our target sounds in noise, in order to mimic the important background noise that can be found in hospital rooms. Furthermore, our intention to transfer the protocol to patients was the reason why we used EEG instead of a more refined technique such as MEG, because EEG can be used at patient's bedside. Finally, the progression from reported to non-reported protocol was meant as a stepping stone for then being able to interpret brain activity in the absence of report in non-communicating patients.

As a result, we have already started transferring this protocol to testing non-communicating patients, in collaboration with two intensive care units in Paris and Lyon, with promising first results.

For all these reasons, we think it is justified to highlight the possible clinical relevance of this protocol, although we are obviously at an early stage. The revised version of the manuscript now includes a more detailed discussion of this point (pages 23-24).

Page 5 "Unlike the identification data, audibility ratings for single trials were not binary". Why wasn't this also true for identification? Wouldn't the authors have predicted that it would be true for both audibility ratings and identification?

There was an ambiguity in our formulation in the previous version of the manuscript; this distinction results from the construction of the tasks themselves: identification performance was assessed using a 2-alternative forced choice ("a" or "e"), whereas audibility report was assessed using a continuous scale.

We corrected this formulation in the new version of the manuscript, page 6:

Importantly, audibility was assessed using a continuous scale so we could derive a meaningful variability measure for behavior. In a 2-alternative forced choice task, such as the one used for identification, variability is maximal at 50% performance by construction (28). In contrast, the continuous measure of audibility imposed no such constraints, as variability could vary freely at all levels tested, provided they were sufficiently remote from floor or ceiling. Thus, any putative discontinuity in sensation may manifest itself by a burst in variability in the audibility judgments, following the modeling described in Fig 1 A-C (see also (16, 17)).

Page 11. "Neural responses due to decision-making should disappear in the absence of a task, but neural responses reflecting conscious access should persist". Yes, but how do you stop participants from making a decision anyway? In this situation (passively listening to vowels embedded in SCN) it seems likely that participants would naturally ask themselves the question from time to time "Did I just hear an 'E'?", particularly if they had previously been involved in the active version of the task. Again, this would be at least partially solved by a predictive study in which naive subjects were told to just do the passive task and then predictions were made about whether they perceived the stimuli or not.

Page 11. "Surprisingly, neither prediction was entirely fulfilled: late waveforms were still observed, but their topographies were very different from the sustained P3-like positivity characteristic of active listening (Fig. 4A)" This possibly confirms my previous point. Perhaps there is a little bit of decision making in the passive task?

We agree that the question of whether participants are performing the task covertly during the passive session is a crucial one.

The primary argument against the idea that people might actually do the task covertly during the passive session is that late topographies are different in passive and active sessions (see Figure 4). However we agree that this does not entirely rule out the possibility that they covertly performed part of the active task during the passive session: for example they might ask themselves whether the vowel was an "a" or an "e" without connecting this decision to a motor plan.

In order to address this question thoroughly **we followed the two suggestions of the reviewer:** (1) we conducted a **complementary analysis** of the main experiment, assessing the effect of task-order, and (2) we performed a **new experiment (Control experiment 1)** recruiting naïve subjects who performed only the passive task (methods p 54). These new results are shown in **Fig S6** and described on **pages 17-18** of the revised manuscript.

Many of my concerns are summarized by the authors cautious final statement on Page 13. "In conclusion the analysis of brain activity during the passive session suggests that, even in the absence of behavioral report, stimulation at threshold yields two types of neural response, presumably corresponding to trials with and without covert conscious perception" Given all my concerns above, I am not sure that *presumably* is enough to support the strong claims the authors are making here.

The new analyses and complementary experiments that were carried out following the reviewer's suggestions provide direct evidence that this late activity, characterized by bifurcation dynamics, does indeed directly predict whether the stimulus becomes conscious or not, independently of the task. Again we are grateful for the reviewer's suggestions, which considerably strengthened the demonstration.

Response to Reviewer #3:

This study exploits an interesting experimental design and advanced EEG analysis with the aim of distilling the neural events underlying conscious access by comparing measurements obtained in the presence (active condition) and in the absence (passive condition) of explicit reports. In line with previous studies employing no-report paradigms (see, for example, Pitts et al., 2014a and 2014b), Sergent and colleagues find that a prominent P3 wave associated with frontal sources of activity is absent or strongly reduced in the no-report (or passive) condition. This result is relevant as the presence of P3b wave associated with a robust activation of frontal cortical areas (obtained in active conditions) was previously proposed as a robust signature of conscious access. Interestingly, the author still find significant classification accuracy, in both the active and passive conditions, between single trials in which the target sound was present and trials in which the target sound was absent. They also show that the variability profile of the behavioral performance in the active condition correlate with the variability of the classifier performance, in a way that can be explained by a bifurcation dynamics model. Importantly, performing cross-classification analysis demonstrates that training the classifier in the passive session (sound present vs sound absent) allows them a significant decoder performance when testing the classifier in the active session.

I find the general approach original and the results potentially relevant for the field. However, there are several issues (both conceptual/interpretational and methodological) that the authors need to address in order to ensure that the results are solid and interpretable.

Major concerns

Conceptual/interpretational

1 – since the primary objective of the paper is to contrast neural responses in the presence and in the absence of report, it is crucial that the ERPs obtained in the active and passive conditions (at -5dB) are clearly presented and compared. Hence, Figure 4 should clearly show, in addition to scalp topographies, the classic traces (voltage vs time) normally employed to illustrate P3-like components. This display is needed to qualify statements that are, in the text, rather unclear and vague, such as “Surprisingly, neither prediction was entirely fulfilled: late waveforms were still observed, but their topographies were very different from the sustained P3-like positivity characteristic of active listening” (pag 10) or “It shows that the P300 is not a “pure” signature of conscious access, since its typical topography is not observed in the absence of report, but it is a composite waveform that does include the core signature of conscious access together with decision-making processes.” (Pag 10).

How does the time course of the two traces actually differ in the report and no-report condition? A simple plot of the trace is fundamental for the reader to have a sense of the main result. When plotting the traces obtained in the two conditions, please use the same Y-scales (also, remove autoscaling in the topographical plots of Fig.4a). I would then replace the qualitative statements reported above with a simple description of this fundamental figure. It is also very useful that the reader can directly compare these

waveforms to those obtained in previous studies, where the P3 has been used as a marker of consciousness in both healthy subjects and unresponsive brain-injured patients.

This has been implemented in the present version of the article, see Fig.4A. A corresponding sentence has been added to the text on page 14:

The evoked potentials at Cz further illustrate this point: the positive deflection observed from 250ms onwards in the active sessions, is replaced by a negative deflection in the passive sessions.

2- Also related to the interpretation, it is difficult to understand how the present result “validates a key prediction of the global workspace model of consciousness (17, 34, 35). In this model, conscious access arises through the broadcasting of local sensory information to a global network of areas”. For example, reference (35) clearly states that the GNW model predicts that conscious access coincides with the “ignition” of a large-scale prefronto-parietal network, whereas the present data suggest that large scale prefronto-parietal network activations are related to reporting but not to conscious access. The source modeling of passive condition (sound present vs absent) in Fig.4 shows a center of mass around the temporal opercular cortex with no local maxima in the frontal areas typically involved in the GNW model. Is there a late non-linear activation in frontal sources that is separate and temporally distinct from the temporo-parietal activation? For example, it would be important to show how do the time series of the currents (dipoles) in the temporal and the frontal cortex differ. The few sources that are weakly active in the inferior part of the frontal lobe are more likely to reflect the smearing typical of distributed inverse solutions rather than the ignition of strong independent prefrontal sources. This is another key point that needs to be clarified in the manuscript.

The new Figure 4 shows the time series of reconstructed activations in three areas within the temporal and frontal lobes (Fig.4D). It is described in page 15 of the Results section, and discussed in page 23:

[...] our results provide a nuanced picture about the role of the frontal lobe in conscious perception, which is currently hotly debated (49, 50). Source reconstruction suggests that some frontal areas might be an integral part of the core network for spontaneous conscious access even in the absence of an overt task, as attested by late and sustained activations in inferior frontal sources during passive listening (Fig.4.C and D). Other frontal areas, in contrast, completely disengaged during passive listening, such as the Supplementary Motor Area. This suggests that the frontal cortex may play a role even in spontaneous task-free conscious processing, and that performing a task further emphasizes frontal activity by recruiting additional frontal territories, notably related to motor planning as decisions are often followed by action. However, this proposition needs to be confirmed using techniques with better spatial resolution, such as fMRI.

3 – How can the authors exclude that the late activation is not a correlate of phonemic processing (even in the passive condition)? Indeed, late components (between 200 and 300 ms) very similar to the ones found in the present work are known to be involved in phonemic processing (Ditinger et al., 2018; Koerner et al., 2017). Why not using simpler tones or clicks with slightly different features as in previous works? Would late component be strongly reduced in this case, especially in the passive condition? This possibility, if not tested, must be acknowledged. More generally, at the present stage, I

would be very cautious in claiming that these components are a core signature of conscious access (abstract and elsewhere).

We implemented the suggested control experiment (Control experiment 2), recruiting 5 new participants who performed both an active and a passive session, for a total of 10 new EEG acquisitions, on a new version of the experiment where the vowels were replaced by simple tones. The stimuli were 2 tones, matched in duration to the vowels used in the main experiment (200ms). Tone 1 and tone 2 were pure tones with respective frequencies $f_1 = 1000\text{Hz}$ and $f_2 = 2236,1\text{Hz}$, and 30ms fade-in and fade-out. In the active sessions, instead of vowel recognition, participants were required to distinguish tone 1 and tone 2 (designated as “low-pitch” and “high-pitch” respectively).

All the details of this experiment can be found in Methods p55: “Control experiment 2: replacing vowels with simple tones”. The results are presented in Fig S7 and in the results section page 18, and suggest that our initial results do generalize to simpler stimuli such as tones.

4 – A further reason for caution: how can a novelty effect be excluded? N150-P250 potentials (vertex waves) can be recoded when stimuli (including auditory) occur with long interstimulus interval (tens of seconds) and then habituate becoming much smaller for interstimulus intervals down to 1 s (Mancini et al., eNeuro 2018). What was the interval between the occurrence of the target stimuli (vowels)? Was there a control for habituation at around 1 Hz? If not, it is important to acknowledge this potential confound due to parallel activation of saliency networks (Liang et al., Cer Cor 2013).

As detailed in the Methods section, the inter-stimulus interval in our experiment ranged between 5 sec and 9 sec. There was always an intervening response screen and task between two occurrence of the stimuli (the task was either related to the stimulus in the active sessions, or unrelated to the stimulus in the passive sessions). Finally the two vowels were equally likely. These conditions are therefore very remote from those needed to contrast habituation and novelty effects.

Furthermore, when comparing “heard” and “not heard” responses for identical stimulation values at threshold, there is no novelty confound since we compare activations to identical stimuli.

Methodological

1- the authors state that “from 250ms onwards, neural variability profiles correlated with the behavioral variability profiles” (Page 7, Line 23), and report four results with multiple comparisons correction and one without (300-400ms). I believe that performing multiple comparisons correction but not taking it into account when interpreting the results is rather misleading. In my opinion, in the case of non-significant results exact-corrected p-values should be reported and interpreted (with the additional information of significance when uncorrected). In a similar way, I believe that p-values by themselves are not very informative, and that they should be accompanied, when possible, by the corresponding statistics and degrees of freedom [and ideally effect sizes]).

We agree with the reviewer that there was a contradiction in applying a correction for multiple comparisons and not taking it into account in our interpretation of the results. In the present version of the paper, this analysis has been replaced by a more sensitive and more straightforward assessment of this neuro-behavioral correlation, using correlation coefficients instead of regression slopes; this **new analysis** confirms a significant correlation between the neural and behavioral variability profiles for all time-windows beyond 200ms, including the 300-400ms time window, even after correcting for multiple comparisons. We now report these statistics in more details, including effect sizes, on **page 9** of the present manuscript:

The trial-to-trial variability, in contrast, displayed either a flat or slightly increasing profile for latencies up to 200ms post-stimulus; then, from 200-250ms onwards, these profiles changed into a non-monotonic profile with a peak in variability around threshold SNR, and correlated significantly with the behavioral variability profiles measured for each individual on the audibility scale (Student t-tests on correlation coefficients between individual neural and audibility profiles of variability, see Methods: in the first three time windows, before 200ms, $t(19) = 0.12, -0.90, 0.71$ respectively, all p-FDR-corrected > 0.5 , effect sizes Cohen's $d < 0.2$; on the fourth time window 200-250ms $t(19) = 3.06$ p-FDR-corrected = 0.010, Cohen's $d = 0.68$; for the following time windows beyond 250ms, all $t(19) > 4.10$, all p-FDR-corrected < 0.01 , all Cohen's $d > 0.9$). This variability peak is consistent with bifurcation dynamics for processing stages beyond 250ms post-stimulus onset.

We also implemented more detailed reports of statistics throughout the manuscript, as requested by reviewer #3.

2- Similarly, the mixed effects model reported in (Page 8, Line 22) is not specified in the methods section. Did it include a random intercept only, random intercept and slope (etc)? Moreover, the results of this analysis (coefficients, degrees of freedom, standard deviations, post-hoc comparisons, etc.) are not reported. Did the authors check the residuals to see if assumptions were met? All this information is important in order to interpret the findings, given that significant results can be easily found with incorrect mixed effects models. Also, when interpreting this result, I assume the authors are focusing on the interaction, however, I believe that the explanation of why this is informative regarding the authors' hypothesis should be included in the text as it might not be self-evident to the reader.

We now include a full description of the mixed-effect model in the Methods section p 45-46:

Activity over time windows of interest: linear mixed effect model (Fig. 3B)

We conducted a linear mixed-effect model on mean activity across trials within a time window of interest (300-400ms), with conditions SNR x report (heard/not heard) (Fig. 3B). Our mixed-effects model included both random intercepts for each participant, as well as random slopes for report (heard/not heard) and SNR and their interaction: this model had significantly lower deviance (83.8) than a model that only included intercepts as random effects (deviance 212.0, $\chi^2_9 = 128.2, p < 10^{-15}$) or one that included only intercepts and main-effect slopes (deviance 154.9, $\chi^2_4 = 71.1, p < 10^{-13}$). We verified that the residuals are approximately normally distributed using a normal Q-Q plot. Our model showed significant effects of report (likelihood ratio test, $F_{1,24.3} = 30.3, p = 1.12 \times 10^{-5}$), SNR ($F_{1,20.5} = 54.7, p = 3.27 \times 10^{-7}$) and their interaction ($F_{1,19.4} = 7.86, p = 0.011$).

Also following the reviewer's advice, the results section now provides more details on the statistics and explains better why it is the interaction terms that is important in this analysis; see p.10-11:

The two models diverge on their predictions for the “not heard” trials. For the unimodal model, mean activity increases with stimulus strength for any trial, heard or not, consistent with a signal detection theory framework (28). By contrast, a bifurcation model predicts that, for “not heard” trials, mean activity should stay the same whatever the stimulus strength, and this activity should be equal to that observed when no stimulus is presented. For “heard” trials, all models predict that increasing stimulus strength should increase neural activity. In other words, the bifurcation model predicts an interaction between consciousness report (heard/not heard) and SNR, whereas the unimodal model predicts no interaction.

3 - In Figure 3A the authors present the results of the window from 300-400ms, which did not survive the multiple comparisons correction in the previous analysis. Why did they choose this particular time window? It seems like this choice was motivated by an a priori hypothesis, rather than in a data-driven manner. This needs to be clarified.

This problem is now solved by the new analysis we describe in response to methodological point 1.

4 - In the analysis of the correlation at every time point, the authors refer to the prediction values (as stated in the methods section) as “neural mean and variability profiles” (in the results section) and as “neural activity” (in the figure legend). And also state that “These prediction values are a trial-by-trial measure of target-related neural activity” (Page 7, Line 14). I believe that equating neural activity with the output of a classifier is misleading, given that classification performance can be influenced by non-neural factors. In addition, this refers to the discrimination between conditions, not to the activity directly associated with each of them. The authors should use a different wording or provide an explanation of why prediction values can be equated to neural activity.

We now make this point clear throughout the paper, starting with the legend and labels used in Fig.2, and in the text, result section page 8:

The MVPA transforms the multivariate EEG activity at each trial, each time point and for each participant into a representation best expressing the neural response to stimulus presence versus stimulus absence (each dot in the illustrative Fig.2.A.). Just as an evoked potential, the MVPA projection can be viewed as a measure of brain activity over a region of interest, but with the region of interest derived from the data to optimally distinguish the presence of a stimulus. This projected activity can be used to try and predict whether the stimulus was indeed present or absent on that trial: the distance from the decision criterion – i.e. the criterion that best separates stimulus present versus absent trials (the plain line in Fig.2.A.) – indicates the strength of the prediction towards stimulus presence (positive distance) or absence (negative distance). In the following, we term this distance the “projected neural activity” and use this continuous measure to assess variability across trials from the MVPA output.

5 - The authors mention that “some trials were composed of noise alone, without superimposed stimuli”, but the number of target-absent trials is not clearly reported,

which is important, given that class imbalances can bias the performance of a classifier. I believe that the number of observations of each class that were used with the classifiers (after trials rejection) should be clearly stated as it is a crucial information in order to evaluate the results.

We agree that this is crucial information. The same number of trials was used for each SNR condition, including the “noise only” condition (SNR -inf). This information was present in the previous version of the manuscript, but it might not have been explicit enough (Methods section “Each session contained 960 trials (2 vowels × 6 sound levels × 80 repetitions), grouped into 20 blocks of 48 trials (about 7 minutes per block”).

In the new version of the manuscript, we further highlighted this information:

Methods section > Stimuli:

The stimuli were presented at five levels of signal-to-noise ratio (-13dB, -11dB, -9dB, -7dB, -5dB). [...]. Some trials were composed of noise alone, without superimposed stimuli. We ran the same number of trials (160) for each of the stimulation levels (including “noise alone”).

Methods section > Data analysis > Preprocessing

On average across participants 94% ($\pm 4\%$) of trials were retained, giving an average of 150 trials per sound level condition (including “noise alone”; range [127-160]).

Methods section > Data analysis > Multivariate pattern analysis (MVPA)

It is important to highlight that there was always a similar number of trials (150 ± 6) in each of the two conditions that were compared (vowel absent versus vowel present at a specific SNR).

6 - The authors state that black contours of Figure 2B (and Figure 4A) indicate periods of above chance classification, however there are statistically significant classification with $AUC < 0.5$. I infer that the authors used a two-tailed t-test. In order to test if performance was above chance, the test should have been one-tailed. Additionally, the type of t-test used is not reported in the methods section and should be described (was it Student or Welch?). Moreover, the validity of these tests relies on the assumption of normality but this was not assessed. I would personally suggest to confirm that no violations of normality were present, to use a non-parametric approach, or to state why a parametric test would be preferred.

When reporting classification performance with temporal generalization, as we do here, we need to use a two-tailed test. Indeed, while training and testing at the same time point should yield either null or positive performance, it is not the case for temporal generalization. A classifier trained at time x might efficiently classify the trials' categories at time y but with reversed polarity, yielding significantly negative AUC. Negative AUCs are actually commonly observed in temporal generalization diagrams (King and Dehaene 2014): it denotes that some neural activity patterns can recur with reversed polarity. This precision has been added page 9 of the new version of the manuscript:

Below chance off-diagonal performance indicated that early patterns of activity recurred later with reverse polarity (King and Dehaene 2014).

We agree that a non-parametric test was actually preferable with these data, since normality is not guaranteed (especially on the diagonal). We therefore replaced our

Student t-tests with non-parametric sign tests. The new versions of Fig2B and 4A show the results of these tests. The legends of these figures now provides more details on the statistical methods used:

Black contours indicate periods where classification deviated significantly from chance at p corrected < 0.05 (non-parametric sign test (Gibbons and Chakraborti 2011), Benjamini-Hochberg FDR correction)

7 - The observation of significant misclassifications rises the question of why this could have happened? Do the authors have an explanation for this phenomenon? In addition, in the generalization across time analysis of Figure 2B there are significant classifications when training at around 250-500 and testing at 1750-2000ms (and vice versa). Can the authors comment on this discrimination between conditions that occurs almost 2 seconds after stimulus presentation?

The question of below-chance off-diagonal performance is already addressed in the previous point.

About late temporal generalization, we added the following comment in the new version of the manuscript, page 9:

Finally, temporal generalization indicated that some patterns of activity observed between 250 and 500ms, were reactivated around 1750-2000ms, probably in anticipation of the response, since the response screen appeared between 2000 and 3000ms.

8 - The colormap of the AUC shows discrete colors, but the AUC is a continuous variable. How were these colormaps constructed? Did the authors discretize these values?

We used a colormap with discrete levels for clarity, but we did not discretize the values.

9 - It is not clear to me how the analysis of the correlation between the neural variability profiles and the behavioral variability profiles was performed. In the methods section the authors state that the mean and variability profiles of the classification performance observed at each time window were regressed with the profile of mean and variability in audibility, at the subject level. Did this regressions include 5 observations only? The results section states a result at the group level, and mentions "t-tests on regression slope", what are the observations included in this test? Does this correspond to a test against 0 using the slopes of individual subjects? Please provide a more detailed description of this analysis.

These analyses are now described in more details in the new version of the manuscript, **Methods section, page 44 (time windows analysis shown in Fig 2 C):**

We also computed the correlation between the neural variability profiles and the behavioral variability profiles at each time window. At the level of individual subjects, we computed the correlation coefficient between the neural variability profile at each time window and the behavioral variability profile of this individual; these individual correlations were thus carried out on as many points as SNR levels in the experiment (6 in the active sessions). We then used Student t-tests on these individual correlation coefficients to assess whether they were significantly different from 0 at the group level.

Methods section, page 46-47 (time sample by time sample analysis shown in Fig 3D):

For each subject and at each time point we derived the correlation coefficient of the mean and variability profiles observed on prediction values with the subject's profile of mean audibility and variability in audibility collected during the active session (Fig.3D). These individual correlations were thus carried out on as many points as SNR levels in the experiment (6 in the active sessions). [...] We then used Student t-tests on these individual correlation coefficients to assess whether they were significantly different from 0 at the group level (Fig.3D and Fig.5B).

NB: questions 10-13 were identical to questions 1-4

14 - In the analysis between passive and active sessions, I suggest the author's to include a control analysis to verify how different were the late waveforms between subjects that performed the active condition first with respect to those who performed it last. I believe that this is an important point in order to avoid possible confounding factors in the interpretation of their results.

We agree that this is an important point. We therefore implemented the control analysis suggested by Reviewer #3. We also performed a new experiment (control experiment 1) recruiting naïve subjects who performed only the passive task. These new results are shown in Fig S6 and presented on pages 17-18 of the revised manuscript:

In order to further insure that this late activity genuinely reflected spontaneous conscious access and not some form of covert decision-making, we tested the effect of the order of the active and passive sessions. The activation patterns observed in the passive sessions were very similar in the groups of subjects who performed the passive session either first or second: there were no significant differences in topographies, time course, or the ability to predict mind-wandering responses based on the bifurcation model (Fig. S6. A-C). This suggests that there were no carry-over effect of the task performed in the active session on activity observed in the passive session. We also tested seven additional naïve subjects who only performed the passive session (see Methods). In these subjects we observed the same activation patterns, with late sustained activity starting around 250-300ms, similarly predicting mind-wandering responses (Fig. S6. D-F).

15 - The authors provide source localization results that were computed using a template and no electrode position digitization. This can create considerable imprecisions in the spatial localization activity of the sources. How were these results at the single subject level?

The primary objective of this study relates to brain dynamics, and we clearly state in the text that our source reconstruction results will need to be confirmed by techniques with better spatial resolution (discussion page 23).

For the reviewer's information, below we show the reconstructed sources at time t=330ms in the active and passive sessions (contrast stimulus present at high snr, -5dB, versus stimulus absent), in 3 subjects. These subjects were selected based on the quality of their ERP response in the active session. Importantly, the group result shown in the

article is a statistical test, so more robust to noise across subjects than a pure grand average of the reconstructed currents.

16 - In (Page 5, Line 22) there is a typo: it says Figs instead of Fig.

This has been corrected.

17 - Throughout the manuscript the authors report mostly binary p-value based statistical results ($p < 0.5 > p$). As mentioned above, I think that, when possible, including all the relevant information required to evaluate the results (statistic, degrees of freedom, etc) and performing the proper checks of each test's assumptions is of vital importance to be able to assess their validity.

Statistics are now reported with the requested details throughout the text.

Minor

- I think that the fact that the order of experimental sessions ("active" and "passive") was counterbalanced across participants should also be explained in the methods section corresponding to the experimental design, in order to avoid misconceptions on potential readers that might not read the article linearly.

This precision was already present in the first paragraph of the Methods section, as well as in the first paragraph of "brain dynamics in the passive sessions" in the Results section. In the revised version we also added it in the introduction page 4:

The order of active and passive sessions was counterbalanced across participants, to test whether performing the auditory task had any impact on subsequent passive listening.

- I believe that mentioning that the AUC is a measure of sensitivity is a terminological

imprecision. A measure of sensitivity would be, for instance, the True Positive Rate. I would suggest the authors to use a more precise wording.

We used the terminology of psychophysics and more particularly signal detection theory. In signal detection theory a distinction is made between sensitivity and hit rate (=true positive rate); see for example MacMillan & Creelman (page 6): « The hit rate, or any measure that depends on responses to only one of the two stimulus classes, cannot be a measure of sensitivity. To speak of sensitivity to a stimulus (as was done, for instance, in early psychophysics) is meaningless in the framework of detection theory ».

However it is true that in the terminology of medical diagnosis “sensitivity” refers to “true positive rate” (and specificity corresponds to correct rejection rate).

When possible we therefore preferred the term “performance” which is neutral in this respect.

References

- Aru, J., T. Bachmann, W. Singer and L. Melloni (2012). "Distilling the neural correlates of consciousness." *Neurosci Biobehav Rev* **36**(2): 737-746.
- Boly, M., M. Massimini, N. Tsuchiya, B. R. Postle, C. Koch and G. Tononi (2017). "Are the Neural Correlates of Consciousness in the Front or in the Back of the Cerebral Cortex? Clinical and Neuroimaging Evidence." *J Neurosci* **37**(40): 9603-9613.
- Corazzol, M., G. Lio, A. Lefevre, G. Deiana, L. Tell, N. Andre-Obadia, P. Bourdillon, M. Guenot, M. Desmurget, J. Luaute and A. Sirigu (2017). "Restoring consciousness with vagus nerve stimulation." *Curr Biol* **27**(18): R994-R996.
- Dehaene, S., C. Sergent and J. P. Changeux (2003). "A neuronal network model linking subjective reports and objective physiological data during conscious perception." *Proc Natl Acad Sci U S A* **100**: 8520-8525.
- Del Cul, A., S. Baillet and S. Dehaene (2007). "Brain dynamics underlying the nonlinear threshold for access to consciousness." *PLoS Biol* **5**(10): e260.
- Dykstra, A. R., P. A. Cariani and A. Gutschalk (2017). "A roadmap for the study of conscious audition and its neural basis." *Philos Trans R Soc Lond B Biol Sci* **372**(1714).
- Faugeras, F., B. Rohaut, N. Weiss, T. Bekinschtein, D. Galanaud, L. Puybasset, F. Bolgert, C. Sergent, L. Cohen, S. Dehaene and L. Naccache (2012). "Event related potentials elicited by violations of auditory regularities in patients with impaired consciousness." *Neuropsychologia* **50**(3): 403-418.
- Faugeras, F., B. Rohaut, N. Weiss, T. A. Bekinschtein, D. Galanaud, L. Puybasset, F. Bolgert, C. Sergent, L. Cohen, S. Dehaene and L. Naccache (2011). "Probing consciousness with event-related potentials in the vegetative state." *Neurology* **77**(3): 264-268.
- Gibbons, J. D. and S. Chakraborti (2011). *Nonparametric statistical inference*. Boca Raton, Taylor & Francis.
- Green, D. M. and J. A. Swets (1974). *Signal detection theory and psychophysics*. Huntington, N.Y., R. E. Krieger Pub. Co.
- Gutschalk, A., C. Micheyl and A. J. Oxenham (2008). "Neural correlates of auditory perceptual awareness under informational masking." *PLoS Biol* **6**(6): e138.
- King, J. R. and S. Dehaene (2014). "Characterizing the dynamics of mental representations: the temporal generalization method." *Trends Cogn Sci* **18**(4): 203-210.
- Kouider, S., C. Stahlhut, S. V. Gelskov, L. S. Barbosa, M. Dutat, V. de Gardelle, A. Christophe, S. Dehaene and G. Dehaene-Lambertz (2013). "A neural marker of perceptual consciousness in infants." *Science* **340**(6130): 376-380.
- Noel, J. P., Y. Ishizawa, S. R. Patel, E. N. Eskandar and M. T. Wallace (2019). "Leveraging Nonhuman Primate Multisensory Neurons and Circuits in Assessing Consciousness Theory." *J Neurosci* **39**(38): 7485-7500.
- Noel, J. P., D. Simon, A. Thelen, A. Maier, R. Blake and M. T. Wallace (2018). "Probing Electrophysiological Indices of Perceptual Awareness across Unisensory and Multisensory Modalities." *J Cogn Neurosci* **30**(6): 814-828.
- Odegaard, B., R. T. Knight and H. Lau (2017). "Should a Few Null Findings Falsify Prefrontal Theories of Conscious Perception?" *J Neurosci* **37**(40): 9593-9602.
- Sanchez, G., T. Hartmann, M. Fusca, G. Demarchi and N. Weisz (2020). "Decoding across sensory modalities reveals common supramodal signatures of conscious perception." *Proc Natl Acad Sci U S A* **117**(13): 7437-7446.
- Sergent, C., S. Baillet and S. Dehaene (2005). "Timing of the brain events underlying access to consciousness during the attentional blink." *Nat Neurosci* **8**(10): 1391-1400.

Sergent, C. and S. Dehaene (2004). "Is consciousness a gradual phenomenon? Evidence for an all-or-none bifurcation during the Attentional Blink." *Psychol Sci* **15**(11): 720-728.

Sergent, C., F. Faugeras, B. Rohaut, F. Perrin, M. Valente, C. Tallon-Baudry, L. Cohen and L. Naccache (2017). "Multidimensional cognitive evaluation of patients with disorders of consciousness using EEG: A proof of concept study." *Neuroimage Clin* **13**: 455-469.

Sergent, C. and L. Naccache (2012). "Imaging neural signatures of consciousness: 'What', 'When', 'Where' and 'How' does it work?" *Archives Italiennes De Biologie* **150**(2-3): 91-106.

Snyder, J. S., B. D. Yerkes and M. A. Pitts (2015). "Testing domain-general theories of perceptual awareness with auditory brain responses." *Trends Cogn Sci* **19**(6): 295-297.

Tsuchiya, N., M. Wilke, S. Frassle and V. A. Lamme (2015). "No-Report Paradigms: Extracting the True Neural Correlates of Consciousness." *Trends Cogn Sci* **19**(12): 757-770.

van Vugt, B., B. Dagnino, D. Vartak, H. Safaai, S. Panzeri, S. Dehaene and P. R. Roelfsema (2018). "The threshold for conscious report: Signal loss and response bias in visual and frontal cortex." *Science* **360**(6388): 537-542.

Windey, B. and A. Cleeremans (2015). "Consciousness as a graded and an all-or-none phenomenon: A conceptual analysis." *Conscious Cogn* **35**: 185-191.

Windey, B., A. Vermeiren, A. Atas and A. Cleeremans (2014). "The graded and dichotomous nature of visual awareness." *Philos Trans R Soc Lond B Biol Sci* **369**(1641): 20130282.

Reviewer #1 (Remarks to the Author):

I thank the authors for their excellent revisions. The paper is very interesting and I look forward to future efforts.

Reviewer #2 (Remarks to the Author):

The authors have gone to considerable lengths to address all of my suggestions and I am very impressed by the new version of the manuscript. I will reserve my comments for their responses to my specific suggestions, rather than those of all the Reviewers, although a brief look at all their responses suggests that they have been equally thorough throughout.

Specifically, I had 4 major issues.

1. The authors originally claimed to have found a universal signature of conscious processing, yet only tested their model in one domain (auditory). They have gone a long way to addressing my concern by conducting a completely new EEG control experiment, using pure tones instead of vowels. The results were replicated, showing that late bifurcation dynamics occur in different experimental settings, although this isn't quite what I asked for (to test the model in a different modality). Thus, it remains possible that this phenomenon is unique to the auditory modality (I count vowels and tones as belonging to the same modality in this context). That said, they have also reworked the text and, most importantly, changed the title to avoid using the adjective "universal", so I am satisfied overall with their response.

2. I was confused by some of their task descriptions e.g. describing their tasks as "simpler" than some other tasks (e.g. visual masking or attentional blink) that have been used in this context. The authors have now clarified what they mean by 'simpler' and why this is important with regards to current debates.

3. I suggested that, to really support their claims, they would carry out the passive version of their task and at various times stop the task and tell the participant whether or not they think they had perceived the stimulus that had just been presented. There are of course several versions of this manipulation that would arrive at the same conclusions. To address this, the authors looked at how well they could predict subjective report based on preceding trial-by-trial activity and the bifurcation model, both in active sessions and in passive sessions. They showed that applying the bifurcation model to brain activity recorded between 250-300ms and 700-800ms after stimulus presentation allowed them to predict, trial-by-trial, the subsequent audibility response (active sessions) or mind-wandering response (passive sessions), which is very good news. As they state themselves, "the results consolidate our claim and provide powerful evidence for the link between conscious content and our proposed neural signature." The authors then went one stage further than this: they performed the same analysis on a new set of data with 7 new participants who only performed the passive session. This experiment replicated what was observed in the passive sessions of the main experiment. Overall, I found these responses entirely satisfactory.

4. I was concerned about a comment in the original submission: "Neural responses due to decision-making should disappear in the absence of a task, but neural responses reflecting conscious access should persist". The crux of my concern was: how do you stop participants from making a decision anyway? The authors have addressed this concern to my satisfaction; indeed, they followed my two suggestions by reanalysing their experimental data to look at the effects of task order and they performed an additional experiment, recruiting naive participants who only performed the passive task. The results are persuasive.

Reviewer #3 (Remarks to the Author):

The authors performed control experiments and provided methodological clarifications that were generally satisfactory. This work of revision have been useful in consolidating and better defining the results, but have enhanced my initial concerns about the interpretation of these findings.

As detailed below, the authors tend to provide a representation of their empirical finding that is overtly speculative and non-parsimonious. The end result is that, although the findings are interesting, the conclusions of the paper are not justified by the empirical data. This is a major problem for a manuscript with a potentially high profile, which needs to be resolved.

The first problem involves the interpretation of ERP waveforms. During the previous round of reviews, I've asked the author to show a plain representation of traces (voltage vs time) and topographies, in line with those typically employed to illustrate P3-like components in previous works (some by the same authors). This request was made in order to provide an objective reference for the reader to judge and in order to qualify and resolve statements that were very ambiguous in the original text. These simple plots (Fig. 4A) now clearly show that the P3 component is completely absent in the passive conscious condition. Indeed, by comparing the active and the passive conditions at the same latencies it is clear that time-courses are different, that voltages have opposite polarity (!) and that topographies are not even comparable (a sustained auditory dipolar configuration in the passive condition). Bearing in mind that, just like any other ERP component, the P3 is jointly defined by a time-course, a polarity and a topography and given that none of these three criteria is met, the overwhelming evidence is that the P3 is absent when subjects consciously detect the stimulus in the passive condition. This is a key result and an important information to be highlighted, given that a series of papers over the last decade have proposed the P3 as a marker of conscious access to be also employed in both healthy subjects and patients.

Unfortunately, the authors are still very vague about this issue, both in the result section and in the discussion:

In the Results section (pag. 14), the authors write: "Thus, if the P3-like late positivity observed in active sessions is due to decision-making, as suggested by some authors (38, 39), it should disappear during passive listening. In contrast, if the P3-like activity is a signature of conscious access, it should remain in the passive sessions. Surprisingly, neither prediction was entirely fulfilled: as illustrated in Fig 4A, late waveforms were still observed in the passive sessions, but, importantly, their topographies were different from the sustained P3-like positivity that characterizes active listening...". Any reader, by inspecting Fig.4A would parsimoniously conclude that prediction 1 is fulfilled and that prediction 2 is falsified. Indeed, nobody in the ERP field would consider the presence of a late waveform per se as a sufficient criterion for a P3 unless the activation has a specific time-course, polarity and topography (by analogy, nobody would call an early activation per se a somatosensory N20, unless it appears as a sharp deflection at around 20 ms with a negative polarity and a dipolar topography over S1). In fact, observing in the passive condition a late EEG waveform with negative polarity and a sustained, bi-temporal auditory-like scalp topography - instead of the typical positive, central P3 waveform - necessarily points to neuronal events that are radically different from those of the P3. Now, I request that the authors make justice to this evidence and that they interpret their findings according to the basic standards of ERP component identifications. It is thus crucial that they make it clear in this section that a P3-like component is absent and that the empirical evidence is that prediction 1 is fulfilled, whereas prediction 2 is falsified.

A similar ambiguity must be resolved in the discussion section (pag. 22), where it reads: "The present results suggest that the P300 is not a "pure" signature of conscious access, since its typical topography is not observed in the absence of overt report. Rather, it is a composite waveform that does include the core signature of conscious access together with decision-making processes." This statement is illogical and sounds very confusing to the reader. Since the P3 (with its time-course, polarity and topography) is absent in the passive conscious condition, it cannot include any core signature of conscious access. Also here, it is mandatory that the authors stick to the empirical evidence by clearly stating that the P3 is only observed in the report condition and missing in the passive conscious condition, indicating that this component is not necessary for conscious access.

The second problem relates to source modeling analysis. In the first round of reviews, I asked the authors to show the time series of the currents (dipoles, or ROI) in the temporal and frontal cortex to demonstrate that there is a late distinct activation in frontal sources that is temporally independent from the temporoparietal activation. This is relevant not only because a distinct ignition of frontal activity is predicted by the GNW, but primarily to infer whether genuine

activations in the frontal cortex are present or not. Given the mathematical nature of distributed inverse solutions (such as the one used here) the effects of a strong local source is known to be artificially smoothed out and to leak into neighboring areas leading to distant currents that are spurious and biologically irrelevant, albeit statistically significant. Given the well-known presence of this artefactual spread of source activation, the existence of independent sources, or of the propagation of activity, is typically assessed by looking at the temporal evolution of local maxima, or by performing dipoles analysis. The original Fig4 already suggested that a strong local maximum was steadily located around the superior temporal gyrus (STG), with artifactual smearing in the surrounding cortex, due to the inherent smoothing of distributed solutions. The additional representation of the time-series now provided by authors in Fig4D, strongly reinforce this conclusion: in the passive condition, the currents in the inferior PFC are much smaller (about half) and have the same time-course as those detected in STG. Hence, no local maxima is present and no independent activation can be detected in the frontal cortex, highly consistent with artefactual smearing and with a lack of genuine activity in frontal regions. To offer a clear counterfactual, the case is completely different when looking at the active condition: here, the supplementary motor area shows indeed a strong local maximum with an independent time-course, consistent with a genuine neuronal activation of the frontal cortex. However, this activation (strikingly resembling the ignition proposed by the GNW model) is clearly task-related and not consciousness-related.

In essence, besides a lack of individual MRI and digitization of EEG electrodes (which are minimal standards for accurate EEG source localization), the empirical evidence in the passive condition is consistent with the presence of a sustained activation with local maxima around STG and no additional sources can be demonstrated. Above and beyond source modeling, this conclusion is confirmed by the static configuration of scalp voltages reported in Fig4A (passive) showing clear bilateral dipoles steadily located in temporal cortex from 200 ms to 600 ms consistent with persistent activity in the auditory cortex. This finding is interesting but cannot be possibly presented as an evidence of a late frontal, or even global, activation in the passive condition. Again, I have to ask the authors to stick to the empirical evidence, to describe their results parsimoniously and in accordance to accepted standards. Thus, sentences like "For both types of session and beyond 250ms, activations were observed in a broad network of areas encompassing auditory cortex and other temporal, parietal or frontal areas." (results, pag.15) or "both temporal and inferior-prefrontal regions responded with sustained activity beyond 250ms, with or without a task" (discussion pag.23) should be changed to indicate that the current data provide no evidence of frontal activation in the passive conditions. Further, the data shown in Fig.4 provide no evidence of global activation in general (the scalp and source configurations reported are typical for classic AEP reflecting activation of the primary auditory cortex). Hence, throughout the manuscript from the abstract to the discussion, the term frontal and/or global referred to the activation found in the passive conscious conditions is not supported by the data and must be removed: abstract, line 24 ("late sustained global activity"), discussion, pag 19 line 24 ("a global brain activity arising between 250 and 700ms post-stimulus") and the whole paragraph about the "global playground" (which, by the way, sounds rather speculative). The fact remains that these activations, plausibly localized in the temporal cortex, are sustained and that they are indicative of bifurcation dynamics, which is very interesting.

(As a side note, the selection of the sources corresponding to the inferior PFC seems rather odd, as the ellipse indicating them in the leftmost column of panel D in Figure 4, points to a region that is not showing any activity in the active condition and barely some activity in the passive condition. How were these sources selected? Why this particular portion of the iFC?)

Given all the above, the initial part of the discussion sounds somehow disconcerting to the reader who has a minimal knowledge about the fundamental predictions of the GNW model. Here the authors state (pag 20, line 1) that "In the following discussion, we show that these results validate and generalize dynamical predictions of the global neuronal workspace theory...". Although some predictions might be consistent with the results of bifurcation analysis, what stands out is that fundamental predictions of the GNW are not validated by the present data. One of such prediction is that the P3 is a neural correlate of conscious access (to be also employed as a marker of consciousness in patients), whereas the present results show that the P3 is absent in conscious trials. Another fundamental prediction of the GNW is that that conscious access coincides with the non-linear "ignition" of a largescale prefronto-parietal network, whereas the present data suggest

that large scale prefrontal activations can only be demonstrated in the case of reporting but not in the case of passive conscious access. If the authors really want to digress on the relationships between their empirical findings and the predictions of the GNW model, they need to be objective and clearly state that the data are not consistent with at least two of the fundamental predictions characterizing the GNW model. Theories are very useful when their predictions are validated but even more useful when their predictions are falsified.

In conclusion, I think that the present results (bifurcation, leading to a sustained local activation) are solid, interesting and that they might open novel lines of future research. However, the fact that they are currently presented in a way that is non-parsimonious and highly speculative (at times, ideological), hampers their empirical relevance and diminishes their scientific impact. Thus, I strongly encourage the authors to revise their text in order to solve this problem in order to valorize this important empirical contribution.

We are delighted that Reviewers 1 and 2 were completely satisfied with our revisions and we take great pride in their supportive appreciations. Again, thank you for the insightful comments that improved the manuscript considerably.

While Reviewer #3 found the additional data and rewriting generally satisfactory, s/he expressed concerns about our lack of clarity in stating that the P300 disappeared in the passive conditions, and asked us to stay closer to the data in the discussion. With hindsight, we agree with those concerns so we have revised the text accordingly. We provide a detailed response to Reviewer #3 below, which highlights the changes made to the manuscript.

We would like to thank again all three reviewers for a very helpful and constructive revision process. We now include them in the acknowledgment section.

Reviewer #1 (Remarks to the Author):

I thank the authors for their excellent revisions. The paper is very interesting and I look forward to future efforts.

Reviewer #2 (Remarks to the Author):

The authors have gone to considerable lengths to address all of my suggestions and I am very impressed by the new version of the manuscript. I will reserve my comments for their responses to my specific suggestions, rather than those of all the Reviewers, although a brief look at all their responses suggests that they have been equally thorough throughout.

Specifically, I had 4 major issues.

1. The authors originally claimed to have found a universal signature of conscious processing, yet only tested their model in one domain (auditory). They have gone a long way to addressing my concern by conducting a completely new EEG control experiment, using pure tones instead of vowels. The results were replicated, showing that late bifurcation dynamics occur in different experimental settings, although this isn't quite what I asked for (to test the model in a different modality). Thus, it remains possible that this phenomenon is unique to the auditory modality (I count vowels and tones as belonging to the same modality in this context). That said, they have also reworked the text and, most importantly, changed the title to avoid using the adjective "universal", so I am satisfied overall with their response.

2. I was confused by some of their task descriptions e.g. describing their tasks as "simpler" than some other tasks (e.g. visual masking or attentional blink) that have been used in this context. The authors have now clarified what they mean by 'simpler' and why this is important with regards to current debates.

3. I suggested that, to really support their claims, they would carry out the passive version of their task and at various times stop the task and tell the participant whether or not they think they had perceived the stimulus that had just been presented. There are of course several versions of this manipulation that would arrive at the same conclusions. To address this, the authors looked at how well they could predict subjective report based on preceding trial-by-trial activity and the bifurcation model, both in active sessions and in passive sessions. They showed that applying the bifurcation model to brain activity recorded between 250-300ms and 700-800ms after stimulus presentation allowed them to predict, trial-by-trial, the subsequent audibility response (active sessions) or mind-wandering response (passive sessions), which is very good news. As they state themselves, "the results consolidate our claim and provide powerful evidence for the link between conscious content and our proposed neural signature." The authors then went one stage further than this: they performed the same analysis on a new set of data with 7 new participants who only performed the passive session. This experiment replicated what was observed in the passive sessions of the main experiment. Overall, I found these responses entirely satisfactory.

4. I was concerned about a comment in the original submission: "Neural responses due to decision-making should disappear in the absence of a task, but neural responses reflecting conscious access should persist". The crux of my concern was: how do you stop participants from making a decision anyway? The authors have addressed this concern to my satisfaction; indeed, they followed my two suggestions by reanalysing their experimental data to look at the effects of task order and they performed an additional experiment, recruiting naive participants who only performed the passive task. The results are persuasive.

Reviewer #3 (Remarks to the Author):

The authors performed control experiments and provided methodological clarifications that were generally satisfactory. This work of revision have been useful in consolidating and better defining the results, but have enhanced my initial concerns about the interpretation of these findings.

As detailed below, the authors tend to provide a representation of their empirical finding that is overtly speculative and non-parsimonious. The end result is that, although the findings are interesting, the conclusions of the paper are not justified by the empirical data. This is a major problem for a manuscript with a potentially high profile, which needs to be resolved.

The first problem involves the interpretation of ERP waveforms. During the previous round of reviews, I've asked the author to show a plain representation of traces (voltage vs time) and topographies, in line with those typically employed to illustrate P3-like components in previous works (some by the same authors). This request was made in order to provide an objective reference for the reader to judge and in order qualify and resolve statements that were very ambiguous in the original text. These simple plots (Fig. 4A) now clearly show that the P3 component is completely absent in the passive conscious condition. Indeed, by comparing the active and the passive conditions at the same latencies it is clear that time-courses are different, that voltages have opposite

polarity (!) and that topographies are not even comparable (a sustained auditory dipolar configuration in the passive condition). Bearing in mind that, just like any other ERP component, the P3 is jointly defined by a time-course, a polarity and a topography and given that none of these three criteria is met, the overwhelming evidence is that the P3 is absent when subjects consciously detect the stimulus in the passive condition. This is a key result and an important information to be highlighted, given that a series papers over the last decade have proposed the P3 as a marker of conscious access to be also employed in both healthy subjects and patients.

Unfortunately, the authors are still very vague about this issue, both in the result section and in the discussion:

In the Results section (pag. 14), the authors write: "Thus, if the P3-like late positivity observed in active sessions is due to decision-making, as suggested by some authors (38, 39), it should disappear during passive listening. In contrast, if the P3-like activity is a signature of conscious access, it should remain in the passive sessions. Surprisingly, neither prediction was entirely fulfilled: as illustrated in Fig 4A, late waveforms were still observed in the passive sessions, but, importantly, their topographies were different from the sustained P3-like positivity that characterizes active listening...". Any reader, by inspecting Fig.4A would parsimoniously conclude that prediction 1 is fulfilled and that prediction 2 is falsified. Indeed, nobody in the ERP field would consider the presence of a late waveform per se as a sufficient criterium for a P3 unless the activation has a specific time-course, polarity and topography (by analogy, nobody would call an early activation per se a somatosensory N20, unless it appears as a sharp deflection at around 20 ms with a negative polarity and a dipolar topography over S1). In fact, observing in the passive condition a late EEG waveform with negative polarity and a sustained, bi-temporal auditory-like scalp topography - instead of the typical positive, central P3 waveform - necessarily points to neuronal events that are radically different from those of the P3. Now, I request that the authors make justice to this evidence and that they interpret their findings according to the basic standards of ERP component identifications. It is thus crucial that they make it clear in this section that a P3-like component is absent and that the empirical evidence is that prediction 1 is fulfilled, whereas prediction 2 is falsified.

We fully understand this concern, and, with hindsight, we fully agree that we were not sufficiently clear in stating that the P300 disappears under the passive condition. In fact, we also agree with this Reviewer that this observation is both clear-cut and possibly one of the most interesting and provoking finding of the study.

We are sorry that we might have come across as trying to obscure this obvious fact in any way in our formulation. If we did, it was truly unintentional, since, as the Reviewer notes, the figures speak for themselves. We hopefully corrected this possible bias in the new version of the manuscript, page 14:

The main hypothesis was that neural responses due to decision-making should disappear in the absence of a stimulus-related task, whereas neural responses reflecting core conscious access should persist. Thus, if late sustained activations only reflect decision-making, as suggested by some authors (13, 38), they should disappear during passive listening. In contrast, if late sustained activations reflect a signature of conscious access, they should remain in the passive sessions. Surprisingly, neither prediction was entirely fulfilled: as illustrated in Fig 4A, the P3-like positivity that characterized late activations during active listening disappeared under passive listening. This corroborates some previous observations (38-40). But,

interestingly, this did not mean that late activity disappeared altogether: indeed, passively listening to the sounds still evoked late sustained activations well beyond the initial auditory components, up until 700ms post-stimulus. Importantly, these activations had a different topography from what was observed in the active session, distinguishing them unequivocally from the P300. The evoked potentials at Cz further illustrate this point: the positive deflection observed from 250ms onwards in the active sessions, typical of P300, is replaced by a negative deflection in the passive sessions.

A similar ambiguity must be resolved in the discussion section (pag. 22), where it reads: “The present results suggest that the P300 is not a “pure” signature of conscious access, since its typical topography is not observed in the absence of overt report. Rather, it is a composite waveform that does include the core signature of conscious access together with decision-making processes.” This statement is illogical and sounds very confusing to the reader. Since the P3 (with its time-course, polarity and topography) is absent in the passive conscious condition, it cannot include any core signature of conscious access. Also here, it mandatory that the authors stick to the empirical evidence by clearly stating that the P3 is only observed in the report condition and missing in the passive conscious condition, indicating that this component is not necessary for conscious access.

We believe that this concern stems from our formulation, and not from actual differences in interpretation. The cross-classification analysis shown in Figure 4b and described on page 15 demonstrates that the late waveform observed in the passive condition is “included” in the late, P3-like waveform observed in the active condition. In other words, the P3-like waveform observed when making this very general contrast of stimulus presence versus absence during active sessions, is a composite waveform that reflects several overlapping components: a central positivity (P300 *stricto sensu*) on top of the late waveform observed in the passive condition, with its two positive temporal foci.

Taking the reviewer’s perspective, we understand that this notion of overlap had to be explained more clearly, so the following reformulation of the text aims to address this. Perhaps more problematic, we also failed to clearly acknowledge that this observation invalidates former key predictions of the global neuronal workspace theory about the P300. We also corrected this involuntary bias in the new version of the manuscript, page 22:

Our study also provides important elements for the current controversy about the P300 EEG component, which is debated as a potential signature of conscious access (13, 39-41). The present results show that conscious access in the absence of decision-making does not produce a P300 waveform, contrary to what has been proposed so far by the global neuronal workspace model (44). This observation is consistent with several other recent experimental results (39-41). But the present study also goes beyond these previous works by clarifying the relationship between the P300 and the signature of conscious access per se. Indeed, using a cross-classification analysis (Fig.4B) we could demonstrate that the late sustained waveform that signs conscious access in the absence of a task (as characterized by bifurcation dynamics), is included in the P3-like waveform observed in the active condition. In other words, the P3-like waveform observed when making this very general contrast of stimulus presence versus absence during active sessions, is actually a composite waveform that includes two overlapping components: the signature of conscious access per se, with its bilateral positivity, and an additional central positivity that corresponds to the P300 in a strict sense, which specifically reflects decision processes (51).

These results thus bring clarifications to previous debates. They show that, when no decision is required, the processes associated with conscious access per se can unfold in the absence of a P300. However, these “core” conscious access mechanisms still correspond to late and sustained activity, within the same time range as the P300. Finally, when a task is required, the additional P300 mechanisms that result from the task are concomitant and probably closely articulated with these conscious access mechanisms. Therefore, our results suggest that, while the P300 is not a signature of conscious access per se, it reflects decision processes that are closely associated with conscious access mechanisms.

Also, as noted by the Reviewer, our proposition about the global workspace and the global playground is very speculative. We believe speculations have their place in a discussion, but we agree that they should be presented as such, and be distinguished from straightforward interpretations of the results. We therefore further highlighted the speculative nature of this proposition in the revised version of the manuscript (p 23-24):

With this in hand, we would like to propose the following update to previous formulations of the global neuronal workspace model. The present results indicate that covert conscious access might be subtended not by a global workspace, but rather by a subset of it, which we may term a “global playground”. This “global playground” would be a broad network of areas among which sensory representations are shared and maintained for several hundreds of milliseconds, thus offering wider cognitive possibilities than automatic unconscious processing, but with no specific agenda. When a task is required, this global playground is augmented by decision-making processes and turns into a global workspace.

The second problem relates to source modeling analysis. In the first round of reviews, I asked the authors to show the time series of the currents (dipoles, or ROI) in the temporal and frontal cortex to demonstrate that there is a late distinct activation in frontal sources that is temporally independent from the temporoparietal activation. This is relevant not only because a distinct ignition of frontal activity is predicted by the GNW, but primarily to infer whether genuine activations in the frontal cortex are present or not. Given the mathematical nature of distributed inverse solutions (such as the one used here) the effects of a strong local source is known to be artificially smoothed out and to leak into neighboring areas leading to distant currents that are spurious and biologically irrelevant, albeit statistically significant. Given the well-known presence of this artefactual spread of source activation, the existence of independent sources, or of the propagation of activity, is typically assessed by looking at the temporal evolution of local maxima, or by performing dipoles analysis. The original Fig4 already suggested that a strong local maximum was steadily located around the superior temporal gyrus (STG), with artefactual smearing in the surrounding cortex, due to the inherent smoothing of distributed solutions. The additional representation of the time-series now provided by authors in Fig4D, strongly reinforce this conclusion: in the passive condition, the currents in the inferior PFC are much smaller (about half) and have the same time-course as those detected in STG. Hence, no local maxima is present and no independent activation can be detected in the frontal cortex, highly consistent with artefactual smearing and with a lack of genuine activity in frontal regions. To offer a clear counterfactual, the case is completely different when looking at the active condition: here, the supplementary motor area shows indeed a strong local maximum with an independent time-course, consistent with a genuine neuronal

activation of the frontal cortex. However, this activation (strikingly resembling the ignition proposed by the GNW model) is clearly task-related and not consciousness-related.

In essence, besides a lack of individual MRI and digitization of EEG electrodes (which are minimal standards for accurate EEG source localization), the empirical evidence in the passive condition is consistent with the presence of a sustained activation with local maxima around STG and no additional sources can be demonstrated. Above and beyond source modeling, this conclusion is confirmed by the static configuration of scalp voltages reported in Fig4A (passive) showing clear bilateral dipoles steadily located in temporal cortex from 200 ms to 600 ms consistent with persistent activity in the auditory cortex. This finding is interesting but cannot be possibly presented as an evidence of a late frontal, or even global, activation in the passive condition.

Again, I have to ask the authors to stick to the empirical evidence, to describe their results parsimoniously and in accordance to accepted standards. Thus, sentences like “For both types of session and beyond 250ms, activations were observed in a broad network of areas encompassing auditory cortex and other temporal, parietal or frontal areas.” (results, pag.15) or “both temporal and inferior-prefrontal regions responded with sustained activity beyond 250ms, with or without a task” (discussion pag.23) should be changed to indicate that the current data provide no evidence of frontal activation in the passive conditions. Further, the data shown in Fig.4 provide no evidence of global activation in general (the scalp and source configurations reported are typical for classic AEP reflecting activation of the primary auditory cortex). Hence, throughout the manuscript from the abstract to the discussion, the term frontal and/or global referred to the activation found in the passive conscious conditions is not supported by the data and must be removed: abstract, line 24 (“late sustained global activity”), discussion, pag 19 line 24 (“a global brain activity arising between 250 and 700ms post-stimulus”) and the whole paragraph about the “global playground” (which, by the way, sounds rather speculative). The fact remains that these activations, plausibly localized in the temporal cortex, are sustained and that they are indicative of bifurcation dynamics, which is very interesting.

(As a side note, the selection of the sources corresponding to the inferior PFC seems rather odd, as the ellipse indicating them in the leftmost column of panel D in Figure 4, points to a region that is not showing any activity in the active condition and barely some activity in the passive condition. How were these sources selected? Why this particular portion of the iFC?)

We recognize that this interpretation of late passive activations as only emanating from local temporal sources is entirely plausible at this stage. We now acknowledge this alternative interpretation in the present version of the manuscript. Still, the present results also undeniably provide elements in favor of the involvement of frontal activity during passive listening, and we now better detail the arguments in favor of this other interpretation. Finally, in the new version of the discussion we insist even more on the limits of EEG source reconstruction and the possibility of artifactual activations.

In the same vein, our results provide interesting new elements about the role of the frontal lobe in conscious perception, which is currently hotly debated (52, 53). According to our source reconstruction analysis, some frontal areas might be an integral part of the core network for spontaneous conscious access even in the absence of an overt task, as attested by late and sustained activations in inferior frontal sources during passive

listening (Fig4.C and D). Other frontal areas, such as the Supplementary Motor Area, in contrast, completely disengaged during passive listening.

At this stage, we cannot exclude the possibility that the frontal activations observed here under passive listening are an artifact of EEG-source reconstruction, merely reflecting a spillover of temporal sources. Indeed, and especially with EEG, source reconstruction is known to be susceptible to misattribution. Therefore, a possible interpretation of our results is that passive listening evokes late and sustained local activations within the temporal lobes, not necessarily connected to a wider network. And indeed, the topography of this late activity is evocative of focal temporal sources (Fig.4A), consistent with this interpretation.

At face value, however, the reconstruction analysis suggests that the frontal cortex plays a role even in spontaneous task-free conscious processing, and that performing a task further emphasizes frontal activity by recruiting additional frontal territories, notably related to motor planning. Additional observations are consistent with this interpretation: first the difference between the active and passive conditions is only visible in highly focal regions within the dorsal frontal areas (Fig.4C). So, if, as commonly admitted, active listening evokes a wide network of areas, then removing focal executive sources should still leave a wide network at play during passive listening. A second argument is independent from source reconstruction and its potential flaws, but rather relates to the dynamics of this late waveform associated with covert conscious access: late and sustained activity is typically associated with brain-wide activations and functional coupling, as suggested both by experimental work (54) and simulations of large scale network models of the primate brain (55).

In conclusion, the present results open an interesting alternative on the type of network subtending late sustained activations in the absence of report; this issue now needs to be addressed using spatially-resolved techniques such as fMRI.

Given all the above, the initial part of the discussion sounds somehow disconcerting to the reader who has a minimal knowledge about the fundamental predictions of the GNW model. Here the authors state (pag 20, line 1) that “In the following discussion, we show that these results validate and generalize dynamical predictions of the global neuronal workspace theory...”. Although some predictions might be consistent with the results of bifurcation analysis, what stands out is that fundamental predictions of the GNW are not validated by the present data. One of such prediction is that the P3 is a neural correlate of conscious access (to be also employed as a marker of consciousness in patients), whereas the present results show that the P3 is absent in conscious trials. Another fundamental prediction of the GNW is that that conscious access coincides with the non-linear "ignition" of a largescale prefronto-parietal network, whereas the present data suggest that large scale prefrontal activations can only be demonstrated in the case of reporting but not in the case of passive conscious access. If the authors really want to digress on the relationships between their empirical findings and the predictions of the GNW model, they need to be objective and clearly state that the data are not consistent with at least two of the fundamental predictions characterizing the GNW model. Theories are very useful when their predictions are validated but even more useful when their predictions are falsified.

It is true that, while the dynamical predictions of the GNW model received some important confirmations by the present study, some of its key predictions about the P300 were, in contrast, clearly invalidated. As detailed in the response to the previous

remarks, it is now unambiguously acknowledged throughout the text. We also emphasized this point at the outset of the discussion, page 20:

In the following discussion, we show that these results validate and generalize dynamical predictions of the global neuronal workspace theory, while introducing an important nuance between non-linear and bifurcation dynamics. We also show that other predictions of the global neuronal workspace model, relating to the role of the P300 waveform, are invalidated by our results, and we propose an update of the model that accounts for these new findings. More generally, we show that the present results can help advance several prominent debates about the neural basis of consciousness. Finally we discuss the perspectives of this new approach, notably in terms of diagnosing consciousness in non-communicating patients.

In conclusion, I think that the present results (bifurcation, leading to a sustained local activation) are solid, interesting and that they might open novel lines of future research. However, the fact that they are currently presented in a way that is non-parsimonious and highly speculative (at times, ideological), hampers their empirical relevance and diminishes their scientific impact.

Thus, I strongly encourage the authors to revise their text in order to solve this problem in order to valorize this important empirical contribution.

With hindsight we recognize that the previous version of the text contained some biases in the discussion of the experimental data. We sincerely thank the Reviewer for helping us provide a more balanced view overall. This is an important asset of the reviewing process. We do believe, with the Reviewer, that this more neutral account can only broaden the appeal of the manuscript and make it more valuable for follow-up studies, which may or may not share our underlying theoretical framework. Finally, we hope the manuscript will open up further scientific discussions on a crucial topic for which we obviously share the same interest as the Reviewer.

Reviewer #3 (Remarks to the Author):

I have appreciated the exchange with the authors, which results in a framing that is more consistent with the empirical data. As briefly outlined below, there remain two inconsistencies that are potentially confusing for readers. Resolving these two issues is easy but important given the high technical and scientific quality of the study.

In pag. 14 it is now stated that "...if late sustained activations only reflect decision-making, as suggested by some authors (13, 38)..." However, in references 13 and 38 it is not suggested that late sustained activations only reflect decision-making. In fact, the authors of these papers contemplate the presence of sustained recurrent activity as a neural correlate of consciousness perception. What they rather suggest, also in refs 13 and 14, is that the P3 and frontal activations reflect decision making in report conditions (a prediction confirmed by the present results). It is important that the opinions expressed in ref 13 and 14 are fairly represented. This also implies stating that prediction 1 is supported by the present data (rather than that "neither prediction was entirely fulfilled"), as I have already suggested in the previous round.

Regarding source localization, I appreciate that the authors concur that the present data (local maxima in the parietal lobe) do not provide evidence for (nor conclusively rules out) a global activation in the passive condition and that "this issue now needs to be addressed using spatially-resolved techniques such as fMRI". Therefore, as already suggested in the previous round, it is important that the abstract reflects this state of affairs and that the term global is removed from line 24.

Response to final comments from Reviewer #3

Reviewer #3 (Remarks to the Author):

I have appreciated the exchange with the authors, which results in a framing that is more consistent with the empirical data. As briefly outlined below, there remain two inconsistencies that are potentially confusing for readers. Resolving these two issues is easy but important given the high technical and scientific quality of the study.

In pag. 14 it is now stated that “...if late sustained activations only reflect decision-making, as suggested by some authors (13, 38)...” However, in references 13 and 38 it is not suggested that late sustained activations only reflect decision-making. In fact, the authors of these papers contemplate the presence of sustained recurrent activity as a neural correlate of consciousness perception. What they rather suggest, also in refs 13 and 14, is that the P3 and frontal activations reflect decision making in report conditions (a prediction confirmed by the present results).

It is important that the opinions expressed in ref 13 and 14 are fairly represented. This also implies stating that prediction 1 is supported by the present data (rather than that “neither prediction was entirely fulfilled”), as I have already suggested in the previous round.

References 13 and 14 have now been removed from this part of the text to avoid misrepresenting the opinion of the authors of these studies. The text now states:

The main hypothesis was that neural responses due to decision-making should disappear in the absence of a stimulus-related task, whereas neural responses reflecting core conscious access should persist. Thus, if late sustained activations only reflect decision-making, they should disappear during passive listening. In contrast, if they reflect a signature of conscious access, they should remain in the passive sessions.

Now that these references have been removed, it is clear that our prediction 1 only refers to the simple, less nuanced option that, if late activity solely reflects decision making, it should disappear with the absence of a task. It is thus legitimate to say that neither this prediction, nor the opposite one (late sustained activity should be maintained) is verified by our results.

This simple option 1 really was an option that we considered when conducting these experiments, as stated from the introduction. These two possibilities represented the most straightforward alternative, before experimental results helped us refine the options. For this reason we wish to maintain this rhetorical part of the text, which reflects our reasoning progression.

Regarding source localization, I appreciate that the authors concur that the present data (local maxima in the parietal lobe) do not provide evidence for (nor conclusively rules out) a global activation in the passive condition and that “this issue now needs to be addressed using spatially-resolved techniques such as fMRI”. Therefore, as already suggested in the previous round, it is important that the abstract reflects this state of affairs and that the term global is removed from line 24.

The term “global” has been removed from that line (see new abstract):

« the same stimulus gives rise to late sustained activity on some trials, and not on others. »